# Optimizing Information-theoretical Generalization Bounds via Anisotropic Noise in SGLD

**Bohan Wang**
University of Science & Technology of China
Microsoft Research Asia

**Huishuai Zhang**[*]
Microsoft Research Asia

**Jieyu Zhang**
University of Washington
Microsoft Research Asia

**Qi Meng**
Microsoft Research Asia

**Wei Chen**[*]
Microsoft Research Asia

**Tie-Yan Liu**
Microsoft Research Asia

## Abstract

Recently, the information-theoretical framework has been proven to be able to obtain non-vacuous generalization bounds for large models trained by Stochastic Gradient Langevin Dynamics (SGLD) with isotropic noise. In this paper, we optimize the information-theoretical generalization bound by manipulating the noise structure in SGLD. We prove that with constraint to guarantee low empirical risk, the optimal noise covariance is the square root of the expected gradient covariance if both the prior and the posterior are jointly optimized. This validates that the optimal noise is quite close to the empirical gradient covariance. Technically, we develop a new information-theoretical bound that enables such an optimization analysis. We then apply matrix analysis to derive the form of optimal noise covariance. Presented constraint and results are validated by the empirical observations.

## 1 Introduction

Generalization ability is one of the core questions in learning theory [4], but remains unclear for deep learning models [17, 38]. Existing generalization bounds based on the capacity control become vacuous for practical deep learning models due to the over-parameterization property [1, 26, 39].

From the information theoretical perspective, recent works [30, 37] bound the generalization error by the mutual information between the dataset and the learned parameters. This result reveals that good generalization occurs when the learned parameters do not depend on a specific dataset too much, which is intuitively reasonable and closely related with the idea of algorithm stability [6, 15, 19, 24, 20] and differential privacy [8, 10]. Meanwhile, based on information-theoretic metrics, one can analyze general classes of updates and models, e.g., stochastic iterative algorithms for non-convex objectives, hence applicable to deep learning. It has been shown that the information theoretical bounds are non-vacuous and closely related with the real generalization error even in deep learning [14, 26, 9, 39].

Notably, the information theoretical bound is realized in [28] via decomposing the mutual information across iterations. This framework can perform a step-wise analysis of Stochastic Gradient Langevin Dynamics (SGLD) [33, 29], by evaluating the mutual information conditional on the previous learned

---

[*]Corresponding Author

35th Conference on Neural Information Processing Systems (NeurIPS 2021).

parameters at each step. We note that the noise added in SGLD is critical for both the mutual information evaluation and the empirical risk minimization. Most existing work focus on SGLD with constant and isotropic noise covariance [22, 25, 14] due to the technical difficulty. However, it is observed that the test accuracy of SGLD with isotropic noise has a considerable gap compared to that of the widely-used Stochastic Gradient Descent (SGD) [40]. This empirical gap motivates us to consider the following question:

*Can we find the optimal noise added in SGLD in terms of generalization?*

An affirmative answer will lead to an algorithm imitating SGD better, which helps us to better understand the generalization behavior of SGD.

Specifically, we propose to optimize the structure of the noise in SGLD such that the generalization bound is minimized while a low empirical risk is guaranteed. To this end, we first show that the trace of the noise covariance in SGLD is a valid constraint that governs the empirical risk behavior both theoretically and empirically. Then we devise a new information theoretical generalization bound that are parallel to those bounds in [25], but facilitate the derivation of the optimal noise. With these technical preparations, we prove that when jointly optimizing both the prior and the posterior, the optimal noise covariance is the square root of the expected gradient covariance, i.e., their eigenvectors are the same and the corresponding eigenvalues of the former are the square root of the latter, and the optimal prior recovers the prior in [25]. This indicates the optimal noise covariance of SGLD would be quite close to the empirical gradient covariance, i.e., the noise covariance of SGD, because of the concentration of measure.

Our result lends support to the belief that the noise introduced by Stochastic Gradient Descent (SGD) is superior to the isotropic noise, which has been widely observed [18, 36, 40, 38]. As an illustrative example, we plot the generalization errors of SGD, SGLD with the isotropic noise and SGLD with the optimal noise in Figure 1, where their training curves behave almost the same (do not show here). We can see the optimal noise captures the behavior of SGD much better than the isotropic noise.

Specifically, our contribution can be summarized as follows:

1. We formulate a problem of finding the optimal noise covariance by optimizing an information-theoretical bound;
2. We develop a new information-theoretical bound to facilitate the analysis of the above optimization problem;
3. We obtain the optimal structure of the noise covariances, and demonstrate the similarity to empirical/expected gradient covariance;

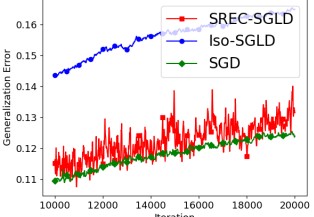

Figure 1: Generalization errors of SGD, SGLD with the isotropic noise (Iso-SGLD) and SGLD with the optimal noise (SREC-SGLD).

## 1.1 Related Works

**Information-theoretical bounds.** Recently, researchers [37, 30, 28] propose to bound the generalization error by the mutual information between output hypothesis and input samples. Negrea et al. [25] further tighten the bound by designing a data-dependent prior. Following [25], Haghifam et al. [14] obtain comparable results through conditional mutual information [31]. Other related work [2, 16, 5, 13] tightens the information theoretical generalization bounds from different perspectives. There are also high probability generalization bound [24, 26, 21] obtained by combining information theory [7] and PAC Bayesian framework [27, 33].

**Effects of the noise covariance.** SGD achieves excellent performance in terms of generalization error in deep learning. In contrast, one recent work [40] demonstrates empirically that even with well-tuned scaling, isotropic SGLD still achieves much worse generalization error than mini-batch SGD. They obtain an optimal covariance matrix (low-rank approximation of Hessian) in terms of *escaping efficiency* at local minima. Furthermore, [34] show empirically large-batch SGD with diagonal Fisher Gaussian noise can recover similar validation performance as small-batch SGD. However, both the noise covariance investigated by [40] and [34] has gap with that of SGD. It has also been proven [23] that the stationary distribution of state-dependent SGLD with the empirical loss approximated by quadratic function obeys the power law, and has a better escaping efficiency than SGLD with the isotropic noise.

## 2 Preliminaries

**Notations.** Here we briefly introduce the notations which will be used throughout this paper.

- (Set operation) For a positive integer $N$, we use $[N]$ to denote the index set $\{1, 2, \cdots, N\}$, and use $\boldsymbol{V}_{[T]}$ to denote the set $\{\boldsymbol{V}_t\}_{t=1}^T$. For a finite set $\boldsymbol{S} = \{\boldsymbol{z}_1, \cdots, \boldsymbol{z}_N\}$, $|\boldsymbol{S}|$ is the cardinality of $\boldsymbol{S}$, and $\boldsymbol{S}_J \triangleq \{\boldsymbol{z}_i\}_{i \in J}$ is a subset of $\boldsymbol{S}$ with indices in set $J \subset [N]$.
- (Probability) $\boldsymbol{z} \sim \boldsymbol{S}$ denotes that $\boldsymbol{z}$ is uniformly sampled from the set $\boldsymbol{S}$. For two random variables $\boldsymbol{X}$ and $\boldsymbol{Y}$, we denote the conditional distribution of $\boldsymbol{X}$ given $\boldsymbol{Y}$ as $\mathbb{P}^{\boldsymbol{Y}}(\boldsymbol{X})$ and denote the conditional expectation of $\boldsymbol{X}$ given $\boldsymbol{Y}$ as $\mathbb{E}^{\boldsymbol{Y}}(\boldsymbol{X})$.
- (Matrix) We use $\mathbb{I}_{d \times d}$ for the $d \times d$ identity matrix, abbreviated as $\mathbb{I}$ when dimension is clear. For a differentiable function $f$, we denote the gradient of $f$ at point $\boldsymbol{W}$ as $\nabla f(\boldsymbol{W})$. For a positive semi-definite matrix $\boldsymbol{A} \in \mathbb{R}^{d \times d}$, we say $\boldsymbol{B} = \boldsymbol{A}^{\frac{1}{2}} \in \mathbb{R}^{d \times d}$ if $\boldsymbol{B} \cdot \boldsymbol{B} = \boldsymbol{A}$.

**Supervised Learning.** In this paper, we focus on the supervised learning. It conducts the empirical risk minimization (ERM) over training data: $\text{Minimize}_{\boldsymbol{W}} \, \mathcal{R}_{\boldsymbol{S}}(\boldsymbol{W}) \triangleq \frac{1}{|\boldsymbol{S}|} \sum_{\boldsymbol{z} \in \boldsymbol{S}} \ell(\boldsymbol{z}; \boldsymbol{W})$, where $\boldsymbol{W} \in \mathbb{R}^d$ is the parameter of the model, $\boldsymbol{S}$ is the training set with each data point i.i.d. sampled from a distribution $\mathcal{D}$, $\ell$ is the (individual) loss function. A *stochastic algorithm* solves the ERM by outputting a distribution $Q^{\boldsymbol{S}}$ of parameter $\boldsymbol{W}$. The *generalization error* measures the gap between the population risk and the training risk as

$$\text{Gerr} \triangleq \mathbb{E}_{\boldsymbol{S}, Q^{\boldsymbol{S}}} \left[ \mathcal{R}_{\mathcal{D}}(\boldsymbol{W}) - \mathcal{R}_{\boldsymbol{S}}(\boldsymbol{W}) \right]. \tag{1}$$

**SGD and SGLD.** The update rule of Stochastic Gradient Descent (SGD) at step $t$ is defined as

$$\boldsymbol{W}_t \leftarrow \boldsymbol{W}_{t-1} - \eta_t \nabla \mathcal{R}_{\boldsymbol{S}_{\boldsymbol{V}_t}}(\boldsymbol{W}_{t-1}), \tag{2}$$

where $\boldsymbol{V}_t$ is sampled uniformly without replacement from $[N]$ with size $b_t$. Given $\boldsymbol{W}_{t-1}$, $\boldsymbol{W}_t$ is random with

$$\mathbb{E}\left[\boldsymbol{W}_t | \boldsymbol{W}_{t-1}\right] = \boldsymbol{W}_{t-1} - \eta_t \nabla \mathcal{R}_{\boldsymbol{S}}\left(\boldsymbol{W}_{t-1}\right), \; \text{Cov}\left[\boldsymbol{W}_t | \boldsymbol{W}_{t-1}\right] = \frac{N - b_t}{b_t(N-1)} \boldsymbol{\Sigma}_{\boldsymbol{S}, \boldsymbol{W}_{t-1}}^{sd},$$

where $\boldsymbol{\Sigma}_{\boldsymbol{S}, \boldsymbol{W}_{t-1}}^{sd} \triangleq \text{Cov}_{\boldsymbol{z} \sim \boldsymbol{S}}\left[\nabla \mathcal{R}_{\boldsymbol{z}}(\boldsymbol{W}_{t-1})\right]$ is the covariance of $\nabla \mathcal{R}_{\boldsymbol{z}}$ with $\boldsymbol{z}$ single drawn from $\boldsymbol{S}$. The superscript $^{sd}$ means "single draw". Similarly, we define the population covariance of gradient as $\boldsymbol{\Sigma}_{\boldsymbol{W}}^{\text{pop}} \triangleq \text{Cov}_{\boldsymbol{z} \sim \mathcal{D}}\left[\nabla \mathcal{R}_{\boldsymbol{z}}(\boldsymbol{W})\right]$. We define the Hessian of $\mathcal{R}_{\boldsymbol{S}}(\boldsymbol{W})$ at point $\boldsymbol{W}$ as $\mathcal{H}_{\boldsymbol{S}, \boldsymbol{W}}$.

The Stochastic Gradient Langevine Dynamics (SGLD) is given by

$$\boldsymbol{W}_t \leftarrow \boldsymbol{W}_{t-1} - \eta_t \nabla \mathcal{R}_{\boldsymbol{S}_{\boldsymbol{V}_t}}(\boldsymbol{W}_{t-1}) + \mathcal{N}(\boldsymbol{0}, \boldsymbol{\Sigma}_t(\boldsymbol{S}, \boldsymbol{W}_{t-1})), \tag{3}$$

where $\boldsymbol{\Sigma}_t(\boldsymbol{S}, \boldsymbol{W}_{t-1}) \in \mathbb{R}^{d \times d}$ is a positive semi-definite matrix with dependence on $\boldsymbol{S}$ and $\boldsymbol{W}_{t-1}$. SGLD (Eq. 3) with $\boldsymbol{\Sigma}_t(\boldsymbol{S}, \boldsymbol{W}_{t-1}) = c_t \mathbb{I}$ where $c_t$ a positive constant, is called *isotropic* SGLD, and SGLD with $\boldsymbol{\Sigma}_t(\boldsymbol{S}, \boldsymbol{W}_{t-1})$ dependent on $\boldsymbol{W}_{t-1}$ is called *state-dependent* SGLD.

**Statistics for iterative algorithms.** Both SGD and SGLD are *stochastic iterative algorithms* [11]. Specifically, the update rule at step $t$ of a stochastic iterative algorithm $\mathcal{A}$ can be generally characterized as $\boldsymbol{W}_t = \mathcal{M}_t(\boldsymbol{W}_{t-1}, \boldsymbol{S}, \boldsymbol{V}_t)$, where $\boldsymbol{V}_t$ is the *auxiliary random variable* at step $t$, e.g., $\boldsymbol{V}_t$ in Eq.(2) and Eq.(3). We use $Q^{\boldsymbol{S}, \boldsymbol{V}_{[T]}}$ to denote the joint distribution of $(\boldsymbol{W}_t)_{t=0}^T$ conditional on $\{\boldsymbol{S}, \boldsymbol{V}_{[T]}\}$, $Q_{i:j}^{\boldsymbol{S}, \boldsymbol{V}_{[T]}}$ to denote the joint distribution of $(\boldsymbol{W}_t)_{t=i}^j$ conditional on $\{\boldsymbol{S}, \boldsymbol{V}_{[T]}\}$, and $Q_{j|j-1}^{\boldsymbol{S}, \boldsymbol{V}_{[T]}}$ to denote the distribution of $\boldsymbol{W}_j$ conditional on $\{\boldsymbol{W}_{j-1}, \boldsymbol{S}, \boldsymbol{V}_{[T]}\}$.

**Decomposition of KL divergence.** The following Lemma is extensively used throughout this paper.

**Lemma 1** (Proposition 2.6, [25]). *Let $Q_{0:T}$ and $P_{0:T}$ are two probability measures on $\mathbb{R}^{d \times (T+1)}$ with $Q_0 = P_0$. Then, the KL divergence between $Q_{0:T}$ and $P_{0:T}$ can be decomposed into*

$$\text{KL}(Q_{0:T} || P_{0:T}) = \sum_{t=1}^T \mathbb{E}_{Q_{0:t-1}} \left[ \text{KL}\left(Q_{t|[t-1]} || P_{t|[t-1]}\right) \right].$$

**Information theoretical generalization bound.** Several existing information-theoretical bounds [14, 25] share similar framework. We state one representative proposed in [25].

**Proposition 1** (Theorem 2.5 in [25]). *Let $\boldsymbol{S}$ be the data set i.i.d. sampled from $\mathcal{D}$, and let the loss function $\ell$ be $[a_1, a_2]$ bounded. Let $\mathcal{A}$ be an algorithm with update rule*

$$\boldsymbol{W}_t \leftarrow \mathcal{M}_t(\boldsymbol{W}_{t-1}, \boldsymbol{S}, \boldsymbol{V}_t), \boldsymbol{W}_0 \sim \mathcal{W}_0.$$

*Let $\mathcal{B}$ be another stochastic iterative optimization algorithm with update rule*

$$\boldsymbol{W}_t \leftarrow \tilde{\mathcal{M}}_t(\boldsymbol{W}_{t-1}, \boldsymbol{S}_{\boldsymbol{J}}, \boldsymbol{V}_t, \boldsymbol{J}), \boldsymbol{W}_0 \sim \mathcal{W}_0.$$

*where $\boldsymbol{J}$ is sampled uniformly without replacement from $[N]$ with size $N-1$. Given $\boldsymbol{S}$, $\boldsymbol{J}$, and $\boldsymbol{V}_{[T]}$, denote the joint distribution of $(\boldsymbol{W}_t)_{t=1}^T$ of $\mathcal{A}$ as $Q^{\boldsymbol{S}, \boldsymbol{V}_{[T]}}$, and the joint distribution of $(\boldsymbol{W}_t)_{t=1}^T$ of $\mathcal{B}$ as $P^{\boldsymbol{J}, \boldsymbol{S}_{\boldsymbol{J}}, \boldsymbol{V}_{[T]}}$. Then, for any $\tilde{\mathcal{M}}_{[T]}$, the generalization error of $\mathcal{A}$ can be bounded as:*

$$\mathbb{E}_{\boldsymbol{S}, \boldsymbol{V}_{[T]}} \left[ \mathcal{R}_{\mathcal{D}} \left( Q_T^{\boldsymbol{S}, \boldsymbol{V}_{[T]}} \right) - \hat{\mathcal{R}}_S \left( Q_T^{\boldsymbol{S}, \boldsymbol{V}_{[T]}} \right) \right] \leq \mathbb{E}_{\boldsymbol{S}, \boldsymbol{V}_{[T]}, \boldsymbol{J}} \sqrt{\frac{(a_2 - a_1)^2}{2} \operatorname{KL} \left( Q^{\boldsymbol{S}, \boldsymbol{V}_{[T]}} \, \big\| \, P^{\boldsymbol{J}, \boldsymbol{S}_{\boldsymbol{J}}, \boldsymbol{V}_{[T]}} \right)}.$$

This proposition is used to obtain the generalization error bound for SGLD [25, Theorem 3.1] by further combing Lemma 1. As $\tilde{\mathcal{M}}_{[T]}$ can be arbitrarily picked, $P$ works as an "Auxiliary Line" and is called the **prior** distribution, while $Q$ is the real distribution of parameters called the **posterior** distribution. In Section 3.2.1, we will argue the difficulty of applying Proposition 1 to analyze the effect of noise structures.

# 3 Formulate the Problem: Proper Constraints and Optimization Target

In this part, we formulate the optimization problem, i.e., finding the optimal noise covariance of SGLD in terms of the information-theoretical generalization bound, by selecting the proper optimization constraint and optimization target. Specifically, in Section 3.1, we argue that the trace of the noise covariance is a proper constraint to ensure the same optimization error; in Section 3.2, we argue that existing generalization bounds are not proper candidates as the optimization target, and propose a new information-theoretical bound parallel to existing ones but easier to analyze.

## 3.1 Constraint on the Covariance to Control the Empirical Risk

We first derive the constraint of noise covariance $\boldsymbol{\Sigma}_t$ in Eq. (3) from the perspective of training performance, under which we optimize the generalization bounds in the rest of this paper.

Without any constraint on the noise covariance, optimizing generalization error is trivial but meaningless: a direct combination of Theorem 1 of [37] and Theorem 1 of [28] shows that for isotropic SGLD with $\boldsymbol{\Sigma}_t = \sigma_t \mathbb{I}$, the generalization error after $T$ step satisfies $\operatorname{Gerr} \leq \mathcal{O}((\sum_{t=1}^T \log(1 + 1/\sigma_t))^{1/2})$. Therefore, as $\sigma_t \to \infty$ for $t \in [T]$, we have $\operatorname{Gerr} \to 0$, but then the update of SGLD is dominated by the noise, leading to arbitrary bad empirical risk. Hence, we need constraints on the covariance in order to control the empirical risk when minimizing the generalization error. Specifically, the expected decrease of the empirical risk for one iteration can be bounded as follows.

**Lemma 2.** *Let empirical risk $\mathcal{R}_{\boldsymbol{S}}(\boldsymbol{W})$ be $\beta$-smooth w.r.t. $\boldsymbol{W}$. Let $\boldsymbol{W}_{[T]}$ be given by state-dependent SGLD (Eq. (3)). Then,*

$$\mathbb{E}_{t+1|t} \mathcal{R}_{\boldsymbol{S}}(\boldsymbol{W}_{t+1}) - \mathcal{R}_{\boldsymbol{S}}(\boldsymbol{W}_t) \leq - \left( 1 - \frac{\beta \eta_{t+1}}{2} \right) \eta_{t+1} \| \nabla \mathcal{R}_{\boldsymbol{S}}(\boldsymbol{W}_t) \|^2 + \frac{\beta}{2} \operatorname{tr} \left( \frac{\eta_{t+1}^2 (N - b_t)}{(N-1) b_t} \boldsymbol{\Sigma}_{\boldsymbol{S}, \boldsymbol{W}_t}^{sd} + \boldsymbol{\Sigma}_{t+1}(\boldsymbol{S}, \boldsymbol{W}_t) \right).$$

The proof can be obtained by a standard analysis in optimization, and we defer it to Appendix B. By Lemma 2, the noise covariance $\boldsymbol{\Sigma}_t$ affects the upper bound of the empirical risk by its *trace*. Therefore, it is reasonable to keep $\operatorname{tr}(\boldsymbol{\Sigma}_t(\boldsymbol{S}, \boldsymbol{W}_{t-1}))$ unchanged while seeking the optimal $\boldsymbol{\Sigma}_t(\boldsymbol{S}, \boldsymbol{W}_{t-1})$ to minimize the generalization error. The constraint is given formally as follows:

**Constraint 1.** *The trace of $\boldsymbol{\Sigma}_t(\boldsymbol{S}, \boldsymbol{W}_{t-1})$ is fixed when optimizing the generalization error. That is, there exist positive constants $c_t(\boldsymbol{S}, \boldsymbol{W}_{t-1})$ depending on $\boldsymbol{S}$ and $\boldsymbol{W}_{t-1}$, such that,*

$$\operatorname{tr}(\boldsymbol{\Sigma}_t(\boldsymbol{S}, \boldsymbol{W}_{t-1})) = c_t(\boldsymbol{S}, \boldsymbol{W}_{t-1}).$$

We do not put any constraint on the value of $c_t$ in our latter analyses. Therefore, it is also possible to manipulate $c_t$ in order to jointly optimize the empirical risk and the generalization bound, which

however, is beyond the scope of this paper and we defer it to future works. Similar constraint is also proposed by [40] from the standpoint of kinetic energy when analyzing the effect of noise structure on the escaping efficiency from saddle points.

We next verify this constraint empirically. We run SGLD with different covariances on a four-layer neural network for the Fashion-MNIST classification problem. Concretely, the noise covariances are chosen respectively as $\boldsymbol{\Sigma}_t^{(1)} = \boldsymbol{\Sigma}_{\boldsymbol{S},\boldsymbol{W}_{t-1}}^{sd}$, i.e., "EC-SGLD" ( Empirical Covariance SGLD) curve in Fig.2, and $\boldsymbol{\Sigma}_t^{(2)} = \frac{1}{d}\operatorname{tr}(\boldsymbol{\Sigma}_{\boldsymbol{S},\boldsymbol{W}_{t-1}}^{sd})\mathbb{I}$, i.e., the "Iso-SGLD (C)" curve in Fig.2. It is easy to verify that $\operatorname{tr}(\boldsymbol{\Sigma}_t^{(1)}) = \operatorname{tr}(\boldsymbol{\Sigma}_t^{(2)})$, which is exactly the Constraint 1. We can see from Fig.2 that the convergence curves corresponding to $\boldsymbol{\Sigma}_t^{(1)}$ and $\boldsymbol{\Sigma}_t^{(2)}$ almost coincide with each other, validating the Constraint 1.

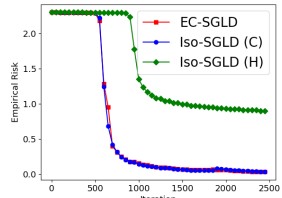

In comparison, another quantity $\operatorname{tr}(\mathcal{H}_{\boldsymbol{S},\boldsymbol{W}_{t-1}}\boldsymbol{\Sigma}_t)$, the trace of the empirical risk's Hessian times the empirical covariance, has also been proposed to govern the convergence behavior [34, Theorem 4.1]. We plot the convergence curve of SGLD with noise covariance $\boldsymbol{\Sigma}_t^{(3)} = \frac{\operatorname{tr}(\mathcal{H}_{\boldsymbol{S},\boldsymbol{W}_{t-1}}\boldsymbol{\Sigma}_{\boldsymbol{S},\boldsymbol{W}_{t-1}}^{sd})}{\operatorname{tr}(\mathcal{H}_{\boldsymbol{S},\boldsymbol{W}_{t-1}})}\mathbb{I}$ ("Iso-SGLD (H)" in Fig.2). While $\operatorname{tr}(\mathcal{H}_{\boldsymbol{S},\boldsymbol{W}_{t-1}}\boldsymbol{\Sigma}_t^{(1)}) = \operatorname{tr}(\mathcal{H}_{\boldsymbol{S},\boldsymbol{W}_{t-1}}\boldsymbol{\Sigma}_t^{(3)})$, there is a significant gap between curves of "EC-SGLD " and "Iso-SGLD (H)" in Fig.2. This implies that $\operatorname{tr}(\mathcal{H}_{\boldsymbol{S},\boldsymbol{W}_{t-1}}\boldsymbol{\Sigma}_t)$ is not a good constraint for the empirical risk of SGLD, validating the Constraint 1 from another side.

Figure 2: Training errors of SGLD with different noise covariances. The experiment is run on the Fashion-MNIST dataset with a four-layer neural network (see Appendix F).

In the rest of this paper, we will optimize the information-theoretical bound under Constraint 1 to ensure low empirical risk.

### 3.2 New Information-theoretical Bounds as the Optimization Target

We first demonstrate that existing information-theoretical bounds are not suitable for the optimization target in Section 3.2.1. Then we propose a new information-theoretical bound as the optimization target in Section 3.2.2.

#### 3.2.1 Difficulties When Applying Traditional Information-theoretical Bounds

By Lemma 1, the generalization error bound in Proposition 1 can be rewritten as

$$\mathbb{E}_{\boldsymbol{S},\boldsymbol{V}_{[T]},\boldsymbol{J}}\sqrt{\frac{(a_2-a_1)^2}{2}\sum_{s=1}^{T}\mathbb{E}_{Q_{s-1}^{\boldsymbol{S},\boldsymbol{V}_{[s-1]}}}\operatorname{KL}\left(Q_{s|(s-1)}^{\boldsymbol{S},\boldsymbol{V}_s}\left\|P_{s|(s-1)}^{\boldsymbol{J},\boldsymbol{S}_{\boldsymbol{J}},\boldsymbol{V}_s}\right.\right)}. \tag{4}$$

For **Problem** 1, there are two difficulties. For a fixed $Q^{\boldsymbol{S},\boldsymbol{V}_{[T]}}$, in order to obtain the optimal $P_{s|(s-1)}^{\boldsymbol{J},\boldsymbol{S}_{\boldsymbol{J}},\boldsymbol{V}_s}$ for any $s \in [T]$, one would first calculate the outside expectation of Eq. (4) over $\boldsymbol{S}_{\boldsymbol{J}^c}$, as $P^{\boldsymbol{S}_{\boldsymbol{J}},\boldsymbol{V}_{[T]}}$ is independent of $\boldsymbol{S}_{\boldsymbol{J}^c}$. However, for each term $\mathbb{E}_{Q_{s-1}^{\boldsymbol{S},\boldsymbol{V}_{[s-1]}}}\operatorname{KL}(Q_{s|(s-1)}^{\boldsymbol{S},\boldsymbol{V}_s}\|P_{s|(s-1)}^{\boldsymbol{J},\boldsymbol{S}_{\boldsymbol{J}},\boldsymbol{V}_s})$ where $s \in [T]$, both the probability measure $Q_{s-1}^{\boldsymbol{S},\boldsymbol{V}_{[s-1]}}$ and the function $\operatorname{KL}(Q_{s|(s-1)}^{\boldsymbol{S},\boldsymbol{V}_s}\|P_{s|(s-1)}^{\boldsymbol{J},\boldsymbol{S}_{\boldsymbol{J}},\boldsymbol{V}_s})$ has dependency on $\boldsymbol{S}_{\boldsymbol{J}^c}$, which makes evaluating the generalization bound with respect to $P^{\boldsymbol{S}_{\boldsymbol{J}},\boldsymbol{V}_{[T]}}$ extremely hard. Furthermore, when we come to optimize the bound w.r.t. $\boldsymbol{\Sigma}_{[T]}$, for each $\boldsymbol{\Sigma}_i$ where $i \in [T-1]$, the Eq. (4) depends on $\boldsymbol{\Sigma}_i$ via two terms $A$ and $B$, where

$$A = \mathbb{E}_{Q_{i-1}^{\boldsymbol{S},\boldsymbol{V}_{[T]}}}\operatorname{KL}\left(Q_{i|(i-1)}^{\boldsymbol{S},\boldsymbol{V}_{[T]}}\|P_{i|(i-1)}^{\boldsymbol{J},\boldsymbol{S}_{\boldsymbol{J}},\boldsymbol{V}_{[T]}}\right), \quad B = \sum_{s=i+1}^{T}\mathbb{E}_{Q_{s-1}^{\boldsymbol{S},\boldsymbol{V}_{[s-1]}}}\operatorname{KL}\left(Q_{s|(s-1)}^{\boldsymbol{S},\boldsymbol{V}_s}\|P_{s|(s-1)}^{\boldsymbol{J},\boldsymbol{S}_{\boldsymbol{J}},\boldsymbol{V}_s}\right).$$

While the first term is easy to deal with as $Q_{i-1}^{\boldsymbol{S},\boldsymbol{V}_{[T]}}$ is irrelevant of $\boldsymbol{\Sigma}_i$, the second term depends on $\boldsymbol{\Sigma}_i$ via the distribution $Q_{s-1}^{\boldsymbol{S},\boldsymbol{V}_{[s-1]}}$ for $s \geq i+1$, and can be very complex (please refer to Appendix C.1).

#### 3.2.2 Information-theoretical Generalization Bound for Identifying the Noise Effect

We then establish a new information-theoretical generalization bound as the optimization target that are parallel to the results in [25] but more suitable for analyzing the effect of noise structure. The

basic idea is to reverse the order of prior and posterior in the KL divergence to let $\mathbf{\Sigma}_t$ only affect one term in the generalization bound for each $t$ when finding the optimal point. The formal theorem is stated as follows:

**Theorem 1.** *Let sample set $\mathbf{S}$, mini-batch $\mathbf{V}_{[T]}$, random subset $\mathbf{J}$, the posterior distribution $Q$ output by Algorithm $\mathcal{A}$ with update rule $\mathcal{M}$ and the prior distribution $P$ output by Algorithm $\mathcal{B}$ with update rule $\tilde{\mathcal{M}}$ be defined as Proposition 1. Let $[a_1, a_2]$ be the range of loss. Then, the generalization error of $\mathcal{A}$ can be bounded as*

$$\text{Gerr} \leq \mathbb{E}_{\mathbf{S}, \mathbf{V}_{[T]}, \mathbf{J}} \sqrt{\frac{(a_2 - a_1)^2}{2} \text{KL}\left(P^{\mathbf{J}, \mathbf{S}_{\mathbf{J}}, \mathbf{V}_{[T]}} \| Q^{\mathbf{S}, \mathbf{V}_{[T]}}\right)}. \tag{5}$$

Compared to Proposition 1, Theorem 1 can be viewed as a parallel version with the positions of the prior distribution $P$ and the posterior distribution $Q$ reversed. This reverse benefits the optimization of the generalization error bound with respect to the noise covariance under Constraint 1 in two ways. First, by Lemma 1, the KL term in Eq. (5) can be further decomposed into

$$\text{KL}\left(P^{\mathbf{S}_{\mathbf{J}}, \mathbf{V}_{[T]}} \| Q^{\mathbf{S}, \mathbf{V}_{[T]}}\right) = \sum_{s=1}^{T} \mathbb{E}_{P_{1:s-1}^{\mathbf{J}, \mathbf{S}_{\mathbf{J}}, \mathbf{V}_{[s-1]}}} \text{KL}\left(P_{s|(s-1)}^{\mathbf{J}, \mathbf{S}_{\mathbf{J}}, \mathbf{V}_s} \| Q_{s|(s-1)}^{\mathbf{S}, \mathbf{V}_s}\right),$$

in which only each KL term depends on each $\mathbf{\Sigma}_s$, respectively. Secondly, for each summand in the above decomposition, only $Q_{s|(s-1)}^{\mathbf{S}, \mathbf{V}_s}$ depends on $\mathbf{S}_{\mathbf{J}^c}$, making it easy to compute the optimal prior in a "greedy" sense (see details in Section 4).

Hence in the rest of this paper, we study the optimal structure of $\mathbf{\Sigma}_t$ in terms of the generalization bound in Theorem 1 under the Constraint 1. Specifically, our ultimate goal can be stated as follows:

**Problem 1.** *What is the optimal structure of the noise covariance $\mathbf{\Sigma}_{[T]}$ for SGLD in terms of the generalization bound in Theorem 1 under the Constraint 1?*

In the rest of this paper, we focus on solving Problem 1 and its variants.

## 4   Obtain the Optimal Noise Covariance with the Greedy Prior

The generalization bound in Theorem 1 depends on both the prior distribution $P$ and the posterior distribution $Q$ (consequently on $\mathbf{\Sigma}_{[T]}$). Therefore it requires searching $P$ and $\mathbf{\Sigma}_{[T]}$ jointly to optimize the generalization error.

In this section, we solve **Problem 1** with greedily selected priors. Due to that the square root in the bound Eq. (5) make the dependency on $\mathbf{\Sigma}_{[T]}$ even more complex, we optimize a slightly different version of the bound by taking the expectation with respect to $\mathbf{S}_{\mathbf{J}^c}$ into the square root, i.e.,

$$\text{Gen}_T \triangleq \mathbb{E}_{\mathbf{S}_{\mathbf{J}}, \mathbf{V}_{[T]}, \mathbf{J}} \sqrt{\frac{(a_2 - a_1)^2}{2} \mathbb{E}_{\mathbf{S}_{\mathbf{J}^c}} \text{KL}\left(P^{\mathbf{J}, \mathbf{S}_{\mathbf{J}}, \mathbf{V}_{[T]}} \| Q^{\mathbf{S}, \mathbf{V}_{[T]}}\right)}. \tag{6}$$

By Jensen's Inequality, Eq. (6) is still a generalization bound, but allows the expectation with respect to $\mathbf{S}_{\mathbf{J}^c}$ to interact with the KL divergence directly. One can easily observe that $\text{Gen}_T$ is a mapping from $P$ and $Q$ (and thus from $P$ and $\mathbf{\Sigma}_{[T]}$) to a positive real. Therefore, we can reformulate Problem 1 mathematically as the following optimization problem **(P1)**:

$$\textbf{(P1)}. \min_{P, \mathbf{\Sigma}_{[T]}} \text{Gen}_T(P, \mathbf{\Sigma}_{[T]}), \text{ subject to: Constraint 1.}$$

For simplicity, we restrict that the considered noise covariance $\mathbf{\Sigma}_{[T]}$ only depends on the parameter $\mathbf{W}$, and is independent of the sample $\mathbf{S}$. By Lemma 1, the KL term in the optimization target $\text{Gen}_T$ can be decomposed into

$$\text{KL}\left(P^{\mathbf{J}, \mathbf{S}_{\mathbf{J}}, \mathbf{V}_{[T]}} \| Q^{\mathbf{S}, \mathbf{V}_{[T]}}\right) = \sum_{s=1}^{T} \mathbb{E}_{P_{s-1}^{\mathbf{J}, \mathbf{S}_{\mathbf{J}}, \mathbf{V}_{[s-1]}}} \text{KL}\left(P_{s|(s-1)}^{\mathbf{J}, \mathbf{S}_{\mathbf{J}}, \mathbf{V}_s} \| Q_{s|(s-1)}^{\mathbf{S}, \mathbf{V}_s}\right). \tag{7}$$

Therefore, when optimizing $\text{Gen}_T$ with respect to prior $P$, both $\text{KL}(P_{s|(s-1)}^{\mathbf{J}, \mathbf{S}_{\mathbf{J}}, \mathbf{V}_s} \| Q_{s|(s-1)}^{\mathbf{S}, \mathbf{V}_s})$ and $P_{i-1}^{\mathbf{J}, \mathbf{S}_{\mathbf{J}}, \mathbf{V}_{[i-1]}}$ $(i > s)$ have dependence on $P_{s|(s-1)}^{\mathbf{J}, \mathbf{S}_{\mathbf{J}}, \mathbf{V}_s}$. Similar to the discussion in Section 3.2.1, the dependence of $P_{i-1}^{\mathbf{J}, \mathbf{S}_{\mathbf{J}}, \mathbf{V}_{[i-1]}}$ with $i > s$ on $P_{s|(s-1)}^{\mathbf{J}, \mathbf{S}_{\mathbf{J}}, \mathbf{V}_s}$ can be very complex. Therefore, we approximate the optimal $P_{s|(s-1)}^{\mathbf{J}, \mathbf{S}_{\mathbf{J}}, \mathbf{V}_s}$ by the greedy prior which is defined as follows:

**Definition 1** (Greedy Prior). *We say $P^*$ is the optimal prior in the greedy sense, or the greedy prior for brevity, if for any $1 \leq s \leq T$ and any $\boldsymbol{S_J}$ and $\boldsymbol{V}_{[T]}$, $P^{*\boldsymbol{J},\boldsymbol{S_J},\boldsymbol{V}_s}_{s|(s-1)} = P^{s\boldsymbol{J},\boldsymbol{S_J},\boldsymbol{V}_s}_{s|(s-1)}$, where $P^s$ is defined as follows:*

$$P^s \triangleq \arg\min_P \left( \min_{\boldsymbol{\Sigma}_{[s]}} \mathrm{Gen}_s(P, \boldsymbol{\Sigma}_{[s]}) \right), \ \textit{subject to: Constraint 1.}$$

Intuitively, the conditional probability of $P^*$ of the step $s$ is the optimal one if we only consider the generalization bound for steps up to $s$, and a special case is that the step $T$ conditional probability of $P^*$ agrees with the step $T$ conditional probability of $P^T$, which is the desired optimal prior. This is why we call $P^*$ "greedy" and use it to approximate the optimal prior.

With the greedy prior, we characterize the optimal noise covariance by the following theorem.

**Theorem 2.** *Let the iteration of SGLD with state-dependent noise $Q^{\boldsymbol{S},\boldsymbol{V}_{[T]}}$ be given as Eq.(3). Under Constraint 1, the greedy optimal prior of step $t$ is given by*

$$\boldsymbol{W}_t = \boldsymbol{W}_{t-1} - \eta_t \left( \frac{|\boldsymbol{V}_t \cap \boldsymbol{J}|}{|\boldsymbol{V}_t|} \nabla \mathcal{R}_{\boldsymbol{S}_{\boldsymbol{V}_t \cap \boldsymbol{J}}}(\boldsymbol{W}_{t-1}) + \frac{|\boldsymbol{V}_t \cap \boldsymbol{J}^c|}{|\boldsymbol{V}_t|} \nabla \mathcal{R}_{\mathcal{D}}(\boldsymbol{W}_{t-1}) \right) + \mathcal{N}\left(\boldsymbol{0}, \boldsymbol{\Sigma}_t^*(\boldsymbol{W}_{t-1})\right),$$

*while the optimal covariance of noise $\boldsymbol{\Sigma}_{[T]}^*$ for $\mathrm{Gen}_T$ with the greedy prior is given by $\boldsymbol{\Sigma}_t^*(\boldsymbol{W}) = \lambda_t(\boldsymbol{W})(\boldsymbol{\Sigma}_{\boldsymbol{W}}^{pop})^{\frac{1}{2}}$ ($\forall t \in [T]$), where $\lambda_t(\boldsymbol{W}) = c_t(\boldsymbol{W})/\mathrm{tr}((\boldsymbol{\Sigma}_{\boldsymbol{W}}^{pop})^{\frac{1}{2}})$.*

As the sample size is large enough, we have $\boldsymbol{\Sigma}_{\boldsymbol{S},\boldsymbol{W}}^{sd} \to \boldsymbol{\Sigma}_{\boldsymbol{W}}^{pop}$ almost surely, which demonstrates the similarity between the solution of **Problem 1** and the noise covariance of SGD. Also, by the Law of Large Numbers, $\nabla \mathcal{R}_{\boldsymbol{S_J}} \approx \nabla \mathcal{R}_{\mathcal{D}}$, and the mean of the greedy optimal prior recovers the mean of the prior used in [25] (one can also refer to Eq. (10) in this paper for the form).

We briefly state the proof skeleton of Theorem 2, with the proof details deferred to Appendix D. To obtain the final optimal noise covariance in Theorem 2, we need to first derive the greedy prior, i.e., the optimal conditional distribution $P_{s|(s-1)}$ of the prior of step $s$ in terms of the generalization bound $\mathrm{Gen}_s$, which has the form

$$\mathbb{E}_{\boldsymbol{S_J},\boldsymbol{V}_{[s]},\boldsymbol{J}} \sqrt{\frac{(a_2-a_1)^2}{2} \mathbb{E}_{\boldsymbol{S}_{\boldsymbol{J}^c}} \sum_{t=1}^{s} \mathbb{E}_{P_{t-1}^{\boldsymbol{J},\boldsymbol{S_J},\boldsymbol{V}_{[t-1]}}} \mathrm{KL}\left(P^{\boldsymbol{J},\boldsymbol{S_J},\boldsymbol{V}_t}_{t|(t-1)} \left\| Q^{\boldsymbol{S},\boldsymbol{V}_t}_{t|(t-1)} \right.\right)}. \tag{8}$$

Typically, solving the optimal $P_{s|(s-1)}$ requires optimizing $\mathrm{Gen}_s$ with respect to all $P_{t|(t-1)}$ $t \in [s]$ and $\boldsymbol{\Sigma}_{[s]}$, which is still very complex. However, we can tackle this problem in a rather elegant way. We first investigate the optimal noise covariance in terms of a single KL divergence term in $\mathrm{Gen}_s$.

**Lemma 3.** *Under Constraint 1, the optimal noise covariance of the following problem*

$$\min_{\boldsymbol{\Sigma}_s} \left( \min_{P^{\boldsymbol{J},\boldsymbol{S_J},\boldsymbol{V}_s}_{s|(s-1)}} \mathbb{E}_{\boldsymbol{S}_{\boldsymbol{J}^c} \sim \mathcal{D}} \mathrm{KL}\left(P^{\boldsymbol{J},\boldsymbol{S_J},\boldsymbol{V}_s}_{s|(s-1)} \left\| Q^{\boldsymbol{S},\boldsymbol{V}_s}_{s|(s-1)} \right.\right) \right). \tag{9}$$

*is attained at $\boldsymbol{\Sigma}_t^*(\boldsymbol{W})$, where $\boldsymbol{\Sigma}_t^*(\boldsymbol{W})$ is defined as Theorem 2.*

By Lemma 3, the optimal solution of Eq. (9) doesn't rely on $\boldsymbol{J}, \boldsymbol{S_J}$, or $\boldsymbol{V}_{[s]}$. On the other hand, by Eq. (8), $\mathbb{E}_{\boldsymbol{S}_{\boldsymbol{J}^c} \sim \mathcal{D}} \mathrm{KL}\left(P^{\boldsymbol{J},\boldsymbol{S_J},\boldsymbol{V}_s}_{s|(s-1)} \left\| Q^{\boldsymbol{S},\boldsymbol{V}_s}_{s|(s-1)} \right.\right)$ ($\forall \boldsymbol{J}, \boldsymbol{S_J}, \boldsymbol{V}_{[s]}$) are the only terms depending on $P^{\boldsymbol{J},\boldsymbol{S_J},\boldsymbol{V}_s}_{s|(s-1)}$ and $\boldsymbol{\Sigma}_s$. Therefore, the optimal $\boldsymbol{\Sigma}_s$ is also $\lambda_t(\boldsymbol{W})(\boldsymbol{\Sigma}_{\boldsymbol{W}}^{pop})^{\frac{1}{2}}$, which is formally stated as the following lemma.

**Lemma 4.** *The optimal $\boldsymbol{\Sigma}_s$ and $P_{s|(s-1)}$ in terms of $\mathrm{Gen}_s$ are the same as $\boldsymbol{\Sigma}_s^*$ and $P^*_{s|(s-1)}$ given by Theorem 2, respectively.*

With the greedy prior derived, we apply it back to the generalization bound $\mathrm{Gen}_T$. As $\mathrm{Gen}_T$ depends on $\boldsymbol{\Sigma}_s$ also through $\mathbb{E}_{\boldsymbol{S}_{\boldsymbol{J}^c} \sim \mathcal{D}} \mathrm{KL}\left(P^{\boldsymbol{J},\boldsymbol{S_J},\boldsymbol{V}_s}_{s|(s-1)} \left\| Q^{\boldsymbol{S},\boldsymbol{V}_s}_{s|(s-1)} \right.\right)$, by applying Lemma 3 again, we derive Theorem 2.

# 5   Extension: Optimal Noise Covariance with Fixed Priors

In existing works [25, 12], the prior distribution is set to be the SGLD with isotropic noise, which (with the notations in Theorem 1) is given by

$$\tilde{\mathcal{M}}_t(\boldsymbol{W}_{t-1}, \boldsymbol{S_J}, \boldsymbol{V}_t, \boldsymbol{J})$$
$$= \boldsymbol{W}_{t-1} - \eta_t \left( \frac{|\boldsymbol{V}_t \cap \boldsymbol{J}|}{|\boldsymbol{V}_t|} \nabla \mathcal{R}_{\boldsymbol{S}_{\boldsymbol{V}_t \cap \boldsymbol{J}}}(\boldsymbol{W}_{t-1}) + \frac{|\boldsymbol{V}_t \cap \boldsymbol{J}^c|}{|\boldsymbol{V}_t|} \nabla \mathcal{R}_{\boldsymbol{S_J}}(\boldsymbol{W}_{t-1}) \right) + \mathcal{N}(\boldsymbol{0}, \sigma_t \mathbb{I}_d), \quad (10)$$

where $\sigma_t > 0$ is the noise scale of prior noise covariance. In our latter analysis, we generalize the prior by allowing $\sigma_t$ depend on $\boldsymbol{W}_{t-1}$. The formal description of the iteration of the prior is deferred to Appendix E.1 for completeness.

Therefore, it is also interesting to see what the optimal noise covariance looks like if the prior is fixed as the one commonly adopted in the existing analyses, e.g., Eq. (10). We still set the optimization constraint the same as Section 4, but change the optimization target a little to

$$\widetilde{\mathrm{Gen}}_T \triangleq \mathbb{E}_{\boldsymbol{S}} \sqrt{\frac{(a_2 - a_1)^2}{2} \mathbb{E}_{\boldsymbol{V}_{[T]}, \boldsymbol{J}} \mathrm{KL}\left(P^{\boldsymbol{J}, \boldsymbol{S_J}, \boldsymbol{V}_{[T]}} \big\| Q^{\boldsymbol{S}, \boldsymbol{V}_{[T]}}\right)}.$$

By Jensen's Inequality and Theorem 1, $\widetilde{\mathrm{Gen}}_T$ is still a generalization bound, but allows us to treat the expectation of the KL divergence with respect to $\boldsymbol{V}_{[T]}$ and $\boldsymbol{J}$ for given $\boldsymbol{S}$ as a whole in optimization. Similar trick is also adopted in [25] to obtain the final generalization error of SGLD. The problem can be mathematically formulated as the following optimization problem **(P2)**:

$$\textbf{(P2)} \quad \min_{\boldsymbol{\Sigma}_{[T]}} \widetilde{\mathrm{Gen}}_T(P, \boldsymbol{\Sigma}_{[T]}), \text{ subject to: Constraint 1,}$$

where $P$ is given by the update rule Eq. (10). As $\boldsymbol{S_J}$ is obtained by removing only one sample from the dataset $\boldsymbol{S}$ and the size of $\boldsymbol{S}$ is usually large in practice, it is reasonable to make the assumption that for any fixed $\boldsymbol{V}_{[T]}$ and $\boldsymbol{S}$, the prior distribution is the same regardless of $\boldsymbol{J}$.

**Assumption 1.** *For any fixed dataset $\boldsymbol{S}$ and mini-batches $\boldsymbol{V}_{[T]}$, the distribution $P^{\boldsymbol{J}, \boldsymbol{S_J}, \boldsymbol{V}_{[T]}}$ is invariant of $\boldsymbol{J}$.*

We also restrict that $c_t(\boldsymbol{S}, \boldsymbol{W}) \leq d\sigma_t$ in order to guarantee the noise scale of the prior comparable to that of the posterior. The optimal noise covariance with prior fixed can then be characterized by the following theorem.

**Theorem 3.** *Let prior and posterior be defined as Eq.(10) and Eq.(3), respectively. Let Assumption 1 hold. Then, the solution of (P2) is given by*

$$\boldsymbol{\Sigma}_t^*(\boldsymbol{S}, \boldsymbol{W}) = \boldsymbol{O}_{\boldsymbol{S}, \boldsymbol{W}}^{sd} \mathrm{Diag}(\tilde{\omega}_1^{\boldsymbol{S}, \boldsymbol{W}}, \cdots, \tilde{\omega}_d^{\boldsymbol{S}, \boldsymbol{W}}) \left(\boldsymbol{O}_{\boldsymbol{S}, \boldsymbol{W}}^{sd}\right)^{\top},$$

*where $\tilde{\omega}_i^{\boldsymbol{S}, \boldsymbol{W}} \geq 0$ ($i \in [d]$) (the exact form is omitted here) and $\left(\boldsymbol{O}_{\boldsymbol{S}, \boldsymbol{W}}^{sd}\right)$ is the orthogonal matrix that diagonalizes $\boldsymbol{\Sigma}_{\boldsymbol{S}, \boldsymbol{W}}^{sd}$ as $\boldsymbol{\Sigma}_{\boldsymbol{S}, \boldsymbol{W}}^{sd} = \boldsymbol{O}_{\boldsymbol{S}, \boldsymbol{W}}^{sd} \mathrm{Diag}(\omega_1^{\boldsymbol{S}, \boldsymbol{W}}, \cdots, \omega_d^{\boldsymbol{S}, \boldsymbol{W}}) \left(\boldsymbol{O}_{\boldsymbol{S}, \boldsymbol{W}}^{sd}\right)^{\top}$. Moreover, for any $i \neq j$, $\tilde{\omega}_i^{\boldsymbol{S}, \boldsymbol{W}} \geq \tilde{\omega}_j^{\boldsymbol{S}, \boldsymbol{W}}$ if and only if $\omega_i^{\boldsymbol{S}, \boldsymbol{W}} \geq \omega_j^{\boldsymbol{S}, \boldsymbol{W}}$.*

The proof together with the the exact formula of $\tilde{\omega}_i^{\boldsymbol{S}, \boldsymbol{W}}$ is deferred to Appendix 5. From Theorem 3, the optimal point $(\boldsymbol{\Sigma}_t^*)_{t=1}^T$ is similar to the empirical gradient covariance matrix $\boldsymbol{\Sigma}^{sd}$ in two ways: 1) $\{\boldsymbol{\Sigma}_t^*\}_{t=1}^T$ share the same eigenvectors with $\boldsymbol{\Sigma}^{sd}$; 2) the corresponding eigenvalues of $\boldsymbol{\Sigma}_t^*$ has the same order as $\boldsymbol{\Sigma}^{sd}$. Though the value of $\tilde{\omega}_i^{S,W}$ is not comparable to $\omega_i^{S,W}$ because $\boldsymbol{\Sigma}_t^*$ is affected by the prior noise and the posterior noise covariance, which are freely chosen, it can be shown that the condition number of $\boldsymbol{\Sigma}_t^*$ is smaller than $\boldsymbol{\Sigma}_{S,W}^{sd}$ (please refer to Appendix E.4 for details).

Theorem 3 also reveals an interesting correlation between the noise covariance matrices of the prior and posterior. While the optimal noise covariance is affected by the prior noise covariance, it also depends on the distance between the means of the prior and the posterior (see Lemma 13 in Appendix E.2), which brings the information of empirical gradient covariance into the optimal noise structure. That being said, the optimal posterior noise covariance is biased to the empirical gradient covariance from prior covariance. We note that such analysis can be easily extended to arbitrary priors.

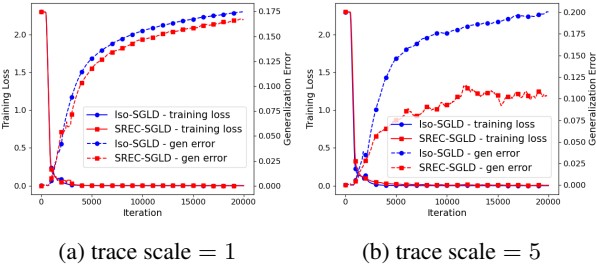

(a) trace scale = 1        (b) trace scale = 5

Figure 3: The training loss and and the generalization error (test loss − training loss) of the Iso-SGLD and SREC-SGLD. Traces of the noise covariance in (a) and (b) are 1 and 5 times of $\mathrm{tr}((\mathbf{\Sigma}^{sd}_{S,\mathbf{W}})^{1/2})$, respectively.

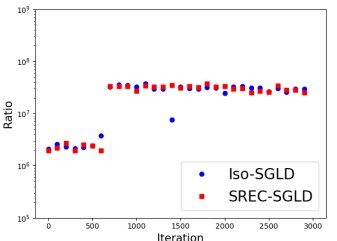

Figure 4: Ratio of the 1st to the 500th largest eigenvalue of the empirical gradient covariance for two SGLDs.

## 6 Empirical Verification

In this section, we support our theoretical findings with some experiments. We adopt the setting in [40] where a four-layer neural network with 11330 parameters is used to conduct the classification task on the Fashion-MNIST except that we use 10000 training samples instead of 1200 used in [40]. We defer detailed settings of the experiments to Appendix F.

We first verify if the empirical gradient covariance is far from isotropic Gaussian distribution. We plot the ratios of the 1st eigenvalue to the 500th largest eigenvalue of the empirical gradient covariance along the training trajectory of both SGD and Iso-SGLD in Figure 4. We can see that throughout the training procedure, the ratios of empirical gradient covariance stay around $10^7$ for both SGD and Iso-SGLD. In constrast, for the isotropic Gaussian, the distribution of eigenvalues follows semi-circle law, and the ratio would be around 1.1 for dimension 11330. This demonstrates that the energy of empirical gradient covariance concentrates in a very small subspace, less than 5% of the total 11330 dimensions. Hence the empirical gradient covariance is highly anisotropic.

We next verify our main claim Theorem 2 that under Constraint 1, the generalization error for SGLD with the optimal noise covariance is much smaller than that for Iso-SGLD. Specifically, we consider the SGLD with noise covariance equal to the (scaled) square root of the empirical gradient covariance, coined "SREC-SGLD". We compare the training loss and the generalization error of SREC-SGLD and Iso-SGLD under Constraint 1 on the noise covariance trace. From Figure 3, we can see that the generalization error of SREC-SGLD is smaller than Iso-SGLD for different noise trace scales, which supports Theorem 2. Moreover, if we look at the training loss curves, the SREC-SGLD and Iso-SGLD behave almost the same for the same noise trace scale, which supports Lemma 2.

Finally though our analysis does not cover the generalization bound for SGD, we further empirically show how SREC-SGLD performs in comparison with SGD and the Iso-SGLD in Figure 1. We can see that with the same noise trace scale, SREC indeed provides a more accurate characterization of SGD compared to Iso-SGLD.

## 7 Conclusion and Future Direction

In this paper, we study the optimal noise covariance of SGLD in terms of its generalization ability. Specifically, we first formulate the optimization problem both by deriving constraints from the optimization performance and proposing the optimization target by constructing a new information theoretical bound. We then solve the problem with both greedy optimal prior and fixed prior. Interestingly, we observe that the optimal noise covariance aligns with the empirical gradient covariance, which indicates the superiority of the noise covariance of SGD in terms of generalization.

## Acknowledgement

The authors would like to thank Mr. Ziming Liu for helpful theoretical discussions.

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
