# Supplementary materials for "Optimizing Information-theoretical Generalization Bound via Anisotropic Noise in SGLD"

The supplementary materials are organized as follows. In Appendix A, we provide some basic lemmas which are used throughout the proofs in the rest of the materials. In Appendix B, we provide the proof of Lemma 2. In Appendix C, D, and E, we provide the detailed proofs of Lemmas and Theorems respectively in Section 3.2, Section 4 , and Section 5. In Appendix F, we provide the detailed settings of the experiments in the main text together with an additional experiments to justify the result of Theorem 2.

## A  Preliminaries

In this section, we provide some basic lemmas that will be used throughout the proof both from probability theory and from matrix analysis.

### A.1  Preparations in Probability Theory

The first lemma is a standard result characterizing the KL divergence between two Gaussian distributions.

**Lemma 5** (KL divergence between Gaussian distributions). *Let* $\mathrm{P}_1$ *and* $\mathrm{P}_2$ *are multivariate Gaussian distributions on* $\mathbb{R}^d$ *with mean and covariance respectively* $\mu_1$, $\mathbf{\Sigma}_1$ *and* $\mu_2$, $\mathbf{\Sigma}_2$*. Then the KL divergence between* $\mathrm{P}_1$ *and* $\mathrm{P}_2$ *are given as follows:*

$$\mathrm{KL}(\mathrm{P}_1||\mathrm{P}_2) = \frac{1}{2}\left(\mathrm{tr}\left(\mathbf{\Sigma}_2^{-1}\mathbf{\Sigma}_1\right) + (\mu_2 - \mu_1)^\top \mathbf{\Sigma}_2^{-1}(\mu_2 - \mu_1) - d + \ln\left(\frac{\det\mathbf{\Sigma}_2}{\det\mathbf{\Sigma}_1}\right)\right).$$

We then provide a lemma which gives the expected difference between two uniform sampling variables.

**Lemma 6** (Two step sampling). *Suppose* $\boldsymbol{z}$ *is a discrete random variable with* $\mathbb{P}(\boldsymbol{z} = \boldsymbol{z}_i) = \frac{1}{N}$, $\forall i = 1, 2, \cdots, N$, *where the support set is* $\mathcal{Z} = \{\boldsymbol{z}_1, \cdots, \boldsymbol{z}_N\} \subset \mathbb{R}^d$*. Suppose further* $\boldsymbol{U}$ *is a random index set with size* $b$ *and sampled uniformly without replacement from* $[N]$*. Suppose* $\boldsymbol{V}$ *is another random index set independent of* $\boldsymbol{U}$ *with size* $N-1$ *and sampled uniformly without replacement from* $[N]$*. Denote subset of* $\mathcal{Z}$ *with index in* $\boldsymbol{U} \cap \boldsymbol{V}^c$ *and* $\boldsymbol{V}$ *respectively as* $\mathcal{Z}_{\boldsymbol{U}\cap\boldsymbol{V}^c} = \{\boldsymbol{z}_i, i \in \boldsymbol{U} \cap \boldsymbol{V}^c\}$, $\mathcal{Z}_{\boldsymbol{V}} = \{\boldsymbol{z}_i, i \in \boldsymbol{V}\}$*, and the average of* $\mathcal{Z}_{\boldsymbol{U}\cap\boldsymbol{V}^c}$ *and* $\mathcal{Z}_{\boldsymbol{V}}$ *respectively as* $\bar{\mathcal{Z}}_{\boldsymbol{U}\cap\boldsymbol{V}^c}$ *and* $\bar{\mathcal{Z}}_{\boldsymbol{V}}$*. Then the following equation holds:*

$$\mathbb{E}_{\boldsymbol{U},\boldsymbol{V}}\left(\frac{(b - |\boldsymbol{U}\cap\boldsymbol{V}|)^2}{b^2}(\bar{\mathcal{Z}}_{\boldsymbol{V}} - \bar{\mathcal{Z}}_{\boldsymbol{U}\cap\boldsymbol{V}^c})(\bar{\mathcal{Z}}_{\boldsymbol{V}} - \bar{\mathcal{Z}}_{\boldsymbol{U}\cap\boldsymbol{V}^c})^\top\right) = \frac{1}{Nb}\left(\frac{N}{N-1}\right)^2 \mathrm{Cov}(\boldsymbol{z}).$$

*Proof.* We rewrite $\mathbb{E}_{\boldsymbol{U},\boldsymbol{V}}\left(\frac{(b-|\boldsymbol{U}\cap\boldsymbol{V}|)^2}{b^2}(\bar{\mathcal{Z}}_{\boldsymbol{V}} - \bar{\mathcal{Z}}_{\boldsymbol{U}\cap\boldsymbol{V}^c})(\bar{\mathcal{Z}}_{\boldsymbol{V}} - \bar{\mathcal{Z}}_{\boldsymbol{U}\cap\boldsymbol{V}^c})^\top\right)$ by taking conditional expectation with respect to $|\boldsymbol{U}\cap\boldsymbol{V}|$ as follows:

$$\mathbb{E}_{\boldsymbol{U},\boldsymbol{V}}\left(\frac{(b - |\boldsymbol{U}\cap\boldsymbol{V}|)^2}{b^2}(\bar{\mathcal{Z}}_{\boldsymbol{V}} - \bar{\mathcal{Z}}_{\boldsymbol{U}\cap\boldsymbol{V}^c})(\bar{\mathcal{Z}}_{\boldsymbol{V}} - \bar{\mathcal{Z}}_{\boldsymbol{U}\cap\boldsymbol{V}^c})^\top\right)$$

$$= \mathbb{E}_{|\boldsymbol{U}\cap\boldsymbol{V}^c|}\mathbb{E}_{\boldsymbol{U},\boldsymbol{V}}^{|\boldsymbol{U}\cap\boldsymbol{V}^c|}\left(\frac{|\boldsymbol{U}\cap\boldsymbol{V}^c|^2}{b^2}(\bar{\mathcal{Z}}_{\boldsymbol{V}} - \bar{\mathcal{Z}}_{\boldsymbol{U}\cap\boldsymbol{V}^c})(\bar{\mathcal{Z}}_{\boldsymbol{V}} - \bar{\mathcal{Z}}_{\boldsymbol{U}\cap\boldsymbol{V}^c})^\top\right)$$

$$= \mathbb{P}(|\boldsymbol{U}\cap\boldsymbol{V}^c| = 1)\mathbb{E}_{\boldsymbol{U},\boldsymbol{V}}^{|\boldsymbol{U}\cap\boldsymbol{V}^c|=1}\left(\frac{1}{b^2}(\bar{\mathcal{Z}}_{\boldsymbol{V}} - \bar{\mathcal{Z}}_{\boldsymbol{V}^c})(\bar{\mathcal{Z}}_{\boldsymbol{V}} - \bar{\mathcal{Z}}_{\boldsymbol{V}^c})^\top\right)$$

$$= \mathbb{P}(|\boldsymbol{U}\cap\boldsymbol{V}^c| = 1)\mathbb{E}_{\boldsymbol{V}}\left(\frac{1}{b^2}(\bar{\mathcal{Z}}_{\boldsymbol{V}} - \bar{\mathcal{Z}}_{\boldsymbol{V}^c})(\bar{\mathcal{Z}}_{\boldsymbol{V}} - \bar{\mathcal{Z}}_{\boldsymbol{V}^c})^\top\right)$$

$$= \frac{1}{Nb}\left(\frac{N}{N-1}\right)^2 \mathrm{Cov}(\boldsymbol{z}).$$

The proof is completed. $\qquad\square$

We provide the following lemma for computing the KL divergence between two joint distributions.

**Lemma 7.** *Let $\boldsymbol{X}$, $\boldsymbol{Y}$, and $\boldsymbol{Z}$ be three random variables with $\boldsymbol{X}$ and $\boldsymbol{Y}$ having the same support set. Then the KL divergence between the joint distribution of $(\boldsymbol{X}, \boldsymbol{Z})$ and $(\boldsymbol{Y}, \boldsymbol{Z})$ can be decomposed into*

$$\mathrm{KL}\left((\boldsymbol{X}, \boldsymbol{Z})\|(\boldsymbol{Y}, \boldsymbol{Z})\right) = \mathbb{E}_{\boldsymbol{Z}} \, \mathrm{KL}\left((\boldsymbol{X}|\boldsymbol{Z})\|(\boldsymbol{Y}|\boldsymbol{Z})\right).$$

*Proof.* By the definition of KL divergence,

$$
\begin{aligned}
\mathrm{KL}\left((\boldsymbol{X}, \boldsymbol{Z})\|(\boldsymbol{Y}, \boldsymbol{Z})\right) &= \int \mathbb{P}(\boldsymbol{X}, \boldsymbol{Z}) \log \frac{\mathbb{P}(\boldsymbol{X}, \boldsymbol{Z})}{\mathbb{P}(\boldsymbol{Y}, \boldsymbol{Z})} \\
&= \int \mathbb{P}(\boldsymbol{Z}) \mathbb{P}^{\boldsymbol{Z}}(\boldsymbol{X}) \log \frac{\mathbb{P}^{\boldsymbol{Z}}(\boldsymbol{X})\mathbb{P}(\boldsymbol{Z})}{\mathbb{P}^{\boldsymbol{Z}}(\boldsymbol{Y})\mathbb{P}(\boldsymbol{Z})} \\
&= \int \mathbb{P}(\boldsymbol{Z}) \int \mathbb{P}^{\boldsymbol{Z}}(\boldsymbol{X}) \log \frac{\mathbb{P}^{\boldsymbol{Z}}(\boldsymbol{X})}{\mathbb{P}^{\boldsymbol{Z}}(\boldsymbol{Y})} \\
&= \mathbb{E}_{\boldsymbol{Z}} \, \mathrm{KL}\left((\boldsymbol{X}|\boldsymbol{Z})\|(\boldsymbol{Y}|\boldsymbol{Z})\right).
\end{aligned}
$$

The proof is completed. $\qquad\square$

In the end of this section, we provide a proof of Lemma 1 using Lemma 7 for the completeness of this paper.

*Proof of Lemma 1.* By Lemma 7, we have

$$
\begin{aligned}
&\mathrm{KL}(Q_{0:T}\|P_{0:T}) \\
&= \mathrm{KL}(Q_{0:T}\|P_{0:T}) - \mathrm{KL}(Q_{0:T}\|(Q_{0:T-1}, P_{T|[T-1]})) + \mathrm{KL}(Q_{0:T}\|(Q_{0:T-1}, P_{T|[T-1]})) \\
&= \int Q_{0:T} \log \frac{Q_{0:T}}{P_{0:T}} - \int Q_{0:T} \log \frac{Q_{0:T}}{Q_{0:T-1}P_{T|[T-1]}} + \mathbb{E}_{Q_{T-1}} \, \mathrm{KL}\left(Q_{T|[T-1]}\|P_{T|[T-1]}\right) \\
&= \int Q_{0:T} \log \frac{Q_{0:T-1}}{P_{0:T-1}} + \mathbb{E}_{Q_{T-1}} \, \mathrm{KL}\left(Q_{T|[T-1]}\|P_{T|[T-1]}\right) \\
&= \int Q_{0:T-1} \log \frac{Q_{0:T-1}}{P_{0:T-1}} + \mathbb{E}_{Q_{T-1}} \, \mathrm{KL}\left(Q_{T|[T-1]}\|P_{T|[T-1]}\right) \\
&= \mathrm{KL}\left(Q_{0:T-1}\|P_{0:T-1}\right) + \mathbb{E}_{Q_{T-1}} \, \mathrm{KL}\left(Q_{T|[T-1]}\|P_{T|[T-1]}\right).
\end{aligned}
$$

The proof is then completed by induction. $\qquad\square$

**Remark 1.** *In this paper, we focus on the case where $Q_{0:T}$ and $P_{0:T}$ obeys the* Markov Property, *i.e., for any t,*

$$Q_{t|[t-1]} = Q_{t|(t-1)}, \; P_{t|[t-1]} = P_{t|(t-1)}.$$

*Therefore, the result in Lemma 1 becomes*

$$\mathrm{KL}(Q_{0:T}\|P_{0:T}) = \sum_{t=1}^{T} \mathbb{E}_{Q_{0:t-1}} \left[\mathrm{KL}\left(Q_{t|(t-1)}\|P_{t|(t-1)}\right)\right].$$

## A.2 Technical Lemmas in Matrix Analysis

We first provide a sufficient and necessary condition of that two symmetric matrices commute, and the proof can be found from any Linear Algebra Textbook (e.g. [32]).

**Lemma 8.** *Let $\boldsymbol{A}$ and $\boldsymbol{B}$ be two $d \times d$ real symmetric matrices. Then, $\boldsymbol{A}$ and $\boldsymbol{B}$ commute (i.e., $\boldsymbol{AB} = \boldsymbol{BA}$), if and only if there exists an orthogonal matrix $\boldsymbol{O}$ which can diagonalize $\boldsymbol{A}$ and $\boldsymbol{B}$ simultaneously, i.e., both $\boldsymbol{O}^\top \boldsymbol{A} \boldsymbol{O}$ and $\boldsymbol{O}^\top \boldsymbol{B} \boldsymbol{O}$ are diagonal.*

The next lemma is a key technique to obtain the optimal noise covariance of Theorem 2 and Theorem 3.

**Lemma 9.** *Let $\boldsymbol{B} \in \mathbb{R}^{d \times d}$ be a (fixed) positive definite matrix with eigenvalues $(\beta_1, \cdots, \beta_d)$, where $\beta_i \geq 0$. Let $\boldsymbol{G} \in \mathbb{R}^{d \times d}$ be a positive definite matrix variable with fixed trace $\operatorname{tr} \boldsymbol{G} = c$, where $c$ is a positive constant and $c \leq \operatorname{tr}(\boldsymbol{B})$. Then the minimum of $\operatorname{tr}(\boldsymbol{G}^{-1}\boldsymbol{B}) + \ln \det(\boldsymbol{G})$ is achieved at $\boldsymbol{G} = \boldsymbol{O}^\top \operatorname{Diag}(\alpha_1, \cdots, \alpha_d)\boldsymbol{O}$, where*

$$\alpha_i^* = \frac{\sqrt{1 - 4\lambda^* \beta_i} - 1}{-2\lambda^*},$$

*$\boldsymbol{O}$ is any orthogonal matrix which diagonalize $\boldsymbol{B}$ as*

$$\boldsymbol{B} = \boldsymbol{O}^\top \operatorname{Diag}(\beta_1, \cdots, \beta_d)\boldsymbol{O},$$

*and $\lambda^* \leq 0$ is the unique solution of*

$$\sum_{i=1}^{d} \frac{2\beta_i}{1 + \sqrt{1 - 4\lambda^* \beta_i}} = c. \tag{11}$$

**Remark 2.** *$f(\lambda) = \sum_{i=1}^{d} \frac{2\beta_i}{1+\sqrt{1-4\lambda\beta_i}}$ is a monotonously increasing function with respect to $\lambda$, which guarantees the uniqueness of the solution of $f(\lambda) = c$.*

Lemma 9 is proved via two steps: 1) we first prove for $\boldsymbol{G}$ with fixed eigenvalues, $\operatorname{tr} \boldsymbol{G}^{-1}\boldsymbol{B} + \ln \det(\boldsymbol{G})$ is optimized if and only if $\boldsymbol{G}$ and $\boldsymbol{B}$ share the same eigenvectors; 2) we then calculate the eigenvalues of the optimal $\boldsymbol{G}$ using the method of Lagrange multipliers. Theorem 3 can then be obtained by applying Lemma 9 and setting $\boldsymbol{G} = \boldsymbol{\Sigma}_t(\boldsymbol{S}, \boldsymbol{W})$ and $\boldsymbol{B} = \sigma_t \mathbb{I} + \frac{\eta_t^2}{Nb_t}\left(\frac{N}{N-1}\right)^2 \boldsymbol{\Sigma}_{\boldsymbol{S},\boldsymbol{W}}^{sd}$. We first prove the eigenvectors of $\boldsymbol{G}$ agree with those of $\boldsymbol{B}$.

**Lemma 10.** *Let $\boldsymbol{G} \in \mathbb{R}^{d \times d}$ be a positive definite matrix variable with fixed eigenvalues $(\alpha_i)_{i=1}^{d}$. Specifically, let $\alpha_1 \geq \alpha_2 \geq \cdots \geq \alpha_d > 0$ be all the eigenvalues of $\boldsymbol{G}$, and $\boldsymbol{G}$ can be any element from the following set*

$$\{\boldsymbol{Q}^\top \operatorname{Diag}(\alpha_1, \cdots, \alpha_d)\boldsymbol{Q} : \boldsymbol{Q} \text{ is orthogonal}\}.$$

*Let $\boldsymbol{B}$ be a fixed positive semi-definite matrix, with eigenvalues $(\beta_i)_{i=1}^{d}$ satisfies $\beta_1 \geq \beta_2 \cdots \geq \beta_d \geq 0$. Then, the optimal (minimal) value of $g(\boldsymbol{G}) = \operatorname{tr}(\boldsymbol{G}^{-1}\boldsymbol{B})$ is achieved when*

$$\boldsymbol{G}^* = \boldsymbol{O}^\top \operatorname{Diag}(\alpha_1, \cdots, \alpha_d)\boldsymbol{O},$$

*where $\boldsymbol{O}$ is any orthogonal matrix which diagonalizes $\boldsymbol{B}$ as*

$$\boldsymbol{B} = \boldsymbol{O}^\top \operatorname{Diag}(\beta_1, \cdots, \beta_d)\boldsymbol{O}$$

*and the optimal value of $g(\boldsymbol{G})$ is $\sum_{i=1}^{d} \frac{\beta_i}{\alpha_i}$.*

*Proof.* Let $\boldsymbol{G}^*$ be a optimal point of $\operatorname{tr}(\boldsymbol{G}^{-1}\boldsymbol{B})$. We will then obtain the condition of $\boldsymbol{G}^*$ by adding a disturbance. Specifically, let $\boldsymbol{A}$ be an anti-symmetric matrix. Then,

$$(\mathbb{I} - \varepsilon\boldsymbol{A})(\mathbb{I} - \varepsilon\boldsymbol{A})^T = \mathbb{I} + \varepsilon^2 \boldsymbol{A}\boldsymbol{A}^\top.$$

As $\varepsilon$ is small enough, $\mathbb{I} + \varepsilon^2 \boldsymbol{A}\boldsymbol{A}^\top$ is inevitable, and positive definite. Therefore, $(\mathbb{I} - \varepsilon\boldsymbol{A})(\mathbb{I} + \varepsilon^2 \boldsymbol{A}\boldsymbol{A}^\top)^{-\frac{1}{2}}$ is orthogonal. As $\varepsilon \to 0$,

$$\lim_{\varepsilon \to 0}(\mathbb{I} - \varepsilon\boldsymbol{A})(\mathbb{I} + \varepsilon^2 \boldsymbol{A}\boldsymbol{A}^\top)^{-\frac{1}{2}} = \mathbb{I},$$

and

$$(\mathbb{I} - \varepsilon\boldsymbol{A})(\mathbb{I} + \varepsilon^2 \boldsymbol{A}\boldsymbol{A}^\top)^{-\frac{1}{2}} - \mathbb{I} = -\varepsilon\boldsymbol{A} + \boldsymbol{o}(\varepsilon).$$

Since $\boldsymbol{G}^*$ is an optimal point of $\operatorname{tr}(\boldsymbol{G}^{-1}\boldsymbol{B})$, we have

$$\operatorname{tr}((\boldsymbol{G}^*)^{-1}\boldsymbol{B}) \leq \operatorname{tr}\left((\mathbb{I} - \varepsilon\boldsymbol{A})(\mathbb{I} + \varepsilon^2 \boldsymbol{A}\boldsymbol{A}^\top)^{-\frac{1}{2}}(\boldsymbol{G}^*)^{-1}\left((\mathbb{I} - \varepsilon\boldsymbol{A})(\mathbb{I} + \varepsilon^2 \boldsymbol{A}\boldsymbol{A}^\top)^{-\frac{1}{2}}\right)^\top \boldsymbol{B}\right)$$

$$= \operatorname{tr}\left((\mathbb{I} - \varepsilon\boldsymbol{A})(\mathbb{I} + \varepsilon^2 \boldsymbol{A}\boldsymbol{A}^\top)^{-\frac{1}{2}}(\boldsymbol{G}^*)^{-1}(\mathbb{I} + \varepsilon^2 \boldsymbol{A}\boldsymbol{A}^\top)^{-\frac{1}{2}}(\mathbb{I} + \varepsilon\boldsymbol{A})\boldsymbol{B}\right),$$

which further leads to

$$-\varepsilon \operatorname{tr}\left(\boldsymbol{A}(\boldsymbol{G}^*)^{-1}\boldsymbol{B}\right) + \varepsilon \operatorname{tr}\left((\boldsymbol{G}^*)^{-1}\boldsymbol{A}\boldsymbol{B}\right) + \boldsymbol{o}(\varepsilon) \geq 0.$$

By letting $\varepsilon \to 0$, we further have

$$-\operatorname{tr}\left(\boldsymbol{A}(\boldsymbol{G}^*)^{-1}\boldsymbol{B}\right) + \operatorname{tr}\left((\boldsymbol{G}^*)^{-1}\boldsymbol{A}\boldsymbol{B}\right) = 0,$$

which further leads to

$$\begin{aligned}
0 &= -\operatorname{tr}\left(\boldsymbol{A}(\boldsymbol{G}^*)^{-1}\boldsymbol{B}\right) + \operatorname{tr}\left((\boldsymbol{G}^*)^{-1}\boldsymbol{A}\boldsymbol{B}\right) \\
&= \operatorname{tr}\left(\boldsymbol{A}^\top(\boldsymbol{G}^*)^{-1}\boldsymbol{B}\right) + \operatorname{tr}\left(\boldsymbol{B}(\boldsymbol{G}^*)^{-1}\boldsymbol{A}\right) \\
&= 2\operatorname{tr}\left(\boldsymbol{B}(\boldsymbol{G}^*)^{-1}\boldsymbol{A}\right).
\end{aligned} \tag{12}$$

Since Eq.(12) holds for any anti-symmetry matrix $\boldsymbol{A}$, let $\boldsymbol{A} = \boldsymbol{E}_{i,j} - \boldsymbol{E}_{j,i}$, where $i, j \in [d]$ and $i \neq j$. By Eq.(12), we have

$$\left(\boldsymbol{B}(\boldsymbol{G}^*)^{-1}\right)_{i,j} = \left(\boldsymbol{B}(\boldsymbol{G}^*)^{-1}\right)_{j,i},$$

which further leads to

$$\boldsymbol{B}(\boldsymbol{G}^*)^{-1} = \left(\boldsymbol{B}(\boldsymbol{G}^*)^{-1}\right)^\top = (\boldsymbol{G}^*)^{-\top}\boldsymbol{B}^\top = (\boldsymbol{G}^*)^{-1}\boldsymbol{B}.$$

By simple rearranging, we have

$$\boldsymbol{G}^*\boldsymbol{B} = \boldsymbol{B}\boldsymbol{G}^*.$$

Therefore, by Lemma 8, we have that there exists an orthogonal matrix $\boldsymbol{O}_0$, such that both $\boldsymbol{O}_0\boldsymbol{G}^*\boldsymbol{O}_0^\top$ and $\boldsymbol{O}_0\boldsymbol{B}\boldsymbol{O}_0^\top$ are diagonal. By multiplying a permutation matrix, we further have there exists an orthogonal matrix $\tilde{\boldsymbol{O}}$ such that $\tilde{\boldsymbol{O}}\boldsymbol{G}^*\tilde{\boldsymbol{O}}^\top$ is diagonal, and

$$\tilde{\boldsymbol{O}}\boldsymbol{B}\tilde{\boldsymbol{O}}^\top = \operatorname{Diag}(\beta_1, \cdots, \beta_d). \tag{13}$$

Since $\tilde{\boldsymbol{O}}\boldsymbol{G}^*\tilde{\boldsymbol{O}}^\top$ is diagonal, there exists a permutation mapping $\mathcal{T} : [d] \to [d]$, such that

$$\tilde{\boldsymbol{O}}\boldsymbol{G}^*\tilde{\boldsymbol{O}}^\top = \operatorname{Diag}\left(\alpha_{\mathcal{T}(1)}, \cdots, \alpha_{\mathcal{T}(d)}\right). \tag{14}$$

Denote the order of $\beta_i$ $(i = 1, 2, \cdots, d)$ as

$$\beta_1 = \cdots = \beta_{s_1} > \beta_{s_1+1} = \cdots = \beta_{s_1+s_2} > \cdots > \beta_{\sum_{i=1}^{k-1} s_i+1} = \cdots = \beta_{\sum_{i=1}^{k} s_i} > 0, \tag{15}$$

where $\sum_{i=1}^{k} s_i = d$, and we denote $s_0 = 0$. Since $\boldsymbol{G}^*$ is the optimal point of $\operatorname{tr}((\boldsymbol{G}^*)^{-1}\boldsymbol{B})$, for any $1 \leq i < j \leq d$ and $\beta_i > \beta_j$, we have $\alpha_{\mathcal{T}(i)} > \alpha_{\mathcal{T}(j)}$: otherwise, let

$$\boldsymbol{G}' = \tilde{\boldsymbol{O}}^\top \operatorname{Diag}\left(\alpha_{\mathcal{T}(1)}, \cdots, \alpha_{\mathcal{T}(i-1)}, \alpha_{\mathcal{T}(j)}, \alpha_{\mathcal{T}(i+1)}, \cdots, \alpha_{\mathcal{T}(j-1)}, \alpha_{\mathcal{T}(i)}, \alpha_{\mathcal{T}(j+1)}, \cdots, \alpha_{\mathcal{T}(d)}\right) \tilde{\boldsymbol{O}},$$

we have

$$\operatorname{tr}((\boldsymbol{G}^*)^{-1}\boldsymbol{B}) > \operatorname{tr}((\boldsymbol{G}')^{-1}\boldsymbol{B}),$$

which contradicts that $\boldsymbol{G}^*$ is optimal.

Therefore, $\mathcal{T}(\sum_{i=1}^{j} s_i + 1), \cdots, \mathcal{T}(\sum_{i=1}^{j+1} s_i)$ is then a permutation of $\sum_{i=1}^{j} s_i + 1, \cdots, \sum_{i=1}^{j+1} s_i$, and there exists permutation matrix $\boldsymbol{Q}$ such that

$$\boldsymbol{Q} = \operatorname{Diag}\left(\boldsymbol{Q}_1, \cdots, \boldsymbol{Q}_k\right), \tag{16}$$

where $\boldsymbol{Q}_i$ is a $s_i \times s_i$ permutation sub-matrix, such that,

$$\boldsymbol{Q}\operatorname{Diag}\left(\alpha_{\mathcal{T}(1)}, \cdots, \alpha_{\mathcal{T}(d)}\right)\boldsymbol{Q}^\top = \operatorname{Diag}\left(\alpha_1, \cdots, \alpha_d\right). \tag{17}$$

Furthermore, by Eq.(16) and Eq.(15), we have

$$\boldsymbol{Q}\operatorname{Diag}\left(\beta_1, \cdots, \beta_d\right)\boldsymbol{Q}^\top = \operatorname{Diag}\left(\beta_1, \cdots, \beta_d\right). \tag{18}$$

Therefore, by Eqs.(13), (14), (17), and (18), we have

$$\boldsymbol{Q}\tilde{\boldsymbol{O}}\boldsymbol{B}\left(\boldsymbol{Q}\tilde{\boldsymbol{O}}\right)^{\top} = \mathrm{Diag}\left(\beta_1, \cdots, \beta_d\right),$$

$$\boldsymbol{Q}\tilde{\boldsymbol{O}}\boldsymbol{G}^*\left(\boldsymbol{Q}\tilde{\boldsymbol{O}}\right)^{\top} = \mathrm{Diag}\left(\alpha_1, \cdots, \alpha_d\right).$$

Furthermore,

$$
\begin{aligned}
&\mathrm{tr}\left(\boldsymbol{G}^{-1}\boldsymbol{B}\right) \\
&= \mathrm{tr}\left(\left(\boldsymbol{Q}\tilde{\boldsymbol{O}}\right)^{\top}\mathrm{Diag}\left(\alpha_1^{-1}, \cdots, \alpha_d^{-1}\right)\left(\boldsymbol{Q}\tilde{\boldsymbol{O}}\right)\left(\boldsymbol{Q}\tilde{\boldsymbol{O}}\right)^{\top}\mathrm{Diag}\left(\beta_1, \cdots, \beta_d\right)\left(\boldsymbol{Q}\tilde{\boldsymbol{O}}\right)\right) \\
&= \sum_{i=1}^{d}\frac{\beta_i}{\alpha_i}.
\end{aligned}
$$

Therefore, the optimal value of $\mathrm{tr}(\boldsymbol{G}^{-1}\boldsymbol{B})$ is $\sum_{i=1}^{d}\frac{\beta_i}{\alpha_i}$, and the corresponding optimal point $\boldsymbol{G}^*$ belongs to the following set

$$\mathcal{G} = \{\boldsymbol{O}^{\top}\mathrm{Diag}(\alpha_1, \cdots, \alpha_d)\boldsymbol{O} : \boldsymbol{B} = \boldsymbol{O}^{\top}\mathrm{Diag}(\beta_1, \cdots, \beta_d)\boldsymbol{O}\}.$$

On the other hand, it is easy to verify that for any element $\boldsymbol{G} \in \mathcal{G}$,

$$\mathrm{tr}(\boldsymbol{G}^{-1}\boldsymbol{B}) = \sum_{i=1}^{d}\frac{\beta_i}{\alpha_i}.$$

The proof is completed. $\qquad\qquad\square$

Lemma 10 indicates that with eigenvalues fixed, the eigenvectors of $\boldsymbol{G}$ should agree with those of $\boldsymbol{B}$ by the order of eigenvalues. We then provide the following lemma to determine the optimal eigenvalues.

**Lemma 11.** *Let $\beta_1, \beta_2, \cdots, \beta_d$ be a series of fixed positive reals. Let $\alpha_1, \alpha_2, \cdots, \alpha_d \in \mathbb{R}^+$ be a series of real variables with constraint $\sum_{i=1}^{d}\alpha_i = c$, where $c$ is a positive real constant which satisfies $c \leq \sum_{i=1}^{d}\beta_i$. Then the minimum of function*

$$f(\alpha_1, \cdots, \alpha_d) = \sum_{i=1}^{d}\frac{\beta_i}{\alpha_i} + \sum_{i=1}^{d}\ln\alpha_i$$

*is achieved at*

$$\alpha_i^* = \frac{\sqrt{1 - 4\lambda^*\beta_i} - 1}{-2\lambda^*},$$

*where $\lambda^* \leq 0$ is the unique solution of*

$$\sum_{i=1}^{d}\frac{2\beta_i}{1 + \sqrt{1 - 4\lambda^*\beta_i}} = c.$$

*Proof.* We find the minimum of $f$ under the constraint that $\alpha_1 + \cdots + \alpha_d = c$ by the method of Lagrange Multiplier. Specifically, as for any $i \in [d]$, $\alpha_i \to 0^+$ or $\alpha_i \to c^-$ will lead to $f(\alpha_1, \cdots, \alpha_d) \to \infty$, we have that for any global optimal (minimal) point $(\alpha_1^*, \cdots, \alpha_d^*)$ of $f$ under the constraint $\alpha_1 + \cdots + \alpha_d = c$, we have that there exist a real $\lambda^*$, such that $((\alpha_1^*, \cdots, \alpha_d^*), \lambda^*)$ is a saddle point of $\mathcal{L}((\alpha_1, \cdots, \alpha_d), \lambda)$, which is defined as

$$\mathcal{L}((\alpha_1, \cdots, \alpha_d), \lambda) = f(\alpha_1, \cdots, \alpha_d) + \lambda(c - \alpha_1 - \cdots - \alpha_d).$$

By taking partial derivative of $\mathcal{L}$ with respect to $\alpha_i$, we have

$$-\lambda^* = -\frac{1}{\alpha_i} + \frac{\beta_i}{\alpha_i^2} = \frac{\beta_i - \alpha_i}{\alpha_i^2}, \tag{19}$$

which further leads to

$$\sum_{i=1}^{d} \beta_i - c = \sum_{i=1}^{d} (\beta_i - \alpha_i) = -\lambda^* \left( \sum_{i=1}^{d} \alpha_i^2 \right).$$

Since $\sum_{i=1}^{d} \beta_i \geq c$, we have $\lambda^* \leq 0$. Therefore, for any $i \in [d]$, the quadratic equation $\beta_i x^2 - x + \lambda^* = 0$ has only one positive solution $\frac{1+\sqrt{1-4\lambda^*\beta_i}}{2\beta_i}$, and

$$\alpha_i^* = \frac{2\beta_i}{1 + \sqrt{1 - 4\lambda^*\beta_i}} = \frac{\sqrt{1 - 4\lambda^*\beta_i} - 1}{-2\lambda^*}.$$

On the other hand, by taking derivative of $\mathcal{L}$ with respect to $\lambda^*$, we have

$$\sum_{i=1}^{d} \alpha_i^* = \sum_{i=1}^{d} \frac{2\beta_i}{1 + \sqrt{1 - 4\lambda^*\beta_i}} = c. \tag{20}$$

Since $\sum_{i=1}^{d} \alpha_i^* = \sum_{i=1}^{d} \frac{2\beta_i}{1+\sqrt{1-4\lambda^*\beta_i}}$ is a monotonously increasing function of $\lambda^*$, there is only one solution of $\lambda^*$ of Eq.(20).

The proof is completed. $\qquad\square$

The proof of Lemma 9 can then be obtained by combining Lemma 10 and Lemma 11 together.

*Proof of Lemma 9.* The original optimization problem can be written as

$$\min_{\mathrm{tr}(\boldsymbol{G})=c} \mathrm{tr}\left(\boldsymbol{G}^{-1}\boldsymbol{B}\right) + \ln\left(\det \boldsymbol{G}\right),$$

which can be further decomposed into

$$\min_{\mathrm{tr}(\boldsymbol{G})=c} \mathrm{tr}\left(\boldsymbol{G}^{-1}\boldsymbol{B}\right) + \ln\left(\det \boldsymbol{G}\right)$$

$$= \min_{\substack{\sum_{i=1}^{d}\alpha_i=c \\ \alpha_1 \geq \cdots \geq \alpha_d > 0}} \min_{\boldsymbol{O} \in \mathcal{O}(d)} \left( \mathrm{tr}\left(\boldsymbol{O}^\top \mathrm{Diag}\left(\alpha_1^{-1}, \cdots, \alpha_d^{-1}\right)\boldsymbol{O}\boldsymbol{B}\right) + \sum_{i=1}^{d} \ln \alpha_i \right)$$

$$\stackrel{(*)}{=} \min_{\sum_{i=1}^{d}\alpha_i=c} \left( \sum_{i=1}^{d} \frac{\beta_i}{\alpha_i} + \sum_{i=1}^{d} \ln \alpha_i \right)$$

$$\stackrel{(**)}{=} \sum_{i=1}^{d} \frac{1 + \sqrt{1 - 4\lambda^*\beta_i}}{2} + \sum_{i=1}^{d} \ln \frac{2\beta_i}{1 + \sqrt{1 - 4\lambda^*\beta_i}},$$

where Eq. $(*)$ is due to Lemma 10, Eq. $(**)$ is due to Lemma 11, and $\lambda^* \leq 0$ is the unique solution of

$$\sum_{i=1}^{d} \frac{2\beta_i}{1 + \sqrt{1 - 4\lambda^*\beta_i}} = c.$$

Furthermore, the optimal point of $\mathrm{tr}(\boldsymbol{G}^{-1}\boldsymbol{B}) + \ln(\det \boldsymbol{G})$ can be calculated as

$$\arg\min_{\mathrm{tr}(\boldsymbol{G})=c} \mathrm{tr}\left(\boldsymbol{G}^{-1}\boldsymbol{B}\right) + \ln\left(\det \boldsymbol{G}\right)$$

$$= \left\{ \boldsymbol{O}^\top \mathrm{Diag}\left( \frac{\sqrt{1 - 4\lambda^*\beta_1} - 1}{-2\lambda^*}, \cdots, \frac{\sqrt{1 - 4\lambda^*\beta_d} - 1}{-2\lambda^*} \right) \boldsymbol{O} : \boldsymbol{B} = \boldsymbol{O}^\top \mathrm{Diag}(\beta_1, \cdots, \beta_d)\boldsymbol{O}, \right.$$

$$\left. \lambda^* = \arg_\lambda \left( \sum_{i=1}^{d} \frac{2\beta_i}{1 + \sqrt{1 - 4\lambda\beta_i}} = c \right) \right\}.$$

The proof is completed. $\qquad\square$

# B Supplementary Materials of Section 3.1

*Proof of Lemma 2.* The $\beta$-smooth condition gives

$$\mathcal{R}_{\boldsymbol{S}}(\boldsymbol{W}_{t+1}) \leq \mathcal{R}_{\boldsymbol{S}}(\boldsymbol{W}_t) + \langle \nabla \mathcal{R}_{\boldsymbol{S}}(\boldsymbol{W}_t), \boldsymbol{W}_{t+1} - \boldsymbol{W}_t \rangle + \frac{\beta}{2} \|\boldsymbol{W}_{t+1} - \boldsymbol{W}_t\|^2. \tag{21}$$

Based on the update rule Eq.(3), we have

$$\boldsymbol{W}_{t+1} - \boldsymbol{W}_t = -\eta_{t+1} \nabla \mathcal{R}_{\boldsymbol{S}_{\boldsymbol{V}_{t+1}}}(\boldsymbol{W}_t) + \varepsilon_{t+1}, \tag{22}$$

where $\varepsilon_{t+1} \sim \mathcal{N}(0, \boldsymbol{\Sigma}_{t+1}(\boldsymbol{S}, \boldsymbol{W}_t))$.

Take expectation on Eq.(21) with respect to $\boldsymbol{W}_{t+1} | \boldsymbol{W}_t$, by $\mathbb{E}^{\boldsymbol{W}_t}(\nabla \mathcal{R}_{\boldsymbol{S}_{\boldsymbol{V}_{t+1}}}(\boldsymbol{W}_t)) = \nabla \mathcal{R}_{\boldsymbol{S}}(\boldsymbol{W}_t)$,

$$\mathbb{E}^{\boldsymbol{W}_t}[\mathcal{R}_{\boldsymbol{S}}(\boldsymbol{W}_{t+1})] \leq \mathcal{R}_{\boldsymbol{S}}(\boldsymbol{W}_t) - \eta_{t+1} \|\nabla \mathcal{R}_{\boldsymbol{S}}(\boldsymbol{W}_t)\|^2 + \frac{\beta}{2} \mathbb{E}^{\boldsymbol{W}_t} \|\boldsymbol{W}_{t+1} - \boldsymbol{W}_t\|^2. \tag{23}$$

Furthermore,

$$\begin{aligned}
&\mathbb{E}^{\boldsymbol{W}_t} \|\boldsymbol{W}_{t+1} - \boldsymbol{W}_t\|^2 \\
&= \mathbb{E}^{\boldsymbol{W}_t} \| - \eta_{t+1} \nabla \mathcal{R}_{\boldsymbol{S}_{\boldsymbol{V}_{t+1}}}(\boldsymbol{W}_t) + \varepsilon_{t+1}\|^2 \\
&\overset{(*)}{=} \mathbb{E}^{\boldsymbol{W}_t} \| - \eta_{t+1} \nabla \mathcal{R}_{\boldsymbol{S}_{\boldsymbol{V}_t}}(\boldsymbol{W}_t)\| + \mathbb{E}^{\boldsymbol{W}_t} \|\varepsilon_{t+1}\|^2 \\
&= \eta_{t+1}^2 \mathbb{E}^{\boldsymbol{W}_t} \|\nabla \mathcal{R}_{\boldsymbol{S}_{\boldsymbol{V}_{t+1}}}(\boldsymbol{W}_t) - \nabla \mathcal{R}_{\boldsymbol{S}}(\boldsymbol{W}_t) + \nabla \mathcal{R}_{\boldsymbol{S}}(\boldsymbol{W}_t)\|^2 + \boldsymbol{\Sigma}_{t+1}(\boldsymbol{S}, \boldsymbol{W}_t) \\
&= \eta_{t+1}^2 \mathbb{E}^{\boldsymbol{W}_t} \|\nabla \mathcal{R}_{\boldsymbol{S}_{\boldsymbol{V}_{t+1}}}(\boldsymbol{W}_t) - \nabla \mathcal{R}_{\boldsymbol{S}}(\boldsymbol{W}_t)\|^2 + \eta_{t+1}^2 \|\nabla \mathcal{R}_{\boldsymbol{S}}(\boldsymbol{W}_t)\|^2 + \boldsymbol{\Sigma}_{t+1}(\boldsymbol{S}, \boldsymbol{W}_t) \\
&= \frac{\eta_{t+1}^2}{N-1} \frac{N - b_{t+1}}{b_{t+1}} \boldsymbol{\Sigma}_{\boldsymbol{S}, \boldsymbol{W}_t}^{sd} + \eta_{t+1}^2 \|\nabla \mathcal{R}_{\boldsymbol{S}}(\boldsymbol{W}_t)\|^2 + \boldsymbol{\Sigma}_{t+1}(\boldsymbol{S}, \boldsymbol{W}_t),
\end{aligned} \tag{24}$$

where Eq.$(*)$ is due to $\varepsilon_{t+1}$ is independent of $\boldsymbol{V}_{t+1}$, and $\mathbb{E}^{\boldsymbol{W}_t} \varepsilon_{t+1} = 0$.

Applying Eq.(24) back to Eq.(23) completes the proof.

$\square$

# C Supplementary Materials of Section 3.2

## C.1 Example to illustrate the difficulty to apply Proposition 1 to solve Problem 1

In this section, we show an example to demonstrate the difficulty for tackling **Problem 1** through Proposition 1. To start with, by the definition of state-dependent SGLD (Eq.(3)), covariance $\boldsymbol{\Sigma}_{[T]}$ is independent of $\boldsymbol{J}$ and $\boldsymbol{V}_{[T]}$. Therefore, the square root separates the expectation with respect to $\boldsymbol{V}_{[T]}$ and $\boldsymbol{J}$ from the KL divergence term in the generalization bound

$$\mathbb{E}_{\boldsymbol{S}, \boldsymbol{V}_{[T]}, \boldsymbol{J}} \sqrt{\frac{(a_2 - a_1)^2}{2} \sum_{s=1}^{T} \mathbb{E}_{Q_{s-1}^{\boldsymbol{S}, \boldsymbol{V}_{[T]}}} \mathrm{KL}\left(Q_{s|(s-1)}^{\boldsymbol{S}, \boldsymbol{V}_{[T]}} \middle\| P_{s|(s-1)}^{\boldsymbol{J}, \boldsymbol{S}_{\boldsymbol{J}}, \boldsymbol{V}_{[T]}}\right)},$$

which makes the dependency of the bound on $\boldsymbol{\Sigma}_{[T]}$ even more complex. However, even though we change the optimization target into

$$\mathbb{E}_{\boldsymbol{S}} \sqrt{\frac{(a_2 - a_1)^2}{2} \mathbb{E}_{\boldsymbol{V}_{[T]}, \boldsymbol{J}} \sum_{s=1}^{T} \mathbb{E}_{Q_{s-1}^{\boldsymbol{S}, \boldsymbol{V}_{[T]}}} \mathrm{KL}\left(Q_{s|(s-1)}^{\boldsymbol{S}, \boldsymbol{V}_{[T]}} \middle\| P_{s|(s-1)}^{\boldsymbol{J}, \boldsymbol{S}_{\boldsymbol{J}}, \boldsymbol{V}_{[T]}}\right)}, \tag{25}$$

which is still a generalization bound by Jensen's Inequality, we demonstrate that the dependency on $\boldsymbol{\Sigma}_{[T]}$ is still too complex to tackle as follows.

To optimize Eq.(25) with respect to $\boldsymbol{\Sigma}_{[T]}(\boldsymbol{S}, \cdot)$ for fixed $\boldsymbol{S}$, we are actually seeking the optimal point of the following optimization problem:

$$\boldsymbol{\Sigma}_{[T]}^*(\boldsymbol{S}, \cdot) = \arg \min_{\boldsymbol{\Sigma}_{[T]}(\boldsymbol{S}, \cdot)} \sqrt{\mathbb{E}_{\boldsymbol{V}_{[T]}, \boldsymbol{J}} \sum_{s=1}^{T} \mathbb{E}_{Q_{s-1}^{\boldsymbol{S}, \boldsymbol{V}_{[T]}}} \mathrm{KL}\left(Q_{s|(s-1)}^{\boldsymbol{S}, \boldsymbol{V}_{[T]}} \middle\| P_{s|(s-1)}^{\boldsymbol{J}, \boldsymbol{S}_{\boldsymbol{J}}, \boldsymbol{V}_{[T]}}\right)}. \tag{26}$$

However, we will show it is technically hard to solve Eq. (26). As discussed in Section 3.2.1, for any fixed index $i \in [T]$, Eq.(26) depends on $\mathbf{\Sigma}_s(\mathbf{S}, \cdot)$ through both $\mathbb{E}_{\mathbf{V}_{[T]}, \mathbf{J}} \mathbb{E}_{Q_{s-1}^{S, \mathbf{v}_{[T]}}}$ $\mathrm{KL}(Q_{s|(s-1)}^{\mathbf{S}, \mathbf{V}_{[T]}} \| P_{s|(s-1)}^{\mathbf{J}, \mathbf{S}_{\mathbf{J}}, \mathbf{V}_{[T]}})$ and $\mathbb{E}_{\mathbf{V}_{[T]}, \mathbf{J}} \mathbb{E}_{Q_{i-1}^{S, \mathbf{v}_{[T]}}} \mathrm{KL}(Q_{i|(i-1)}^{\mathbf{S}, \mathbf{V}_{[T]}} \| P_{i|(i-1)}^{\mathbf{J}, \mathbf{S}_{\mathbf{J}}, \mathbf{V}_{[T]}})$ for $\forall i > s$. Specifically, we adopt the update rule for prior for all the steps and posterior for all steps $t \neq s$ to be the isotropic SGLD in [25], i.e.,

$$\text{Posterior: } \boldsymbol{W}_t = \boldsymbol{W}_{t-1} - \eta_t \nabla \mathcal{R}_{\boldsymbol{S}_{\boldsymbol{V}_t}}(\boldsymbol{W}_{t-1}) + \mathcal{N}(\mathbf{0}, \sigma_t \mathbb{I})$$

$$\text{Prior: } \boldsymbol{W}_t = \boldsymbol{W}_{t-1} - \eta_t \left( \frac{|\boldsymbol{V}_t \cap \boldsymbol{J}|}{|\boldsymbol{V}_t|} \nabla \mathcal{R}_{\boldsymbol{S}_{\boldsymbol{V}_t \cap \boldsymbol{J}}}(\boldsymbol{W}_{t-1}) + \frac{|\boldsymbol{V}_t \cap \boldsymbol{J}^c|}{|\boldsymbol{V}_t|} \nabla \mathcal{R}_{\boldsymbol{S}_{\boldsymbol{J}}}(\boldsymbol{W}_{t-1}) \right) + \mathcal{N}(\mathbf{0}, \sigma_t \mathbb{I}),$$

while we only optimize the noise covariance $\mathbf{\Sigma}_s(\mathbf{S}, \cdot)$ of step $s$:

$$\boldsymbol{W}_s = \boldsymbol{W}_{s-1} - \eta_s \nabla \mathcal{R}_{\boldsymbol{S}_{\boldsymbol{V}_s}}(\boldsymbol{W}_{s-1}) + \mathcal{N}(\mathbf{0}, \mathbf{\Sigma}_s(\boldsymbol{S}, \boldsymbol{W}_{s-1})).$$

By simple calculation, for any step $t \in [T]$, given the same $\boldsymbol{W}_{t-1}$, $\boldsymbol{V}_t$, $\boldsymbol{J}$, and $\boldsymbol{S}$, the mean between the prior and posterior can be calculated as

$$\mu^{\boldsymbol{S}, \boldsymbol{V}_t, \boldsymbol{J}, \boldsymbol{W}_{t-1}}$$
$$= -\eta_t \left( \frac{|\boldsymbol{V}_t \cap \boldsymbol{J}|}{|\boldsymbol{V}_t|} \nabla \mathcal{R}_{\boldsymbol{S}_{\boldsymbol{V}_t \cap \boldsymbol{J}}}(\boldsymbol{W}_{t-1}) + \frac{|\boldsymbol{V}_t \cap \boldsymbol{J}^c|}{|\boldsymbol{V}_t|} \nabla \mathcal{R}_{\boldsymbol{S}_{\boldsymbol{J}}}(\boldsymbol{W}_{t-1}) \right) + \eta_t \nabla \mathcal{R}_{\boldsymbol{S}_{\boldsymbol{V}_t}}(\boldsymbol{W}_{t-1})$$
$$= \eta_t \frac{|\boldsymbol{V}_t \cap \boldsymbol{J}^c|}{|\boldsymbol{V}_t|} \left( \nabla \mathcal{R}_{\boldsymbol{S}_{\boldsymbol{V}_t \cap \boldsymbol{J}^c}}(\boldsymbol{W}_{t-1}) - \nabla \mathcal{R}_{\boldsymbol{S}_{\boldsymbol{J}}}(\boldsymbol{W}_{t-1}) \right). \tag{27}$$

Therefore, by Lemma 5 and Lemma 6, the expected KL divergence $\mathbb{E}_{\boldsymbol{V}_{[T]}, \boldsymbol{J}} \mathbb{E}_{Q_{i-1}^{S, \mathbf{v}_{[T]}}} \mathrm{KL}$ $(Q_{i|(i-1)}^{\boldsymbol{S}, \boldsymbol{V}_{[T]}} \| P_{i|(i-1)}^{\boldsymbol{J}, \boldsymbol{S}_{\boldsymbol{J}}, \boldsymbol{V}_{[T]}})$ can be calculated as

$$\mathbb{E}_{\boldsymbol{V}_{[T]}, \boldsymbol{J}} \mathbb{E}_{Q_{i-1}^{S, \mathbf{v}_{[T]}}} \mathrm{KL}\left( Q_{i|(i-1)}^{\boldsymbol{S}, \boldsymbol{V}_{[T]}} \middle\| P_{i|(i-1)}^{\boldsymbol{J}, \boldsymbol{S}_{\boldsymbol{J}}, \boldsymbol{V}_{[T]}} \right)$$
$$= \frac{1}{2} \mathbb{E}_{\boldsymbol{V}_{[T]}, \boldsymbol{J}} \mathbb{E}_{Q_{i-1}^{S, \mathbf{v}_{[T]}}} \left( \sigma_i^{-1} \mu^{\boldsymbol{S}, \boldsymbol{V}_t, \boldsymbol{J}, \boldsymbol{W}_{i-1}} \left( \mu^{\boldsymbol{S}, \boldsymbol{V}_t, \boldsymbol{J}, \boldsymbol{W}_{i-1}} \right)^\top \right)$$
$$= \frac{1}{2\sigma_i} \frac{1}{Nb_i} \left( \frac{N}{N-1} \right)^2 \mathbb{E}_{\boldsymbol{V}_{[i-1]}} \mathbb{E}_{Q_{i-1}^{S, \mathbf{v}_{[i-1]}}} \mathbf{\Sigma}_{\boldsymbol{S}, \boldsymbol{W}_{i-1}}^{sd}.$$

Therefore, the exact form of $\mathbb{E}_{\boldsymbol{V}_{[T]}, \boldsymbol{J}} \mathbb{E}_{Q_{i-1}^{S, \mathbf{v}_{[T]}}} \mathrm{KL}(Q_{i|(i-1)}^{\boldsymbol{S}, \boldsymbol{V}_{[T]}} \| P_{i|(i-1)}^{\boldsymbol{J}, \boldsymbol{S}_{\boldsymbol{J}}, \boldsymbol{V}_{[T]}})$ requires taking expectation to $\mathbf{\Sigma}_{\boldsymbol{S}, \boldsymbol{W}_{i-1}}^{sd}$ with respect to Gaussian distribution with covariance $\mathbf{\Sigma}_s$, and can be complex due to the complex structure of the model. Specifically, if $i = s + 1$, then $\mathbb{E}_{\boldsymbol{V}_{[T]}, \boldsymbol{J}} \mathbb{E}_{Q_{i-1}^{S, \mathbf{v}_{[T]}}}$ $\mathrm{KL}(Q_{i|(i-1)}^{\boldsymbol{S}, \boldsymbol{V}_{[T]}} \| P_{i|(i-1)}^{\boldsymbol{J}, \boldsymbol{S}_{\boldsymbol{J}}, \boldsymbol{V}_{[T]}})$ can be further written as

$$\mathbb{E}_{\boldsymbol{V}_{[T]}, \boldsymbol{J}} \mathbb{E}_{Q_s^{S, \mathbf{v}_{[T]}}} \mathrm{KL}\left( Q_{s+1|s}^{\boldsymbol{S}, \boldsymbol{V}_{[T]}} \middle\| P_{s+1|s}^{\boldsymbol{J}, \boldsymbol{S}_{\boldsymbol{J}}, \boldsymbol{V}_{[T]}} \right)$$
$$= \frac{1}{2} \frac{1}{Nb_{s+1}} \left( \frac{N}{N-1} \right)^2 \mathbb{E}_{\boldsymbol{V}_{[s]}} \mathbb{E}_{Q_{s-1}^{S, \mathbf{v}_{[s-1]}}} \mathbb{E}_{Q_{s|(s-1)}^{S, \mathbf{v}_s}} \mathbf{\Sigma}_{\boldsymbol{S}, \boldsymbol{W}_s}^{sd}.$$

Therefore, we need to optimize $\mathbb{E}_{\boldsymbol{V}_s} \mathbb{E}_{Q_{s|(s-1)}^{S, \mathbf{v}_s}} \mathbf{\Sigma}_{\boldsymbol{S}, \boldsymbol{W}_s}^{sd}$, which can be further written as

$$\mathbb{E}_{\boldsymbol{V}_s} \mathbb{E}_{Q_{s|(s-1)}^{S, \mathbf{v}_s}} \mathbf{\Sigma}_{\boldsymbol{S}, \boldsymbol{W}_s}^{sd} = \mathbb{E}_{\boldsymbol{V}_s} \mathbb{E}_{\mathcal{N}(-\eta_s \nabla \mathcal{R}_{\boldsymbol{S}_{\boldsymbol{V}_s}}(\boldsymbol{W}_{s-1}), \mathbf{\Sigma}_s(\boldsymbol{S}, \boldsymbol{W}_{s-1}))} \mathbf{\Sigma}_{\boldsymbol{S}, \boldsymbol{W}_s}^{sd}.$$

The explicit form of $\mathbb{E}_{\boldsymbol{V}_s} \mathbb{E}_{\mathcal{N}(-\eta_s \nabla \mathcal{R}_{\boldsymbol{S}_{\boldsymbol{V}_s}}(\boldsymbol{W}_{s-1}), \mathbf{\Sigma}_s(\boldsymbol{S}, \boldsymbol{W}_{s-1}))} \mathbf{\Sigma}_{\boldsymbol{S}, \boldsymbol{W}_s}^{sd}$ can be obtained only when $\mathbf{\Sigma}_{\boldsymbol{S}, \boldsymbol{W}_s}^{sd}$ is some simple functions with respect to $\boldsymbol{W}_s$ (e.g. quadratic functions), which makes the optimal of $\mathbb{E}_{\boldsymbol{V}_s} \mathbb{E}_{\mathcal{N}(-\eta_s \nabla \mathcal{R}_{\boldsymbol{S}_{\boldsymbol{V}_s}}(\boldsymbol{W}_{s-1}), \mathbf{\Sigma}_s(\boldsymbol{S}, \boldsymbol{W}_{s-1}))} \mathbf{\Sigma}_{\boldsymbol{S}, \boldsymbol{W}_s}^{sd}$ complicated due to the complex structure of $\mathcal{R}_{\boldsymbol{S}}$ and $\mathbf{\Sigma}_{\boldsymbol{S}, \boldsymbol{W}}^{sd}$ in pratical learning problems.

## C.2 Proof of Theorem 1

*Proof of Theorem 1.* For any two random measures $P^{\boldsymbol{J},\boldsymbol{S_J},\boldsymbol{V}_{[T]}}, Q^{\boldsymbol{S},\boldsymbol{V}_{[T]}}$, by the Donsker-Varadhan variational formula [3], for any function $g$ satisfying $Q^{\boldsymbol{S},\boldsymbol{V}_{[T]}}(\exp g) < \infty$, we have

$$\mathrm{KL}(P^{\boldsymbol{J},\boldsymbol{S_J},\boldsymbol{V}_{[T]}} || Q^{\boldsymbol{S},\boldsymbol{V}_{[T]}}) \geq P^{\boldsymbol{J},\boldsymbol{S_J},\boldsymbol{V}_{[T]}}(g) - Q^{\boldsymbol{S},\boldsymbol{V}_{[T]}}(g) - \log Q^{\boldsymbol{S},\boldsymbol{V}_{[T]}}\left(\exp(g - Q^{\boldsymbol{S},\boldsymbol{V}_{[T]}}(g))\right).$$

Letting $g(\boldsymbol{W}) = \lambda\left(\hat{\mathcal{R}}_{\boldsymbol{S_{J^c}}}(\boldsymbol{W}) - \mathcal{R}_{\mathcal{D}}(\boldsymbol{W})\right)$, we further have

$$\mathrm{KL}(P^{\boldsymbol{J},\boldsymbol{S_J},\boldsymbol{V}_{[T]}} || Q^{\boldsymbol{S},\boldsymbol{V}_{[T]}})$$
$$\geq \lambda\left(\mathcal{R}_{\mathcal{D}}(Q^{\boldsymbol{S},\boldsymbol{V}_{[T]}}) - \hat{\mathcal{R}}_{\boldsymbol{S_{J^c}}}(Q^{\boldsymbol{S},\boldsymbol{V}_{[T]}}) - \left(\mathcal{R}_{\mathcal{D}}(P^{\boldsymbol{J},\boldsymbol{S_J},\boldsymbol{V}_{[T]}}) - \hat{\mathcal{R}}_{\boldsymbol{S_{J^c}}}(P^{\boldsymbol{J},\boldsymbol{S_J},\boldsymbol{V}_{[T]}})\right)\right)$$
$$- \log Q^{\boldsymbol{S},\boldsymbol{V}_{[T]}}\left(\exp\left(\lambda\left(\hat{\mathcal{R}}_{\boldsymbol{S_{J^c}}} - \mathcal{R}_{\mathcal{D}} - \left(\hat{\mathcal{R}}_{\boldsymbol{S_{J^c}}}(Q^{\boldsymbol{S},\boldsymbol{V}_{[T]}}) - \mathcal{R}_{\mathcal{D}}(Q^{\boldsymbol{S},\boldsymbol{V}_{[T]}})\right)\right)\right)\right).$$

On the other hand, since $\ell \in [a_1, a_2]$, $\lambda\left(\hat{\mathcal{R}}_{\boldsymbol{S_{J^c}}}(\boldsymbol{W}) - \mathcal{R}_{\mathcal{D}}(\boldsymbol{W})\right)$ is $\frac{\lambda(a_2 - a_1)}{2}$ subgaussian. Therefore,

$$\left(\mathcal{R}_{\mathcal{D}}(Q^{\boldsymbol{S},\boldsymbol{V}_{[T]}}) - \hat{\mathcal{R}}_{\boldsymbol{S_{J^c}}}(Q^{\boldsymbol{S},\boldsymbol{V}_{[T]}})\right) - \left(\mathcal{R}_{\mathcal{D}}(P^{\boldsymbol{J},\boldsymbol{S_J},\boldsymbol{V}_{[T]}}) - \hat{\mathcal{R}}_{\boldsymbol{S_{J^c}}}(P^{\boldsymbol{J},\boldsymbol{S_J},\boldsymbol{V}_{[T]}})\right)$$
$$\leq \inf_{\lambda > 0} \frac{\mathrm{KL}(P^{\boldsymbol{J},\boldsymbol{S_J},\boldsymbol{V}_{[T]}} || Q^{\boldsymbol{S},\boldsymbol{V}_{[T]}}) + \frac{1}{8}\lambda^2(a_2 - a_1)^2}{\lambda}.$$

Since $P^{\boldsymbol{J},\boldsymbol{S_J},\boldsymbol{V}_{[T]}}$ is independent of $\boldsymbol{S_{J^c}}$ then we have $\mathbb{E}^{\boldsymbol{S_J},\boldsymbol{J},\boldsymbol{V}_{[T]}}\left[\mathcal{R}_{\mathcal{D}}(P^{\boldsymbol{J},\boldsymbol{S_J},\boldsymbol{V}_{[T]}}) - \hat{\mathcal{R}}_{\boldsymbol{S_{J^c}}}(P^{\boldsymbol{J},\boldsymbol{S_J},\boldsymbol{V}_{[T]}})\right] = 0$. Hence, by averaging over $\boldsymbol{S_{J^c}}$ (equivalently, taking the conditional expectation conditional on $(\boldsymbol{S_J}, \boldsymbol{J}, \boldsymbol{V}_{[T]})$) we have, with probability one

$$\mathbb{E}^{\boldsymbol{S_J},\boldsymbol{J},\boldsymbol{V}_{[T]}}\left[\mathcal{R}_{\mathcal{D}}(Q^{\boldsymbol{S},\boldsymbol{V}_{[T]}}) - \hat{\mathcal{R}}_{\boldsymbol{S_{J^c}}}(Q^{\boldsymbol{S},\boldsymbol{V}_{[T]}})\right]$$
$$= \mathbb{E}^{\boldsymbol{S_J},\boldsymbol{J},\boldsymbol{V}_{[T]}}\left[\mathcal{R}_{\mathcal{D}}(Q^{\boldsymbol{S},\boldsymbol{V}_{[T]}}) - \hat{\mathcal{R}}_{\boldsymbol{S_{J^c}}}(Q^{\boldsymbol{S},\boldsymbol{V}_{[T]}}) - \left(\mathcal{R}_{\mathcal{D}}(P^{\boldsymbol{J},\boldsymbol{S_J},\boldsymbol{V}_{[T]}}) - \hat{\mathcal{R}}_{\boldsymbol{S_{J^c}}}(P^{\boldsymbol{J},\boldsymbol{S_J},\boldsymbol{V}_{[T]}})\right)\right]$$
$$\leq \mathbb{E}^{\boldsymbol{S_J},\boldsymbol{J},\boldsymbol{V}_{[T]}}\left(\inf_{\lambda > 0} \frac{\mathrm{KL}(P^{\boldsymbol{J},\boldsymbol{S_J},\boldsymbol{V}_{[T]}} || Q^{\boldsymbol{S},\boldsymbol{V}_{[T]}}) + \frac{1}{8}\lambda^2(a_2 - a_1)^2}{\lambda}\right)$$

Finally, by taking the full expectation, since $\boldsymbol{J} \perp\!\!\!\perp Q^{\boldsymbol{S},\boldsymbol{V}_{[T]}}$ we get:

$$\mathbb{E}_{\boldsymbol{S},\boldsymbol{V}_{[T]}}\left[\mathcal{R}_{\mathcal{D}}(Q^{\boldsymbol{S},\boldsymbol{V}_{[T]}}) - \hat{\mathcal{R}}_{\boldsymbol{S}}(Q^{\boldsymbol{S},\boldsymbol{V}_{[T]}})\right] \leq \mathbb{E}_{\boldsymbol{S},\boldsymbol{V}_{[T]},\boldsymbol{J}}\left[\inf_{\lambda > 0} \frac{\mathrm{KL}(P^{\boldsymbol{J},\boldsymbol{S_J},\boldsymbol{V}_{[T]}} || Q^{\boldsymbol{S},\boldsymbol{V}_{[T]}}) + \frac{1}{8}\lambda^2(a_2 - a_1)^2}{\lambda}\right]$$

where the final $\mathrm{KL}(P^{\boldsymbol{J},\boldsymbol{S_J},\boldsymbol{V}_{[T]}} || Q^{\boldsymbol{S},\boldsymbol{V}_{[T]}})$ on the right hand side is between two random measures, and hence is a random variable depending on $(\boldsymbol{S}, \boldsymbol{J}, \boldsymbol{V}_{[T]})$; and the expectation on the right hand side integrates over $(\boldsymbol{S}, \boldsymbol{J}, \boldsymbol{V}_{[T]})$.

Since

$$\frac{\mathrm{KL}(P^{\boldsymbol{J},\boldsymbol{S_J},\boldsymbol{V}_{[T]}} || Q^{\boldsymbol{S},\boldsymbol{V}_{[T]}}) + \frac{1}{8}\lambda^2(a_2 - a_1)^2}{\lambda} \geq \sqrt{\frac{1}{2}(a_2 - a_1)^2 \mathrm{KL}(P^{\boldsymbol{J},\boldsymbol{S_J},\boldsymbol{V}_{[T]}} || Q^{\boldsymbol{S},\boldsymbol{V}_{[T]}})},$$

the proof is completed. $\qquad\square$

## D  Supplementary of Section 4

In this section, we provide the proof of Theorem 2. Specifically, as mentioned in the main body, optimizing $\mathrm{Gen}_T$ with greedily selected prior involves three steps. (1). we first prove Lemma 3, which provides the optimal solution of noise covariance and prior for one single KL divergence term in the generalization bound $\mathrm{Gen}_T$; (2). as the optimal solution of noise covariance in Lemma 3 is independent of $\boldsymbol{S_J}$, $\boldsymbol{V}_{[T]}$, and $\boldsymbol{V}_{[T]}$, we are then able to obtain the greedy prior by Lemma 4; (3). applying the greedy prior back to $\mathrm{Gen}_T$, we are finally able to derive Theorem 2.

We start by restating Lemma 12 and providing its proof.

**Lemma 12** (Lemma 3, restated). *For any $s \in [T]$, $\boldsymbol{J}$, $\boldsymbol{S_J}$, and $\boldsymbol{V}_{[T]}$, under Constraint 1,*

$$\min_{P^{\boldsymbol{J},\boldsymbol{S_J},\boldsymbol{V}_s}_{s|(s-1)}} \mathbb{E}_{\boldsymbol{S}_{\boldsymbol{J}^c} \sim \mathcal{D}} \operatorname{KL}\left(P^{\boldsymbol{J},\boldsymbol{S_J},\boldsymbol{V}_s}_{s|(s-1)} \middle\| Q^{\boldsymbol{S},\boldsymbol{V}_s}_{s|(s-1)}\right) \tag{28}$$

*(1). is independent of $\boldsymbol{\Sigma}_s$ when $\boldsymbol{V}_s \cap \boldsymbol{J}^c = \emptyset$, and (2). is minimized at $\boldsymbol{\Sigma}_s(\boldsymbol{W}) = \lambda_s(\boldsymbol{W})\,(\boldsymbol{\Sigma}^{pop}_{\boldsymbol{W}})^{\frac{1}{2}}$, $\forall \boldsymbol{W}$, when $\boldsymbol{V}_s \cap \boldsymbol{J}^c \neq \emptyset$, where $\lambda_s(\boldsymbol{W}) = c_s(\boldsymbol{W})/\operatorname{tr}((\boldsymbol{\Sigma}^{pop}_{\boldsymbol{W}})^{\frac{1}{2}})$.*

*Proof.* We first calculate $\min_{P^{\boldsymbol{J},\boldsymbol{S_J},\boldsymbol{V}_s}_{s|(s-1)}} \mathbb{E}_{\boldsymbol{S}_{\boldsymbol{J}^c} \sim \mathcal{D}} \operatorname{KL}\left(P^{\boldsymbol{J},\boldsymbol{S_J},\boldsymbol{V}_s}_{s|(s-1)} \middle\| Q^{\boldsymbol{S},\boldsymbol{V}_s}_{s|(s-1)}\right)$ for any $\boldsymbol{\Sigma}_s$. By applying the definition of the KL divergence, we have

$$\arg\min_{P^{\boldsymbol{J},\boldsymbol{S_J},\boldsymbol{V}_s}_{s|(s-1)}} \mathbb{E}_{\boldsymbol{S}_{\boldsymbol{J}^c} \sim \mathcal{D}} \operatorname{KL}\left(P^{\boldsymbol{J},\boldsymbol{S_J},\boldsymbol{V}_s}_{s|(s-1)} \middle\| Q^{\boldsymbol{S},\boldsymbol{V}_{[T]}}_{s|(s-1)}\right)$$

$$= \arg\min_{P^{\boldsymbol{J},\boldsymbol{S_J},\boldsymbol{V}_s}_{s|(s-1)}} \mathbb{E}_{\boldsymbol{S}_{\boldsymbol{J}^c} \sim \mathcal{D}} \int P^{\boldsymbol{J},\boldsymbol{S_J},\boldsymbol{V}_s}_{s|(s-1)}(\boldsymbol{W}_s) \log \frac{P^{\boldsymbol{J},\boldsymbol{S_J},\boldsymbol{V}_s}_{s|(s-1)}(\boldsymbol{W}_s)}{Q^{\boldsymbol{S},\boldsymbol{V}_{[T]}}_{s|(s-1)}(\boldsymbol{W}_s)} \mathrm{d}\boldsymbol{W}_s$$

$$\overset{(*)}{=} \arg\min_{P^{\boldsymbol{J},\boldsymbol{S_J},\boldsymbol{V}_s}_{s|(s-1)}} \int P^{\boldsymbol{J},\boldsymbol{S_J},\boldsymbol{V}_s}_{s|(s-1)}(\boldsymbol{W}_s) \log \frac{P^{\boldsymbol{J},\boldsymbol{S_J},\boldsymbol{V}_s}_{s|(s-1)}(\boldsymbol{W}_s)}{e^{\mathbb{E}_{\boldsymbol{S}_{\boldsymbol{J}^c} \sim \mathcal{D}} \log Q^{\boldsymbol{S},\boldsymbol{V}_{[T]}}_{s|(s-1)}(\boldsymbol{W}_s)}} \mathrm{d}\boldsymbol{W}_s, \tag{29}$$

where Eq. $(*)$ is due to the independence of $P$ on $\boldsymbol{S}_{\boldsymbol{J}^c}$.

Let

$$\tilde{Q}^{\boldsymbol{J},\boldsymbol{S_J},\boldsymbol{V}_{[T]}}_{s|(s-1)}(\boldsymbol{W}) = \frac{e^{\mathbb{E}_{\boldsymbol{S}_{\boldsymbol{J}^c} \sim \mathcal{D}} \log Q^{\boldsymbol{S},\boldsymbol{V}_{[T]}}_{s|(s-1)}(\boldsymbol{W})}}{\int e^{\mathbb{E}_{\boldsymbol{S}_{\boldsymbol{J}^c} \sim \mathcal{D}} \log Q^{\boldsymbol{S},\boldsymbol{V}_{[T]}}_{s|(s-1)}(\tilde{\boldsymbol{W}})} \mathrm{d}\tilde{\boldsymbol{W}}}, \tag{30}$$

and $\tilde{Q}^{\boldsymbol{J},\boldsymbol{S_J},\boldsymbol{V}_{[T]}}_{s|(s-1)}(\boldsymbol{W})$ is then a probability measure on $\mathbb{R}^d$. Applying Eq. (30) back to Eq. (29), we obtain

$$\arg\min_{P^{\boldsymbol{J},\boldsymbol{S_J},\boldsymbol{V}_s}_{s|(s-1)}} \left( \int P^{\boldsymbol{J},\boldsymbol{S_J},\boldsymbol{V}_s}_{s|(s-1)}(\boldsymbol{W}_s) \log \frac{P^{\boldsymbol{J},\boldsymbol{S_J},\boldsymbol{V}_s}_{s|(s-1)}(\boldsymbol{W}_s)}{\tilde{Q}^{\boldsymbol{J},\boldsymbol{S_J},\boldsymbol{V}_{[T]}}_{s|(s-1)}(\boldsymbol{W}_s)} \mathrm{d}\boldsymbol{W}_s \right.$$

$$\left. - \int P^{\boldsymbol{J},\boldsymbol{S_J},\boldsymbol{V}_s}_{s|(s-1)}(\boldsymbol{W}_s) \log \left( \int e^{\mathbb{E}_{\boldsymbol{S}_{\boldsymbol{J}^c} \sim \mathcal{D}} \log Q^{\boldsymbol{S},\boldsymbol{V}_{[T]}}_{s|(s-1)}(\tilde{\boldsymbol{W}})} \mathrm{d}\tilde{\boldsymbol{W}} \right) \mathrm{d}\boldsymbol{W}_s \right)$$

$$= \arg\min_{P^{\boldsymbol{J},\boldsymbol{S_J},\boldsymbol{V}_s}_{s|(s-1)}} \left( \int P^{\boldsymbol{J},\boldsymbol{S_J},\boldsymbol{V}_s}_{s|(s-1)}(\boldsymbol{W}_s) \log \frac{P^{\boldsymbol{J},\boldsymbol{S_J},\boldsymbol{V}_s}_{s|(s-1)}(\boldsymbol{W}_s)}{\tilde{Q}^{\boldsymbol{J},\boldsymbol{S_J},\boldsymbol{V}_{[T]}}_{s|(s-1)}(\boldsymbol{W}_s)} \mathrm{d}\boldsymbol{W}_s - \log \left( \int e^{\mathbb{E}_{\boldsymbol{S}_{\boldsymbol{J}^c} \sim \mathcal{D}} \log Q^{\boldsymbol{S},\boldsymbol{V}_{[T]}}_{s|(s-1)}(\tilde{\boldsymbol{W}})} \mathrm{d}\tilde{\boldsymbol{W}} \right) \right)$$

$$= \arg\min_{P^{\boldsymbol{J},\boldsymbol{S_J},\boldsymbol{V}_s}_{s|(s-1)}} \left( \int P^{\boldsymbol{J},\boldsymbol{S_J},\boldsymbol{V}_s}_{s|(s-1)}(\boldsymbol{W}_s) \log \frac{P^{\boldsymbol{J},\boldsymbol{S_J},\boldsymbol{V}_s}_{s|(s-1)}(\boldsymbol{W}_s)}{\tilde{Q}^{\boldsymbol{J},\boldsymbol{S_J},\boldsymbol{V}_{[T]}}_{s|(s-1)}(\boldsymbol{W}_s)} \mathrm{d}\boldsymbol{W}_s \right)$$

$$= \arg\min_{P^{\boldsymbol{J},\boldsymbol{S_J},\boldsymbol{V}_s}_{s|(s-1)}} \operatorname{KL}\left(P \| \tilde{Q}^{\boldsymbol{J},\boldsymbol{S_J},\boldsymbol{V}_{[T]}}_{s|(s-1)}\right). \tag{31}$$

The minimum of Eq.(31) is achieved if and only if $P^{\boldsymbol{J},\boldsymbol{S_J},\boldsymbol{V}_s}_{s|(s-1)} = \tilde{Q}^{\boldsymbol{J},\boldsymbol{S_J},\boldsymbol{V}_{[T]}}_{s|(s-1)}$, and we only need to calculate the exact form of $\tilde{Q}^{\boldsymbol{J},\boldsymbol{S_J},\boldsymbol{V}_{[T]}}_{s|(s-1)}$. Since $\boldsymbol{W}_s|(\boldsymbol{W}_{s-1}, \boldsymbol{S}, \boldsymbol{V}_s) \sim \mathcal{N}(\boldsymbol{W}_{s-1} - \eta_s \nabla_{\boldsymbol{W}_{s-1}} \mathcal{R}_{\boldsymbol{S}_{\boldsymbol{V}_s}}(\boldsymbol{W}_{s-1}), \boldsymbol{\Sigma}_s(\boldsymbol{W}_{s-1}))$, we have

$$\exp \mathbb{E}_{\boldsymbol{S}_{\boldsymbol{J}^c} \sim \mathcal{D}} \log Q^{\boldsymbol{S},\boldsymbol{V}_{[T]}}_{s|(s-1)}(\boldsymbol{W})$$

$$= \exp\left( \mathbb{E}_{\boldsymbol{S}_{\boldsymbol{J}^c} \sim \mathcal{D}}\left( -\frac{1}{2}(\boldsymbol{W} - \boldsymbol{W}_{s-1} + \eta_s \nabla_{\boldsymbol{W}_{s-1}} \mathcal{R}_{\boldsymbol{S}_{\boldsymbol{V}_s}}(\boldsymbol{W}_{s-1}))^\top \boldsymbol{\Sigma}_s(\boldsymbol{W}_{s-1})^{-1}(\boldsymbol{W} - \boldsymbol{W}_{s-1}\right.\right.$$

$$\left.\left. + \eta_s \nabla_{\boldsymbol{W}_{s-1}} \mathcal{R}_{\boldsymbol{S}_{\boldsymbol{V}_s}}(\boldsymbol{W}_{s-1})) - \frac{d}{2}\log 2\pi - \frac{1}{2}\log\det(\boldsymbol{\Sigma}_s(\boldsymbol{W}_{s-1})) \right)\right)$$

$$= \exp\left( \mathbb{E}_{\boldsymbol{S}_{\boldsymbol{J}^c} \sim \mathcal{D}}\left( -\frac{1}{2}(\boldsymbol{W} - \boldsymbol{W}_{s-1} + \eta_s \nabla_{\boldsymbol{W}_{s-1}} \mathcal{R}_{\boldsymbol{S}_{\boldsymbol{V}_s}}(\boldsymbol{W}_{s-1}))^\top \boldsymbol{\Sigma}_s(\boldsymbol{W}_{s-1})^{-1}(\boldsymbol{W} - \boldsymbol{W}_{s-1}\right.\right.$$

$$\left.\left. + \eta_s \nabla_{\boldsymbol{W}_{s-1}} \mathcal{R}_{\boldsymbol{S}_{\boldsymbol{V}_s}}(\boldsymbol{W}_{s-1}) \right) - \frac{d}{2}\log 2\pi - \frac{1}{2}\log\det(\boldsymbol{\Sigma}_s(\boldsymbol{W}_{s-1})) \right). \tag{32}$$

On the other hand,

$$
\mathbb{E}_{\boldsymbol{S}_{\boldsymbol{J}^c} \sim \mathcal{D}} \left( -\frac{1}{2} (\boldsymbol{W} - \boldsymbol{W}_{s-1} + \eta_s \nabla_{\boldsymbol{W}_{s-1}} \mathcal{R}_{\boldsymbol{S}_{\boldsymbol{V}_s}} (\boldsymbol{W}_{s-1}))^\top \boldsymbol{\Sigma}_s (\boldsymbol{W}_{s-1})^{-1} (\boldsymbol{W} - \boldsymbol{W}_{s-1} \right.
$$
$$
\left. + \eta_s \nabla_{\boldsymbol{W}_{s-1}} \mathcal{R}_{\boldsymbol{S}_{\boldsymbol{V}_s}} (\boldsymbol{W}_{s-1})) \right)
$$
$$
= -\frac{1}{2} \mathbb{E}_{\boldsymbol{S}_{\boldsymbol{J}^c} \sim \mathcal{D}} \left( \boldsymbol{W} - \boldsymbol{W}_{s-1} + \eta_s \left( \frac{|\boldsymbol{V}_s \cap \boldsymbol{J}|}{|\boldsymbol{V}_s|} \nabla \mathcal{R}_{\boldsymbol{S}_{\boldsymbol{V}_s \cap \boldsymbol{J}}} (\boldsymbol{W}_{s-1}) + \frac{|\boldsymbol{V}_s \cap \boldsymbol{J}^c|}{|\boldsymbol{V}_s|} \nabla \mathcal{R}_{\boldsymbol{S}_{\boldsymbol{V}_s \cap \boldsymbol{J}^c}} (\boldsymbol{W}_{s-1}) \right) \right)^\top
$$
$$
\cdot \boldsymbol{\Sigma}_s (\boldsymbol{W}_{s-1})^{-1} \left( \boldsymbol{W} - \boldsymbol{W}_{s-1} + \eta_s \left( \frac{|\boldsymbol{V}_s \cap \boldsymbol{J}|}{|\boldsymbol{V}_s|} \nabla \mathcal{R}_{\boldsymbol{S}_{\boldsymbol{V}_s \cap \boldsymbol{J}}} (\boldsymbol{W}_{s-1}) + \frac{|\boldsymbol{V}_s \cap \boldsymbol{J}^c|}{|\boldsymbol{V}_s|} \nabla \mathcal{R}_{\boldsymbol{S}_{\boldsymbol{V}_s \cap \boldsymbol{J}^c}} (\boldsymbol{W}_{s-1}) \right) \right)
$$
$$
= -\frac{1}{2} \left( \boldsymbol{W} - \boldsymbol{W}_{s-1} + \eta_s \left( \frac{|\boldsymbol{V}_s \cap \boldsymbol{J}|}{|\boldsymbol{V}_s|} \nabla \mathcal{R}_{\boldsymbol{S}_{\boldsymbol{V}_s \cap \boldsymbol{J}}} (\boldsymbol{W}_{s-1}) + \frac{|\boldsymbol{V}_s \cap \boldsymbol{J}^c|}{|\boldsymbol{V}_s|} \nabla \mathcal{R}_{\mathcal{D}} (\boldsymbol{W}_{s-1}) \right) \right)^\top
$$
$$
\cdot \boldsymbol{\Sigma}_s (\boldsymbol{W}_{s-1})^{-1} \left( \boldsymbol{W} - \boldsymbol{W}_{s-1} + \eta_s \left( \frac{|\boldsymbol{V}_s \cap \boldsymbol{J}|}{|\boldsymbol{V}_s|} \nabla \mathcal{R}_{\boldsymbol{S}_{\boldsymbol{V}_s \cap \boldsymbol{J}}} (\boldsymbol{W}_{s-1}) + \frac{|\boldsymbol{V}_s \cap \boldsymbol{J}^c|}{|\boldsymbol{V}_s|} \nabla \mathcal{R}_{\mathcal{D}} (\boldsymbol{W}_{s-1}) \right) \right)
$$
$$
- \frac{1}{2} \mathbb{E}_{\boldsymbol{S}_{\boldsymbol{J}^c} \sim \mathcal{D}} \eta_s^2 \frac{|\boldsymbol{V}_s \cap \boldsymbol{J}^c|^2}{|\boldsymbol{V}_s|^2} \left( \nabla \mathcal{R}_{\mathcal{D}} (\boldsymbol{W}_{s-1}) - \nabla \mathcal{R}_{\boldsymbol{S}_{\boldsymbol{V}_s \cap \boldsymbol{J}^c}} (\boldsymbol{W}_{s-1}) \right)^\top \boldsymbol{\Sigma}_s (\boldsymbol{W}_{s-1})^{-1}
$$
$$
\cdot \left( \nabla \mathcal{R}_{\mathcal{D}} (\boldsymbol{W}_{s-1}) - \nabla \mathcal{R}_{\boldsymbol{S}_{\boldsymbol{V}_s \cap \boldsymbol{J}^c}} (\boldsymbol{W}_{s-1}) \right). \tag{33}
$$

By combining Eq.(32) and Eq.(33), we further have

$$
\exp \mathbb{E}_{\boldsymbol{S}_{\boldsymbol{J}^c} \sim \mathcal{D}} \log Q_{s|(s-1)}^{\boldsymbol{S}, \boldsymbol{V}_{[T]}} (\boldsymbol{W})
$$
$$
= \frac{1}{(2\pi)^{-\frac{d}{2}} \det(\boldsymbol{\Sigma}_s (\boldsymbol{W}_{s-1}))^{\frac{1}{2}}} \exp \left( -\frac{1}{2} \left( \boldsymbol{W} - \boldsymbol{W}_{s-1} + \eta_s \left( \frac{|\boldsymbol{V}_s \cap \boldsymbol{J}|}{|\boldsymbol{V}_s|} \nabla \mathcal{R}_{\boldsymbol{S}_{\boldsymbol{V}_s \cap \boldsymbol{J}}} (\boldsymbol{W}_{s-1}) \right. \right. \right.
$$
$$
\left. + \frac{|\boldsymbol{V}_s \cap \boldsymbol{J}^c|}{|\boldsymbol{V}_s|} \nabla \mathcal{R}_{\mathcal{D}} (\boldsymbol{W}_{s-1}) \right) \right)^\top \boldsymbol{\Sigma}_s (\boldsymbol{W}_{s-1})^{-1} \left( \boldsymbol{W} - \boldsymbol{W}_{s-1} + \eta_s \left( \frac{|\boldsymbol{V}_s \cap \boldsymbol{J}|}{|\boldsymbol{V}_s|} \nabla \mathcal{R}_{\boldsymbol{S}_{\boldsymbol{V}_s \cap \boldsymbol{J}}} (\boldsymbol{W}_{s-1}) \right. \right.
$$
$$
\left. \left. + \frac{|\boldsymbol{V}_s \cap \boldsymbol{J}^c|}{|\boldsymbol{V}_s|} \nabla \mathcal{R}_{\mathcal{D}} (\boldsymbol{W}_{s-1}) \right) \right) \exp \mathbb{E}_{\boldsymbol{S}_{\boldsymbol{J}^c}} \left( -\frac{1}{2} \eta_s^2 \frac{|\boldsymbol{V}_s \cap \boldsymbol{J}^c|^2}{|\boldsymbol{V}_s|^2} \left( \nabla \mathcal{R}_{\mathcal{D}} (\boldsymbol{W}_{s-1}) - \nabla \mathcal{R}_{\boldsymbol{S}_{\boldsymbol{V}_s \cap \boldsymbol{J}^c}} (\boldsymbol{W}_{s-1}) \right)^\top \right.
$$
$$
\left. \cdot \boldsymbol{\Sigma}_s (\boldsymbol{W}_{s-1})^{-1} \left( \nabla \mathcal{R}_{\mathcal{D}} (\boldsymbol{W}_{s-1}) - \nabla \mathcal{R}_{\boldsymbol{S}_{\boldsymbol{V}_s \cap \boldsymbol{J}^c}} (\boldsymbol{W}_{s-1}) \right) \right). \tag{34}
$$

Therefore, by taking integration with respect to $\tilde{\boldsymbol{W}}$, we have,

$$
\int e^{\mathbb{E}_{\boldsymbol{S}_{\boldsymbol{J}^c} \sim \mathcal{D}} \log Q_{s|(s-1)}^{\boldsymbol{S}, \boldsymbol{V}_{[T]}} (\tilde{\boldsymbol{W}})} \mathrm{d} \tilde{\boldsymbol{W}}
$$
$$
= \exp \mathbb{E}_{\boldsymbol{S}_{\boldsymbol{J}^c}} \left( -\frac{1}{2} \eta_s^2 \frac{|\boldsymbol{V}_s \cap \boldsymbol{J}^c|^2}{|\boldsymbol{V}_s|^2} \left( \nabla \mathcal{R}_{\mathcal{D}} (\boldsymbol{W}_{s-1}) - \nabla \mathcal{R}_{\boldsymbol{S}_{\boldsymbol{V}_s \cap \boldsymbol{J}^c}} (\boldsymbol{W}_{s-1}) \right)^\top \boldsymbol{\Sigma}_s (\boldsymbol{W}_{s-1})^{-1} \right.
$$
$$
\left. \cdot \left( \nabla \mathcal{R}_{\mathcal{D}} (\boldsymbol{W}_{s-1}) - \nabla \mathcal{R}_{\boldsymbol{S}_{\boldsymbol{V}_s \cap \boldsymbol{J}^c}} (\boldsymbol{W}_{s-1}) \right) \right). \tag{35}
$$

Therefore, by Eq.(30), Eq.(34), and Eq.(35), we have

$$
\arg \min_{P_{s|(s-1)}^{\boldsymbol{J}, \boldsymbol{S}_{\boldsymbol{J}}, \boldsymbol{V}_s}} \mathbb{E}_{\boldsymbol{S}_{\boldsymbol{J}^c} \sim \mathcal{D}} \mathrm{KL} \left( P_{s|(s-1)}^{\boldsymbol{J}, \boldsymbol{S}_{\boldsymbol{J}}, \boldsymbol{V}_s} \left\| Q_{s|(s-1)}^{\boldsymbol{S}, \boldsymbol{V}_{[T]}} \right) = \tilde{Q}_{s|(s-1)}^{\boldsymbol{J}, \boldsymbol{S}_{\boldsymbol{J}}, \boldsymbol{V}_{[T]}} \tag{36}
$$
$$
\sim \mathcal{N} \left( \boldsymbol{W}_{s-1} - \eta_s \left( \frac{|\boldsymbol{V}_s \cap \boldsymbol{J}|}{|\boldsymbol{V}_s|} \nabla \mathcal{R}_{\boldsymbol{S}_{\boldsymbol{V}_s \cap \boldsymbol{J}}} (\boldsymbol{W}_{s-1}) + \frac{|\boldsymbol{V}_s \cap \boldsymbol{J}^c|}{|\boldsymbol{V}_s|} \nabla \mathcal{R}_{\mathcal{D}} (\boldsymbol{W}_{s-1}) \right), \boldsymbol{\Sigma}_s (\boldsymbol{W}_{s-1}) \right).
$$

Applying Eq. (36) back to $\mathbb{E}_{\boldsymbol{S}_{\boldsymbol{J}^c}\sim\mathcal{D}}\,\mathrm{KL}\left(P^{\boldsymbol{J},\boldsymbol{S}_{\boldsymbol{J}},\boldsymbol{V}_s}_{s|(s-1)}\,\middle\|\,Q^{\boldsymbol{S},\boldsymbol{V}_{[T]}}_{s|(s-1)}\right)$, we obtain

$$\min_{P^{\boldsymbol{J},\boldsymbol{S}_{\boldsymbol{J}},\boldsymbol{V}_s}_{s|(s-1)}}\,\mathbb{E}_{\boldsymbol{S}_{\boldsymbol{J}^c}\sim\mathcal{D}}\,\mathrm{KL}\left(P^{\boldsymbol{J},\boldsymbol{S}_{\boldsymbol{J}},\boldsymbol{V}_s}_{s|(s-1)}\,\middle\|\,Q^{\boldsymbol{S},\boldsymbol{V}_{[T]}}_{s|(s-1)}\right)=\mathbb{E}_{\boldsymbol{S}_{\boldsymbol{J}^c}\sim\mathcal{D}}\,\mathrm{KL}\left(\tilde{Q}^{\boldsymbol{J},\boldsymbol{S}_{\boldsymbol{J}},\boldsymbol{V}_{[T]}}_{s|(s-1)}\,\middle\|\,Q^{\boldsymbol{S},\boldsymbol{V}_{[T]}}_{s|(s-1)}\right)$$

$$=\int\tilde{Q}^{\boldsymbol{S}_{\boldsymbol{J}},\boldsymbol{V}_{[s]}}_{t|(t-1)}(\boldsymbol{W}_s)\log\frac{\tilde{Q}^{\boldsymbol{S}_{\boldsymbol{J}},\boldsymbol{V}_{[s]}}_{t|(t-1)}(\boldsymbol{W}_s)}{e^{\mathbb{E}_{\boldsymbol{S}_{\boldsymbol{J}^c}\sim\mathcal{D}}\log Q^{\boldsymbol{S},\boldsymbol{V}_{[s]}}_{t|(t-1)}(\boldsymbol{W}_s)}}\mathrm{d}\boldsymbol{W}_s$$

$$\overset{(\circ)}{=}-\int\tilde{Q}^{\boldsymbol{S}_{\boldsymbol{J}},\boldsymbol{V}_{[s]}}_{t|(t-1)}(\boldsymbol{W}_s)\log\int e^{\mathbb{E}_{\boldsymbol{S}_{\boldsymbol{J}^c}\sim\mathcal{D}}\log Q^{\boldsymbol{S},\boldsymbol{V}_{[s]}}_{t|(t-1)}(\tilde{\boldsymbol{W}})}\mathrm{d}\tilde{\boldsymbol{W}}\mathrm{d}\boldsymbol{W}_s$$

$$=-\log\int e^{\mathbb{E}_{\boldsymbol{S}_{\boldsymbol{J}^c}\sim\mathcal{D}}\log Q^{\boldsymbol{S},\boldsymbol{V}_{[s]}}_{t|(t-1)}(\tilde{\boldsymbol{W}})}\mathrm{d}\tilde{\boldsymbol{W}}$$

$$\overset{(\bullet)}{=}\frac{1}{2}\eta_t^2\mathbb{E}_{\boldsymbol{S}_{\boldsymbol{J}^c}\sim\mathcal{D}}\frac{|\boldsymbol{V}_t\cap\boldsymbol{J}^c|^2}{|\boldsymbol{V}_t|^2}\left(\nabla\mathcal{R}_{\mathcal{D}}(\boldsymbol{W}_{t-1})-\nabla\mathcal{R}_{\boldsymbol{S}_{\boldsymbol{V}_t\cap\boldsymbol{J}^c}}(\boldsymbol{W}_{t-1})\right)^{\top}\boldsymbol{\Sigma}_t(\boldsymbol{W}_{t-1})^{-1}$$

$$\cdot\left(\nabla\mathcal{R}_{\mathcal{D}}(\boldsymbol{W}_{t-1})-\nabla\mathcal{R}_{\boldsymbol{S}_{\boldsymbol{V}_t\cap\boldsymbol{J}^c}}(\boldsymbol{W}_{t-1})\right)$$

$$=\frac{1}{2}\eta_t^2\mathbb{E}_{\boldsymbol{S}_{\boldsymbol{J}^c}\sim\mathcal{D}}\,\mathrm{tr}\left(\boldsymbol{\Sigma}_t(\boldsymbol{W}_{t-1})^{-1}\frac{|\boldsymbol{V}_t\cap\boldsymbol{J}^c|^2}{|\boldsymbol{V}_t|^2}\left(\nabla\mathcal{R}_{\mathcal{D}}(\boldsymbol{W}_{t-1})-\nabla\mathcal{R}_{\boldsymbol{S}_{\boldsymbol{V}_t\cap\boldsymbol{J}^c}}(\boldsymbol{W}_{t-1})\right)^{\top}\right.$$

$$\left.\cdot\left(\nabla\mathcal{R}_{\mathcal{D}}(\boldsymbol{W}_{t-1})-\nabla\mathcal{R}_{\boldsymbol{S}_{\boldsymbol{V}_t\cap\boldsymbol{J}^c}}(\boldsymbol{W}_{t-1})\right)\right)$$

$$\overset{(\Diamond)}{=}\begin{cases}0 & ,\boldsymbol{V}_t\cap\boldsymbol{J}^c=\emptyset;\\\dfrac{1}{2}\dfrac{\eta_t^2 N}{b_t(N-1)^2}\,\mathrm{tr}\left(\boldsymbol{\Sigma}_t(\boldsymbol{W}_{t-1})^{-1}\boldsymbol{\Sigma}^{pop}_{\boldsymbol{W}_{t-1}}\right), & \boldsymbol{V}_t\cap\boldsymbol{J}^c\neq\emptyset.\end{cases}\tag{37}$$

where Eq. ($\circ$) is due to the definition of $\tilde{Q}^{\boldsymbol{S}_{\boldsymbol{J}},\boldsymbol{V}_{[s]}}_{t|(t-1)}$ (Eq.(30)), Eq. ($\bullet$) is due to Eq.(35) and Eq. ($\Diamond$) is due to Lemma 6.

Therefore, when $\boldsymbol{V}_t\cap\boldsymbol{J}^c=\emptyset$, Eq.(28) is independent of $\boldsymbol{\Sigma}_s$. On the other hand, if $\boldsymbol{V}_t\cap\boldsymbol{J}^c\neq\emptyset$, we only need to solve

$$\boldsymbol{\Sigma}_s(\boldsymbol{W})^*=\arg\min_{\mathrm{tr}(\boldsymbol{\Sigma}_s(\boldsymbol{W}))=c_s(\boldsymbol{W})}\mathrm{tr}\left(\boldsymbol{\Sigma}_s(\boldsymbol{W})^{-1}\boldsymbol{\Sigma}^{pop}_{\boldsymbol{W}}\right),\text{ subject to Constraint 1.}\tag{38}$$

We complete the proof by solving Problem (38). Specifically, let the eigenvalues of $\boldsymbol{\Sigma}^{pop}_{\boldsymbol{W}}$ be $(\omega^{pop}_i)^d_{i=1}$ (the value is by non-increasing order with respect to index) we first fix the eigenvalues of $\boldsymbol{\Sigma}_s(\boldsymbol{W})$ to be $\alpha_{[d]}$ with $\alpha_i\geq 0$ (the value is by non-increasing order with respect to index), $i\in[d]$. Then, by Lemma 10, the minimum of $\mathrm{tr}\left(\boldsymbol{\Sigma}_s(\boldsymbol{W})^{-1}\boldsymbol{\Sigma}^{pop}_{\boldsymbol{W}}\right)$ is achieved when

$$\boldsymbol{\Sigma}_s(\boldsymbol{W})\in\left\{P^{\top}\left(\alpha_{[d]}\right)P:P\text{ is orthogonal and }\boldsymbol{\Sigma}^{pop}_{\boldsymbol{W}}=P^{\top}\left(\omega^{pop}_{[d]}\right)P\right\},\tag{39}$$

and

$$\mathrm{tr}\left(\boldsymbol{\Sigma}_s(\boldsymbol{W})^{-1}\boldsymbol{\Sigma}^{pop}_{\boldsymbol{W}}\right)=\sum_{i=1}^d\frac{\omega^{pop}_i}{\alpha_i}.$$

We then optimize $\sum_{i=1}^d\frac{\omega^{pop}_i}{\alpha_i}$ under the constraint $\sum_{i=1}^d\alpha_i=c_s(\boldsymbol{W}_{s-1})$. By the Cauchy-Schwarz inequality,

$$c_s(\boldsymbol{W}_{s-1})\left(\sum_{i=1}^d\frac{\omega^{pop}_i}{\alpha_i}\right)=\left(\sum_{i=1}^d\frac{\omega^{pop}_i}{\alpha_i}\right)\left(\sum_{i=1}^d\alpha_i\right)\overset{(*)}{\geq}\left(\sum_{i=1}^d\sqrt{\omega^{pop}_i}\right)^2,\tag{40}$$

where equality in inequality $(*)$ holds when $\alpha_i^2/\omega^{pop}_i$ is invariant of $i$. By combining Eq.(39) and Eq.(40), the proof is completed. $\square$

By Lemma 12, the optimal noise covariances $\boldsymbol{\Sigma}_s$ of all KL divergence terms $\mathbb{E}_{\boldsymbol{S}_{\boldsymbol{J}^c}\sim\mathcal{D}}\,\mathrm{KL}\left(P^{\boldsymbol{J},\boldsymbol{S}_{\boldsymbol{J}},\boldsymbol{V}_s}_{s|(s-1)}\,\middle\|\,Q^{\boldsymbol{S},\boldsymbol{V}_s}_{s|(s-1)}\right)$ are the same regardless of $\boldsymbol{V}_s$, $\boldsymbol{J}$, and $\boldsymbol{S}_{\boldsymbol{J}}$, which helps us to obtain Lemma 4.

*Proof of Lemma 4.* To begin with, denote the optimal noise covariance of first $s$-step in terms of the generalization bound $\text{Gen}_s$ as $\mathbf{\Sigma}^s$ under Constraint 1, i.e.,

$$\mathbf{\Sigma}^s_{[s]} \triangleq \arg\min_{\mathbf{\Sigma}_{[s]}} \left( \min_P \text{Gen}_s(P, \mathbf{\Sigma}_{[s]}) \right), \text{ subject to: Constraint 1,}$$

we also define $Q^s$ accordingly as the posterior distribution with noise covariance $\mathbf{\Sigma}^s$. Also, recall that $P^s$ is the optimal prior in terms of the generalization bound $\text{Gen}_s$ under Constraint 1, i.e.,

$$P^s = \arg\min_P \left( \min_{\mathbf{\Sigma}_{[s]}} \text{Gen}_s(P, \mathbf{\Sigma}_{[s]}) \right), \text{ subject to: Constraint 1.}$$

We would like to derive the form of $\mathbf{\Sigma}^s_s$ and $P^s_{s|(s-1)}$.

Specifically, we have

$$P^s_{s|(s-1)} = \arg\min_{P_{s|(s-1)}} \left( \text{Gen}_s(P, \mathbf{\Sigma}^s_{[s]}) \right), \text{ subject to: } P_{t|(t-1)} = P^s_{t|(t-1)} (t < s),$$

and

$$\mathbf{\Sigma}^s_s = \arg\min_{\mathbf{\Sigma}_s} \left( \text{Gen}_s(P^s, \mathbf{\Sigma}_{[s]}) \right), \text{ subject to: Constraint 1 and } \mathbf{\Sigma}_t = \mathbf{\Sigma}^s_{t|(t-1)} (t < s).$$

That is, to obtain the desired $\mathbf{\Sigma}^s_s$ and $P^s_{s|(s-1)}$, we only need to solve

$$\min_{\mathbf{\Sigma}_s, P^s_{s|(s-1)}} \text{Gen}_s(P, \mathbf{\Sigma}_{[s]}), \text{ subject to: } P_{t|(t-1)} = P^s_{t|(t-1)} (t < s) \text{ and } \mathbf{\Sigma}_t = \mathbf{\Sigma}^s_{t|(t-1)} (t < s).$$

On the other hand, with $P_{t|(t-1)} = P^s_{t|(t-1)} (t < s)$ and $\mathbf{\Sigma}_t = \mathbf{\Sigma}^s_{t|(t-1)} (t < s)$ and under Constraint 1, we have

$$\min_{\mathbf{\Sigma}_s, P_{s|(s-1)}} \text{Gen}_s(P, \mathbf{\Sigma}_{[s]})$$

$$= \min_{\mathbf{\Sigma}_s, P_{s|(s-1)}} \mathbb{E}_{\boldsymbol{S}_{\boldsymbol{J}}, \boldsymbol{V}_{[s]}, \boldsymbol{J}} \sqrt{\frac{(a_2 - a_1)^2}{2} \mathbb{E}_{\boldsymbol{S}_{\boldsymbol{J}^c}} \text{KL}\left( P^{\boldsymbol{J}, \boldsymbol{S}_{\boldsymbol{J}}, \boldsymbol{V}_{[s]}} \middle\| Q^{\boldsymbol{S}, \boldsymbol{V}_{[s]}} \right)}$$

$$= \min_{\mathbf{\Sigma}_s, P_{s|(s-1)}} \mathbb{E}_{\boldsymbol{S}_{\boldsymbol{J}}, \boldsymbol{V}_{[s]}, \boldsymbol{J}} \sqrt{\frac{(a_2 - a_1)^2}{2} \mathbb{E}_{\boldsymbol{S}_{\boldsymbol{J}^c}} \sum_{t=1}^s \mathbb{E}_{P^{\boldsymbol{J}, \boldsymbol{S}_{\boldsymbol{J}}, \boldsymbol{V}_{[s]}}_{t-1}} \text{KL}\left( P^{\boldsymbol{J}, \boldsymbol{S}_{\boldsymbol{J}}, \boldsymbol{V}_s}_{t|(t-1)} \middle\| Q^{\boldsymbol{S}, \boldsymbol{V}_s}_{t|(t-1)} \right)}$$

$$= \min_{\mathbf{\Sigma}_s, P_{s|(s-1)}} \mathbb{E}_{\boldsymbol{S}_{\boldsymbol{J}}, \boldsymbol{V}_{[s]}, \boldsymbol{J}} \left[ \sqrt{\frac{(a_2 - a_1)^2}{2} \mathbb{E}_{\boldsymbol{S}_{\boldsymbol{J}^c}} \sum_{t=1}^s \mathbb{E}_{P^s{}^{\boldsymbol{J}, \boldsymbol{S}_{\boldsymbol{J}}, \boldsymbol{V}_{[s]}}_{t-1}} \text{KL}\left( P^s{}^{\boldsymbol{J}, \boldsymbol{S}_{\boldsymbol{J}}, \boldsymbol{V}_s}_{t|(t-1)} \middle\| Q^s{}^{\boldsymbol{S}, \boldsymbol{V}_s}_{t|(t-1)} \right)} \right.$$

$$\left. + \frac{(a_2 - a_1)^2}{2} \mathbb{E}_{\boldsymbol{S}_{\boldsymbol{J}^c}} \mathbb{E}_{P^{\boldsymbol{J}, \boldsymbol{S}_{\boldsymbol{J}}, \boldsymbol{V}_{[s]}}_{s-1}} \text{KL}\left( P^{\boldsymbol{J}, \boldsymbol{S}_{\boldsymbol{J}}, \boldsymbol{V}_s}_{s|(s-1)} \middle\| Q^{\boldsymbol{S}, \boldsymbol{V}_s}_{s|(s-1)} \right) \right]$$

$$\overset{(*)}{\geq} \mathbb{E}_{\boldsymbol{S}_{\boldsymbol{J}}, \boldsymbol{V}_{[s]}, \boldsymbol{J}} \left[ \sqrt{\frac{(a_2 - a_1)^2}{2} \mathbb{E}_{\boldsymbol{S}_{\boldsymbol{J}^c}} \sum_{t=1}^s \mathbb{E}_{P^s{}^{\boldsymbol{J}, \boldsymbol{S}_{\boldsymbol{J}}, \boldsymbol{V}_s}_{t-1}} \text{KL}\left( P^s{}^{\boldsymbol{J}, \boldsymbol{S}_{\boldsymbol{J}}, \boldsymbol{V}_s}_{t|(t-1)} \middle\| Q^s{}^{\boldsymbol{S}, \boldsymbol{V}_s}_{t|(t-1)} \right)} \right.$$

$$\left. + \frac{(a_2 - a_1)^2}{2} \mathbb{E}_{\boldsymbol{S}_{\boldsymbol{J}^c}} \mathbb{E}_{P^{\boldsymbol{J}, \boldsymbol{S}_{\boldsymbol{J}}, \boldsymbol{V}_{[s]}}_{s-1}} \min_{\mathbf{\Sigma}_s, P^{\boldsymbol{J}, \boldsymbol{S}_{\boldsymbol{J}}, \boldsymbol{V}_s}_{s|(s-1)}} \text{KL}\left( P^{\boldsymbol{J}, \boldsymbol{S}_{\boldsymbol{J}}, \boldsymbol{V}_s}_{s|(s-1)} \middle\| Q^{\boldsymbol{S}, \boldsymbol{V}_s}_{s|(s-1)} \right) \right].$$

By Lemma 12, $\min_{\mathbf{\Sigma}_s, P^{\boldsymbol{J}, \boldsymbol{S}_{\boldsymbol{J}}, \boldsymbol{V}_s}_{s|(s-1)}} \text{KL}\left( P^{\boldsymbol{J}, \boldsymbol{S}_{\boldsymbol{J}}, \boldsymbol{V}_s}_{s|(s-1)} \middle\| Q^{\boldsymbol{S}, \boldsymbol{V}_s}_{s|(s-1)} \right)$ is attained at $\mathbf{\Sigma}_s(\boldsymbol{W}) = \lambda_s(\boldsymbol{W}) (\mathbf{\Sigma}^{pop}_{\boldsymbol{W}})^{\frac{1}{2}}$, which is not dependent on $\boldsymbol{J}, \boldsymbol{S}_{\boldsymbol{J}}, \boldsymbol{V}_s$, and

$$P^{\boldsymbol{J}, \boldsymbol{S}_{\boldsymbol{J}}, \boldsymbol{V}_s}_{s|(s-1)} \sim \mathcal{N}\left( \boldsymbol{W}_{s-1} - \eta_s \left( \frac{|\boldsymbol{V}_s \cap \boldsymbol{J}|}{|\boldsymbol{V}_s|} \nabla \mathcal{R}_{\boldsymbol{S}_{\boldsymbol{V}_s} \cap \boldsymbol{J}}(\boldsymbol{W}_{s-1}) + \frac{|\boldsymbol{V}_s \cap \boldsymbol{J}^c|}{|\boldsymbol{V}_s|} \nabla \mathcal{R}_{\mathcal{D}}(\boldsymbol{W}_{s-1}) \right), \lambda_s(\boldsymbol{W}) (\mathbf{\Sigma}^{pop}_{\boldsymbol{W}})^{\frac{1}{2}} \right).$$

Therefore, Inequality $(*)$ holds, and the proof is completed.

$\square$

By Lemma 4, we obtain the form of $P^*$, i.e.,

$$P^{*\boldsymbol{J},\boldsymbol{S_J},\boldsymbol{V_s}}_{s|(s-1)} \sim \mathcal{N}\left(\boldsymbol{W}_{s-1} - \eta_s \left(\frac{|\boldsymbol{V_s} \cap \boldsymbol{J}|}{|\boldsymbol{V_s}|}\nabla\mathcal{R}_{\boldsymbol{S}_{\boldsymbol{V_s} \cap \boldsymbol{J}}}(\boldsymbol{W}_{s-1}) + \frac{|\boldsymbol{V_s} \cap \boldsymbol{J}^c|}{|\boldsymbol{V_s}|}\nabla\mathcal{R}_{\mathcal{D}}(\boldsymbol{W}_{s-1})\right), \lambda_s(\boldsymbol{W})(\boldsymbol{\Sigma}_{\boldsymbol{W}}^{pop})^{\frac{1}{2}}\right),$$

which allows us to further derive Theorem 2.

*Proof of Theorem 2.* By the definition of $\mathrm{Gen}_T$, with prior the greedy prior and under Constraint 1, we have

$$\min_{\boldsymbol{\Sigma}_{[T]}} \mathrm{Gen}_T(P^*, \boldsymbol{\Sigma}_{[T]})$$

$$= \min_{\boldsymbol{\Sigma}_{[T]}} \mathbb{E}_{\boldsymbol{S_J},\boldsymbol{V}_{[T]},\boldsymbol{J}} \sqrt{\frac{(a_2-a_1)^2}{2}\mathbb{E}_{\boldsymbol{S}_{\boldsymbol{J}^c}}\mathrm{KL}\left(P^{*\boldsymbol{J},\boldsymbol{S_J},\boldsymbol{V}_{[T]}}\left\|Q^{\boldsymbol{S},\boldsymbol{V}_{[T]}}\right.\right)}$$

$$= \min_{\boldsymbol{\Sigma}_{[T]}} \mathbb{E}_{\boldsymbol{S_J},\boldsymbol{V}_{[T]},\boldsymbol{J}} \sqrt{\frac{(a_2-a_1)^2}{2}\mathbb{E}_{\boldsymbol{S}_{\boldsymbol{J}^c}}\sum_{t=1}^{T}\mathbb{E}_{P^{*\boldsymbol{J},\boldsymbol{S_J},\boldsymbol{V}_{[t-1]}}_{t-1}}\mathrm{KL}\left(P^{*\boldsymbol{J},\boldsymbol{S_J},\boldsymbol{V}_t}_{t|(t-1)}\left\|Q^{\boldsymbol{S},\boldsymbol{V}_t}_{t|(t-1)}\right.\right)}$$

$$\overset{(\bullet)}{\geq} \mathbb{E}_{\boldsymbol{S_J},\boldsymbol{V}_{[T]},\boldsymbol{J}} \sqrt{\frac{(a_2-a_1)^2}{2}\sum_{t=1}^{T}\mathbb{E}_{P^{*\boldsymbol{J},\boldsymbol{S_J},\boldsymbol{V}_{[t-1]}}_{t-1}}\min_{\boldsymbol{\Sigma}_t}\mathbb{E}_{\boldsymbol{S}_{\boldsymbol{J}^c}}\mathrm{KL}\left(P^{*\boldsymbol{J},\boldsymbol{S_J},\boldsymbol{V}_t}_{t|(t-1)}\left\|Q^{\boldsymbol{S},\boldsymbol{V}_t}_{t|(t-1)}\right.\right)}$$

$$\overset{(*)}{=} \mathbb{E}_{\boldsymbol{S_J},\boldsymbol{V}_{[T]},\boldsymbol{J}} \sqrt{\frac{(a_2-a_1)^2}{2}\sum_{t=1}^{T}\mathbb{E}_{P^{*\boldsymbol{J},\boldsymbol{S_J},\boldsymbol{V}_{[t-1]}}_{t-1}}\min_{\boldsymbol{\Sigma}_t,P^{\boldsymbol{J},\boldsymbol{S_J},\boldsymbol{V}_t}_{t|(t-1)}}\mathbb{E}_{\boldsymbol{S}_{\boldsymbol{J}^c}}\mathrm{KL}\left(P^{\boldsymbol{J},\boldsymbol{S_J},\boldsymbol{V}_t}_{t|(t-1)}\left\|Q^{\boldsymbol{S},\boldsymbol{V}_t}_{t|(t-1)}\right.\right)},$$

where Eq. $(*)$ is due to that by the proof of Lemma 4, $P^{*\boldsymbol{J},\boldsymbol{S_J},\boldsymbol{V_s}}_{s|(s-1)}$ is the same as the prior minimizing $\mathbb{E}_{\boldsymbol{S}_{\boldsymbol{J}^c}\sim\mathcal{D}}\mathrm{KL}\left(P^{\boldsymbol{J},\boldsymbol{S_J},\boldsymbol{V_s}}_{s|(s-1)}\left\|Q^{\boldsymbol{S},\boldsymbol{V_s}}_{s|(s-1)}\right.\right)$ for any given $\boldsymbol{J},\boldsymbol{S_J},\boldsymbol{V_s}$. Therefore, $\min_{\boldsymbol{\Sigma}_t}\mathbb{E}_{\boldsymbol{S}_{\boldsymbol{J}^c}}\mathrm{KL}\left(P^{*\boldsymbol{J},\boldsymbol{S_J},\boldsymbol{V}_t}_{t|(t-1)}\left\|Q^{\boldsymbol{S},\boldsymbol{V}_t}_{t|(t-1)}\right.\right)$ is attained when $\boldsymbol{\Sigma}_t(\boldsymbol{W}) = \lambda_s(\boldsymbol{W})(\boldsymbol{\Sigma}_{\boldsymbol{W}}^{pop})^{\frac{1}{2}}$, which is independent of $\boldsymbol{J},\boldsymbol{S_J},\boldsymbol{V_s}$, and Inequality ($\bullet$) holds. Therefore, $\min_{\boldsymbol{\Sigma}_{[T]}}\mathrm{Gen}_T(P^*, \boldsymbol{\Sigma}_{[T]})$ is also attained at $\boldsymbol{\Sigma}_t(\boldsymbol{W}) = \lambda_s(\boldsymbol{W})(\boldsymbol{\Sigma}_{\boldsymbol{W}}^{pop})^{\frac{1}{2}}$.

The proof is completed.

$\square$

# E   Supplementary materials of Section 5

## E.1   Formal Description of the Prior in Section 5

In this section, we provide a detailed description of the update rule of the prior defined by Eq.(10).

---

**Algorithm 1:** Iteration of Prior

---

**Input:** Sample set $\boldsymbol{S}$ with size $N$, initialization distribution $\mathcal{W}_0$, total step $T$, learning rate $(\eta_t)_{t=1}^{T}$

**Output:** $\boldsymbol{W}_{[T]}, \boldsymbol{J}$

1 Initialize $\boldsymbol{W}_0$ according to $\mathcal{W}_0$; initialize $\boldsymbol{J}$ by uniformly sampling $N-1$ elements from $[N]$ without replacement; set $t = 0$

2 **while** $t < T$ **do**

3     Uniformly sample index set $\boldsymbol{V}_t \subset [N]$ such that $|\boldsymbol{V}_t| = b_t$ without replacement and independent of $\boldsymbol{J}$

4     **if** $\boldsymbol{V}_t \subset \boldsymbol{J}$ **then**

5        $\boldsymbol{W}_t = \boldsymbol{W}_{t-1} - \eta_t\nabla\mathcal{R}_{\boldsymbol{S}_{\boldsymbol{V}_t}}(\boldsymbol{W}_{t-1}) + \mathcal{N}(\boldsymbol{0}, \sigma_t\mathbb{I}_d)$

6     **else**

7        $\boldsymbol{W}_t = \boldsymbol{W}_{t-1} - \eta_t\frac{b_t-1}{b_t}\nabla\mathcal{R}_{\boldsymbol{S}_{\boldsymbol{V}_t \cap \boldsymbol{J}}}(\boldsymbol{W}_{t-1}) - \eta_t\frac{1}{b_t}\nabla\mathcal{R}_{\boldsymbol{S_J}}(\boldsymbol{W}_{t-1}) + \mathcal{N}(\boldsymbol{0}, \sigma_t\mathbb{I}_d)$

8     $t = t + 1$

---

## E.2 Calculation of the Generalization Bound

To obtain the optimal noise covariance of **(P2)**, we first derive the explicit form of the generalization bound $\widetilde{\text{Gen}}_T$ with the prior given by Eq. (10) as the following lemma:

**Lemma 13** (Calculate $\widetilde{\text{Gen}}_T$). *Let Assumption 1 hold. Let the prior $P$ is given by the update rule Eq. (10). Then, the generalization bound $\widetilde{\text{Gen}}_T$ can be represented as*

$$\widetilde{\text{Gen}}_T = \mathbb{E}_{\boldsymbol{S_J}} \sqrt{\frac{(a_2 - a_1)^2}{2} \sum_{t=1}^{T} \mathbb{E}_{\boldsymbol{S_{J^c}}, \boldsymbol{V}_{[t-1]}} \mathbb{E}_{P_{s-1}^{\boldsymbol{J}, \boldsymbol{S_J}, \boldsymbol{V}_{[s-1]}}} A_t(\boldsymbol{S}, \boldsymbol{W}_{t-1})},$$

*where $A_t(\boldsymbol{S}, \boldsymbol{W})$ is given as*

$$A_t(\boldsymbol{S}, \boldsymbol{W}) \triangleq \frac{1}{2} \left( \sigma_t(\boldsymbol{W}) \operatorname{tr} \left( \boldsymbol{\Sigma}_t(\boldsymbol{S}, \boldsymbol{W})^{-1} \right) + \ln \left( \det \boldsymbol{\Sigma}_t(\boldsymbol{S}, \boldsymbol{W}) \right) - d \right)$$
$$+ \frac{\eta_t^2}{2Nb_t} \left( \frac{N}{N-1} \right)^2 \operatorname{tr} \left( \boldsymbol{\Sigma}_t(\boldsymbol{S})^{-1} \boldsymbol{\Sigma}_{\boldsymbol{S}, \boldsymbol{W}_{t-1}}^{sd} \right) - \frac{1}{2} d \ln \sigma_t(\boldsymbol{W}).$$

*Proof.* By the definition of $\widetilde{\text{Gen}}_T$, we have

$$\widetilde{\text{Gen}}_T = \mathbb{E}_{\boldsymbol{S}} \sqrt{\frac{(a_2 - a_1)^2}{2} \mathbb{E}_{\boldsymbol{V}_{[T]}, \boldsymbol{J}} \operatorname{KL} \left( P^{\boldsymbol{J}, \boldsymbol{S_J}, \boldsymbol{V}_{[T]}} \middle\| Q^{\boldsymbol{S}, \boldsymbol{V}_{[T]}} \right)},$$

which by the decomposition of KL divergence (Lemma 1) further leads to

$$\widetilde{\text{Gen}}_T = \mathbb{E}_{\boldsymbol{S}} \sqrt{\frac{(a_2 - a_1)^2}{2} \mathbb{E}_{\boldsymbol{V}_{[T]}, \boldsymbol{J}} \sum_{s=1}^{T} \mathbb{E}_{P_{s-1}^{\boldsymbol{J}, \boldsymbol{S_J}, \boldsymbol{V}_{[T]}}} \operatorname{KL} \left( P_{s|(s-1)}^{\boldsymbol{J}, \boldsymbol{S_J}, \boldsymbol{V}_{[T]}} \middle\| Q_{s|(s-1)}^{\boldsymbol{S}, \boldsymbol{V}_{[T]}} \right)}$$

$$= \mathbb{E}_{\boldsymbol{S}} \sqrt{\frac{(a_2 - a_1)^2}{2} \sum_{s=1}^{T} \mathbb{E}_{\boldsymbol{V}_{[s]}, \boldsymbol{J}} \mathbb{E}_{P_{s-1}^{\boldsymbol{J}, \boldsymbol{S_J}, \boldsymbol{V}_{[s-1]}}} \operatorname{KL} \left( P_{s|(s-1)}^{\boldsymbol{J}, \boldsymbol{S_J}, \boldsymbol{V}_s} \middle\| Q_{s|(s-1)}^{\boldsymbol{S}, \boldsymbol{V}_s} \right)}$$

$$\overset{(*)}{=} \mathbb{E}_{\boldsymbol{S}} \sqrt{\frac{(a_2 - a_1)^2}{2} \sum_{s=1}^{T} \mathbb{E}_{\boldsymbol{V}_{[s-1]}} \mathbb{E}_{P_{s-1}^{\boldsymbol{J}, \boldsymbol{S_J}, \boldsymbol{V}_{[s-1]}}} \mathbb{E}_{\boldsymbol{V}_s, \boldsymbol{J}} \operatorname{KL} \left( P_{s|(s-1)}^{\boldsymbol{J}, \boldsymbol{S_J}, \boldsymbol{V}_s} \middle\| Q_{s|(s-1)}^{\boldsymbol{S}, \boldsymbol{V}_s} \right)}$$

$$\overset{(**)}{=} \mathbb{E}_{\boldsymbol{S}} \sqrt{\frac{(a_2 - a_1)^2}{2} \sum_{s=1}^{T} \mathbb{E}_{\boldsymbol{V}_{[s-1]}, \boldsymbol{J}} \mathbb{E}_{P_{s-1}^{\boldsymbol{J}, \boldsymbol{S_J}, \boldsymbol{V}_{[s-1]}}} \mathbb{E}_{\boldsymbol{V}_s, \boldsymbol{J}} \operatorname{KL} \left( P_{s|(s-1)}^{\boldsymbol{J}, \boldsymbol{S_J}, \boldsymbol{V}_s} \middle\| Q_{s|(s-1)}^{\boldsymbol{S}, \boldsymbol{V}_s} \right)},$$

where in Eq. $(*)$ we exchange the order between $\mathbb{E}_{P_{s-1}^{\boldsymbol{J}, \boldsymbol{S_J}, \boldsymbol{V}_{[s-1]}}}$ and $\mathbb{E}_{\boldsymbol{V}_s, \boldsymbol{J}}$ due to Assumption 1, and Eq. $(**)$ is due to that $\mathbb{E}_{P_{s-1}^{\boldsymbol{J}, \boldsymbol{S_J}, \boldsymbol{V}_{[s-1]}}} \mathbb{E}_{\boldsymbol{V}_s, \boldsymbol{J}} \operatorname{KL} \left( P_{s|(s-1)}^{\boldsymbol{J}, \boldsymbol{S_J}, \boldsymbol{V}_s} \middle\| Q_{s|(s-1)}^{\boldsymbol{S}, \boldsymbol{V}_s} \right)$ is independent of $\boldsymbol{J}$ by Assumption 1.

Therefore, we only need to prove $\mathbb{E}_{\boldsymbol{V}_t, \boldsymbol{J}} \operatorname{KL} \left( P_{t|(t-1)}^{\boldsymbol{J}, \boldsymbol{S_J}, \boldsymbol{V}_t} \middle\| Q_{t|(t-1)}^{\boldsymbol{S}, \boldsymbol{V}_t} \right) = A(t)$, which can be obtained by

$$\mathbb{E}_{\boldsymbol{V}_t,\boldsymbol{J}} \, \mathrm{KL} \left( P_{t|(t-1)}^{\boldsymbol{J},\boldsymbol{S}_{\boldsymbol{J}},\boldsymbol{V}_t} \, \Big\| \, Q_{t|(t-1)}^{\boldsymbol{S},\boldsymbol{V}_t} \right)$$

$$\overset{(\bullet)}{=} \frac{1}{2} \mathbb{E}_{\boldsymbol{V}_t,\boldsymbol{J}} \bigg( \left( \mu^{\boldsymbol{S},\boldsymbol{V}_t,\boldsymbol{J},\boldsymbol{W}_{t-1}} \right)^{\top} \boldsymbol{\Sigma}_t(\boldsymbol{S},\boldsymbol{W}_{t-1})^{-1} \mu^{\boldsymbol{S},\boldsymbol{V}_t,\boldsymbol{J},\boldsymbol{W}_{t-1}} + \ln \frac{\det \boldsymbol{\Sigma}_t(\boldsymbol{S},\boldsymbol{W}_{t-1})}{\sigma_t(\boldsymbol{W}_{t-1})^d}$$

$$+ \, \mathrm{tr} \left( \sigma_t(\boldsymbol{W}_{t-1}) \boldsymbol{\Sigma}_t(\boldsymbol{S},\boldsymbol{W}_{t-1})^{-1} \right) \bigg) - \frac{d}{2}$$

$$= \frac{1}{2} \mathbb{E}_{\boldsymbol{V}_t,\boldsymbol{J}} \bigg( \mathrm{tr} \left( \boldsymbol{\Sigma}_t(\boldsymbol{S},\boldsymbol{W}_{t-1})^{-1} \mu^{\boldsymbol{S},\boldsymbol{V}_t,\boldsymbol{J},\boldsymbol{W}_{t-1}} \left( \mu^{\boldsymbol{S},\boldsymbol{V}_t,\boldsymbol{J},\boldsymbol{W}_{t-1}} \right)^{\top} \right) + \ln \frac{\det \boldsymbol{\Sigma}_t(\boldsymbol{S},\boldsymbol{W}_{t-1})}{\sigma_t(\boldsymbol{W}_{t-1})^d}$$

$$+ \, \mathrm{tr} \left( \sigma_t(\boldsymbol{W}_{t-1}) \boldsymbol{\Sigma}_t(\boldsymbol{S},\boldsymbol{W}_{t-1})^{-1} \right) \bigg) - \frac{d}{2}$$

$$= \frac{1}{2} \mathrm{tr} \left( \boldsymbol{\Sigma}_t(\boldsymbol{S},\boldsymbol{W}_{t-1})^{-1} \mathbb{E}_{\boldsymbol{J},\boldsymbol{V}_t} \mu^{\boldsymbol{S},\boldsymbol{V}_t,\boldsymbol{J},\boldsymbol{W}_{t-1}} \left( \mu^{\boldsymbol{S},\boldsymbol{V}_t,\boldsymbol{J},\boldsymbol{W}_{t-1}} \right)^{\top} \right) + \frac{1}{2} \ln \frac{\det \boldsymbol{\Sigma}_t(\boldsymbol{S},\boldsymbol{W}_{t-1})}{\sigma_t(\boldsymbol{W}_{t-1})^d}$$

$$+ \frac{1}{2} \mathrm{tr} \left( \sigma_t(\boldsymbol{W}_{t-1}) \boldsymbol{\Sigma}_t(\boldsymbol{S},\boldsymbol{W}_{t-1})^{-1} \right) - \frac{d}{2}$$

$$\overset{(\circ)}{=} \frac{1}{2} \left( \sigma_t(\boldsymbol{W}_{t-1}) \, \mathrm{tr} \left( \boldsymbol{\Sigma}_t(\boldsymbol{S},\boldsymbol{W}_{t-1})^{-1} \right) + \ln \left( \det \boldsymbol{\Sigma}_t(\boldsymbol{S},\boldsymbol{W}_{t-1}) \right) - d \right) - \frac{1}{2} d \ln \sigma_t(\boldsymbol{W}_{t-1})$$

$$+ \frac{\eta_t^2}{2 N b_t} \left( \frac{N}{N-1} \right)^2 \mathrm{tr} \left( \boldsymbol{\Sigma}_t(\boldsymbol{S},\boldsymbol{W}_{t-1})^{-1} \boldsymbol{\Sigma}_{\boldsymbol{S},\boldsymbol{W}_{t-1}}^{sd} \right),$$

where Eq. ($\bullet$) is due to Lemma 5, where $\mu^{\boldsymbol{S},\boldsymbol{V}_t,\boldsymbol{J},\boldsymbol{W}_{t-1}}$ is defined by Eq.(27), and Eq. ($\circ$) is obtained by Lemma 6.

The proof is completed. $\qquad\square$

By Lemma 13, for any $t \in [T]$, $\boldsymbol{S}$, and $\boldsymbol{W}_{t-1}$, $\mathrm{Gen}_{[T]}$ depend on $\boldsymbol{\Sigma}_t(\boldsymbol{W}, \mathrm{Gen}_{[T]})$ only through $A_t(\boldsymbol{S},\boldsymbol{W})$, and the solution of optimizing $A_t$ with respect to $\boldsymbol{\Sigma}_t$ under Constraint 1 has already been given by Lemma 9. We then complete the proof of Theorem 3 in the next section by combining Lemma 13 and Lemma 9 together.

### E.3 Proof of Theorem 3

In this section, we first restate Theorem 3 with explicit form of $\tilde{\omega}_i^{\boldsymbol{S},\boldsymbol{W}}$ (omitted in the main text). We then provide the proof of the theorem by Lemma 13 and Lemma 9.

**Theorem 4.** *Let prior and posterior be defined as Eq.(10) and Eq.(3), respectively. Then, with Assumption 1, the solution of (**P2**) is given by*

$$\boldsymbol{\Sigma}_t^*(\boldsymbol{S},\boldsymbol{W}) = \boldsymbol{Q}_{\boldsymbol{S},\boldsymbol{W}}^{sd} \, \mathrm{Diag}(\tilde{\omega}_{t,1}^{\boldsymbol{S},\boldsymbol{W}}, \cdots, \tilde{\omega}_{t,d}^{\boldsymbol{S},\boldsymbol{W}}) \left( \boldsymbol{Q}_{\boldsymbol{S},\boldsymbol{W}}^{sd} \right)^{\top},$$

*where*

$$\tilde{\omega}_{t,i}^{\boldsymbol{S},\boldsymbol{W}} = \frac{\sqrt{1 - 4\lambda^* \left( \mathbb{E}_{\boldsymbol{J}} \sigma_t(\boldsymbol{S}_{\boldsymbol{J}},\boldsymbol{W}) + \frac{\eta_t^2}{N b_t} \left( \frac{N}{N-1} \right)^2 \omega_i^{\boldsymbol{S},\boldsymbol{W}} \right)} - 1}{-2\lambda^*},$$

*$\lambda^*$ is determined by $\sum_{i=1}^d \tilde{\omega}_{t,i}^{\boldsymbol{S},\boldsymbol{W}} = c_t(\boldsymbol{S},\boldsymbol{W})$, and $\boldsymbol{Q}_{\boldsymbol{S},\boldsymbol{W}}^{sd}$ is the orthogonal matrix that diagonalizes $\boldsymbol{\Sigma}_{\boldsymbol{S},\boldsymbol{W}}^{sd}$ as*

$$\boldsymbol{\Sigma}_{\boldsymbol{S},\boldsymbol{W}}^{sd} = \boldsymbol{Q}_{\boldsymbol{S},\boldsymbol{W}}^{sd} \, \mathrm{Diag}(\omega_1^{\boldsymbol{S},\boldsymbol{W}}, \cdots, \omega_d^{\boldsymbol{S},\boldsymbol{W}}) \left( \boldsymbol{Q}_{\boldsymbol{S},\boldsymbol{W}}^{sd} \right)^{\top}.$$

*Proof of Theorem 3.* By Lemma 13, $\text{Gen}_T$ depends on $\boldsymbol{\Sigma}_t(\boldsymbol{S}, \boldsymbol{W})$ only through $A_t(\boldsymbol{S}, \boldsymbol{W})$, and we have

$$
\begin{aligned}
&\boldsymbol{\Sigma}_s^*(\boldsymbol{S}, \boldsymbol{W}) \\
&= \arg \min_{\text{Constraint 1}} A_s(\boldsymbol{S}, \boldsymbol{W}) \\
&= \arg \min_{\text{tr}(\boldsymbol{\Sigma})=c_s(\boldsymbol{S}, \boldsymbol{W})} \frac{1}{2} \left( \text{tr} \left( \boldsymbol{\Sigma}^{-1} \left( \sigma_s(\boldsymbol{S_J}, \boldsymbol{W}_{s-1})\mathbb{I} + \frac{\eta_s^2}{Nb_s} \left( \frac{N}{N-1} \right)^2 \boldsymbol{\Sigma}_{\boldsymbol{S}, \boldsymbol{W}_{s-1}}^{sd} \right) \right) \right. \\
&\qquad \left. - d \ln \sigma_s(\boldsymbol{S_J}, \boldsymbol{W}_{s-1}) - d + \ln\left( \det \boldsymbol{\Sigma} \right) \right) \\
&= \arg \min_{\text{tr}(\boldsymbol{\Sigma})=c_s(\boldsymbol{S}, \boldsymbol{W})} \frac{1}{2} \left( \text{tr} \left( \boldsymbol{\Sigma}^{-1} \left( \sigma_s(\boldsymbol{S_J}, \boldsymbol{W}_{s-1})\mathbb{I} + \frac{\eta_s^2}{Nb_s} \left( \frac{N}{N-1} \right)^2 \boldsymbol{\Sigma}_{\boldsymbol{S}, \boldsymbol{W}_{s-1}}^{sd} \right) \right) \right. \\
&\qquad \left. + \ln\left( \det \boldsymbol{\Sigma} \right) \right).
\end{aligned}
$$

Applying Lemma 9 completes the proof.

$\square$

## E.4 Smaller Condition Number

In this section, we demonstrate why the optimal noise of Theorem 3 has smaller condition number than $\boldsymbol{\Sigma}^{sd}$ as the following corollary.

**Corollary 1.** *The optimal noise covariance $\boldsymbol{\Sigma}^*$ given by Theorem 3 has smaller condition number than $\boldsymbol{\Sigma}^{sd}$.*

*Proof.* We prove this claim following two steps.

Firstly, the noise covariance of the prior is isotropic, has condition number 1, and push the condition number of $\sigma_t\mathbb{I} + \frac{\eta_t^2}{Nb_t} \left( \frac{N}{N-1} \right)^2 \Sigma_{\boldsymbol{S}, \boldsymbol{W}}^{sd}$ smaller than $\Sigma_{\boldsymbol{S}, \boldsymbol{W}}^{sd}$.

Secondly, the optimal solution $G$ of Lemma 9 always has a smaller condition number than $B$, which implies that $\Sigma_t^*(\boldsymbol{S}, \boldsymbol{W})$ has smaller condition number than $B = \sigma_t\mathbb{I} + \frac{\eta_t^2}{Nb_t} \left( \frac{N}{N-1} \right)^2 \Sigma_{\boldsymbol{S}, \boldsymbol{W}}^{sd}$. Hence the condition number of $\Sigma_t^*(\boldsymbol{S}, \boldsymbol{W})$ is smaller than $\Sigma_{\boldsymbol{S}, \boldsymbol{W}}^{sd}$. $\square$

# F Experiments

In this section, we introduce the settings of the experiments in Fig. (1) Fig.(2), Fig.(3), and Fig.(4). We further include an additional experiment comparing the generalization error between SGLD with square rooted empirical gradient covariance (SREC-SGLD) (the optimal noise covariance in Theorem 2) and SGLD with empirical gradient covariance (EC-SGLD) subject to Constraint 1.

## F.1 Experiment settings

For both Fig. (1), Fig.(2), Fig. (3), and Fig. (4), we adopt the same setting as the Fashion-MNIST experiment of [40, Section D.3] despite enlarging the training set. Specifically, we use the 4-layer convolutional neural network as our model to conduct multi-class classification on Fashion-MNIST [35]. Concretely, this convolutional neural network can be expressed in order as: convolutional layer with 10 channel and filter size $5 \times 5$, max-pool layer with kernel size 2 and stride 2, convolutional layer with 10 channel and filter size $5 \times 5$, max-pool with kernel size 2, two fully connected layer with width 50. Our training set consists of 10,000 examples uniformly sampled without replacement from the Fashion-MNIST dataset. Our training set is larger than that in [40] (which only contains 1200 samples), but is still one sixth of the whole Fashion-MNIST dataset due to the computational burden of gradient descent (without mini-batch) in the SGLD. The learning rates of all SGLD are set to 0.07, which is exactly the same as [40]. We also set the learning rate of SGD in Fig. (1) to 0.07 for fair comparison with SGLD.

**Empirical gradient covariance:** We use top 100 eigenvalues to approximate the empirical gradient covariance matrix. Specifically, we decompose the matrix $\Sigma_{\boldsymbol{S},\boldsymbol{W}}^{sd}$ into $(Q_{\boldsymbol{S},\boldsymbol{W}})^\top(\omega_{[d]}^{\boldsymbol{S},\boldsymbol{W}})Q_{\boldsymbol{S},\boldsymbol{W}}$, and use $(Q_{\boldsymbol{S},\boldsymbol{W}})^\top(\omega_{[100]}^{\boldsymbol{S},\boldsymbol{W}}, \mathbf{0}_{d-100})Q_{\boldsymbol{S},\boldsymbol{W}}$ to approximate $\Sigma_{\boldsymbol{S},\boldsymbol{W}}^{sd}$.

**Noise Scale:** In Fig. (1) and Fig.(2), the traces of all SGLDs are set to be $\mathrm{tr}(\Sigma_{\boldsymbol{S},\boldsymbol{W}}^{sd})$; in Fig. (3), the traces are set to be $\mathrm{tr}((\Sigma_{\boldsymbol{S},\boldsymbol{W}}^{sd})^{1/2})$ and $5\,\mathrm{tr}((\Sigma_{\boldsymbol{S},\boldsymbol{W}}^{sd})^{1/2})$, respectively in (a) and (b); in Fig. (4), the traces are set to be $\mathrm{tr}((\Sigma_{\boldsymbol{S},\boldsymbol{W}}^{sd})^{1/2})$.

**Noise frequency:** Similar to [40], we re-estimate the noise structure of all SGLDs every 10 epochs to ease the computational burden.

### F.2 Comparison between EC-SGLD and SREC-SGLD

We further conduct an experiment to compare the generalization performance between Iso-SGLD, EC-SGLD and SREC-SGLD, with the traces of the covariance are all set to be $5\,\mathrm{tr}((\Sigma_{\boldsymbol{S},\boldsymbol{W}}^{sd})^{1/2})$, and all other settings consistent with Appendix F.1. The generalization error along the iteration of SREC-SGLD, Iso-SGLD, and EC-SGLD is plotted as Fig. 5, where one can easily observe the generalization error of SREC-SGLD is the smallest, which supports Theorem 2.

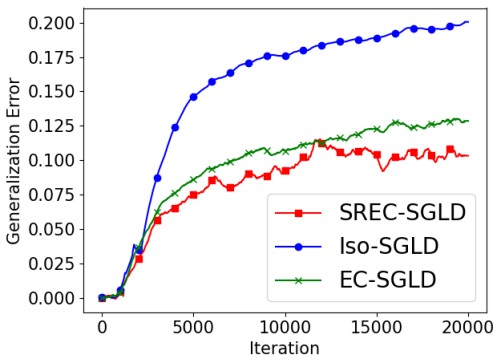

Figure 5: Comparison of generalization error for SGLDs with different noise structures. Traces of the covariances are all set to be $5\,\mathrm{tr}((\Sigma_{\boldsymbol{S},\boldsymbol{W}}^{sd})^{1/2})$.