# OpenReview forum: "Optimizing Information-theoretical Generalization Bound via Anisotropic Noise of SGLD"
_NeurIPS.cc/2021/Conference — NeurIPS 2021 Poster_

### Official Review · Reviewer_TVaS · 2021-07-10

**Rating:** 5
**Confidence:** 3

**Summary:**

This paper studies optimization of the information theoretical generalization bounds of Stochastic Gradient Langevin Dynamics (SGLD) with respect to the noise covariance. Here are the main findings:
* When updating rule is fixed and only optimizing noise covariance under the constraint that the trace of the covariance is fixed, the noise covariance of SGLD is similar to the empirical gradient covariance
* When both updating rule and the noise covariance are jointly minimized, the noise covariance of SGLD is the square root of the expected gradient covariance.
The two results can potentially be used to improve information theoretical generalization bounds obtained in the literature. The theoretical results are validated by some numerical experiments.


**Limitations And Societal Impact:**

Yes

**Main Review:**

SGLD is a popular stochastic optimization algorithm used in practical training of neural networks. Its generalization property is an important issue and is worth to study. The present work aims (and manages) to improve information theoretical generalization bounds obtained in the literature (e.g. Refs [24,13]) by optimizing the noise covariance. However, I do have some concerns on the results.
* My major concern is about the usefulness of the information theoretical generalization bounds. From my perspective, the generalization bound in Theorem 2 seems very brutal. The reason is that I would expect the generalization error to decrease or at least not increase with respect to the training time $T$, while the right hand side of Eqn (7) increases linearly in $T$ as it involves the relative entropy between two measures on the whole path space of trajectories. If the upper bound is brutal, then I do not see the point of optimizing the right hand side since it is not likely to improve the actual generalization error.
* Speaking about the information theoretical bound, I would think of a lower bound (rather than upper bound) for the generalization error involving information quantities, such as relative entropy or mutual information. This is because it is the lower bound that reflects the fundamental information limit of a learning algorithm (SGLD in this case).
* In Figure 2, I do not see why the generalization error could increase along the epoch. Is the figure showing the information theoretical upper bound or the actual generalization error? If it is the former, please specify this in the figure.
* The improvement of SREGC-SGLD over IS-SGLD seems not convincing in Figure 2.c since all the curves are nearly aligned with each other.


**Time Spent Reviewing:**

48 hours

---

> ### Author Response · Authors · 2021-08-10
> **Thank you for the comments. Here are our point-to-point responses.**
>
> **Q1**: Usefulness of the information-theoretical bounds. It may not reasonable that the generalization error increasing with time.
>
> **A1**: Thank you for the comment. There might be a misunderstanding about the ``generalization error``. The reviewer may refer to  the **test error** (error on the test data) as generalization error, which usually decreases with the training time $T$. However, in this paper, we use generalization error to represent the gap between the training error and the test error.
>
> This generalization error usually increases with the training time $T$. This is observed practically and  the reason of doing ``early stopping''. Intuitively,  as the training goes,  the model get fitting to the training data more and more, and the gap between the performances on the training set and test data  gets larger. This theoretically aligns with other learning principles, e.g., minimum description length and Kolmogorov complexity.
>
> It has been shown that information-theoretical generalization bounds are non-vacuous and closely related with the real generalization error even in deep learning (see Figure 1. (b) in [24] and Table 1 and Figure 1 in [13]). Our experiments (Figure 2. (a) and Figure 2. (b)) shows that the real generalization error increases with time (while the test loss does not), which indicates the information-theoretical bound is reasonable.
>
>
> **Q2**: Speaking about the information theoretical bound, I would think of a lower bound (rather than upper bound) for the generalization error involving information quantities, such as relative entropy or mutual information. This is because it is the lower bound that reflects the fundamental information limit of a learning algorithm (SGLD in this case).
>
> **A2**:  Thank you for the suggestion. We are not aware of existing work on lower  bound on generalization by using information theory, and anticipate that the reviewer could provide more explicit formulation about this.
>
>
> **Q3**: In Figure 2, I do not see why the generalization error could increase along the epoch. Is the figure showing the information theoretical upper bound or the actual generalization error? If it is the former, please specify this in the figure.
>
> **A3**:  It is the actual generalization error, i.e., the gap between the training error and the test error. Figure 2 shows that the real generalization error increases along the trajectory, which agrees with the information-theoretical generalization bounds.
>
>
> **Q4**: The improvement of SREGC-SGLD over IS-SGLD seems not convincing in Figure 2.c since all the curves are nearly aligned with each other.
>
> **A4**: There could be a misunderstanding. Figure 2.c is the training loss of all the optimizers, which is to verify the Lemma 2 that larger trace of noise covariance makes the convergence rate slower.
>
> The Figure 2.a and Figure 2.b show the generalization errors of SREGC-SGLD and IS-SGLD with different noise scales, which clearly demonstrates the advantage of SREGC-SGLD over IS-SGLD in terms of achieving smaller generalization error.

---

> > ### Comment · Reviewer_TVaS · 2021-08-18
> > **Reply to authors**
> >
> > I want to thank the authors for the feedback. I apologize that I was indeed talking about test author instead of the generalization as explained in the paper and reply. However, I am still not convinced why the time-increasing generalization error bound could be useful. I will keep my score unchanged.

---

> > > ### Author Response · Authors · 2021-08-21
> > > **Reply to the concern why the time-increasing generalization error bound could be useful.**
> > >
> > > **Q**:However, I am still not convinced why the time-increasing generalization error bound could be useful.
> > >
> > >
> > > **A**: First, the time-increasing behavior of the  generalization error is not something we propose but rather empirical facts. The empirical generalization error (the gap between the **training loss** and the **test loss**) increases over training iterations in the real-world experiments, which can be observed as Figure 2, the figure in https://www.dropbox.com/s/bl3lazy055kc9ac/ComparedtoSGD.png?dl=0, and the figures in https://www.dropbox.com/s/72o1b5mdv14cb2h/CIFAR10_Experiment.zip?dl=0 which trains ResNet18 on CIFAR 10 (Standard hyperparameter setting: lr 0.1 and batch size 128, lr decay 1/10 at epoch 100 and 150 without data augmentation and weight decay as these regularization tricks are out of the scope of the theory). Such empirical observation is well captured by the information theoretical bounds.
> > >
> > > Second, while increasing with iterations, the information-theoretical bounds usually converge to some finite value instead of diverging to infinity. See curves  in Figure 1 (b,c) of [23] https://arxiv.org/pdf/1911.02151.pdf, and curve "Negrea et al. 2019" and curve "CMI bound (ours)" in the left column of Figure 1 in [13] https://arxiv.org/pdf/2004.12983.pdf.

---

### Official Review · Reviewer_Zcyu · 2021-07-16

**Rating:** 6
**Confidence:** 3

**Summary:**

This paper study the interplay between generalization bound and the covariance of perturbed gradient descent. The authors invoke a recent (information-theoretic) bound for stochastic optimization methods. Then, they optimize the upper bound with respect to trace constraint on the covariance of noise in each step and experimentally show that this optimization may be effective for training neural networks.

**Limitations And Societal Impact:**

Yes

**Main Review:**

I could not follow the statement of theorems and theoretical results. Definitely, the paper requires refinement of notations and also writing.

I can not understand the notations in Propostion1. In equation 4, what is exactly the risk for a joint distribution $Q_T^{S,V_T}$? The input of the risk has to be a parameter, not the joint distribution of a stochastic process.

I am not sure whether I understood the statement of Proposition 1 and Theorem 1. Suppose that $T=0$, then the starting distribution is the same so the right side of equations 4 and 6 is zero. But the left side is $R_D(W_0) -R_S(W_0)$ which is not necessarily less than zero. Intuitively, if the algorithm is deterministic, still the generalization error is not zero. Should not use an absolute value for the left side of the generalization error?

The optimality is not defined well. The target objective is optimizing the divergence of test error from the training error under a trace constraint for the covariance of the noise, which is meaningless. The optimal algorithm requires to achieve the lowest level of the test error. Indeed, authors require to minimize $$ E [R_D(W_T)] $$  which I am not sure whether is possible.

One of the claims in the introduction is that the optimal covariance matrices found in this paper match the covariance of stochastic gradient descent. I saw in experiments there is one result indicating anti-isotropy of the noise of stochastic gradients which is a known fact in the literature. Is there any other result that shows the optimal covariance matrices extracted by the proposed algorithm is similar to the covariance structure of stochastic gradients during training?


**Post Rebuttal**


I thank the authors for their detailed responses that helped me to understand the results better. Now, I understand Lemma 2 and the role of the trace constraint in optimization. After a more careful study, I decided to increase my score. This paper is rather difficult to follow due to the use of complicated notations and I highly recommend the authors to improve the writing.

**Time Spent Reviewing:**

3.5

---

> ### Author Response · Authors · 2021-08-10
> **Thank you for the comments. Here are our point-to-point responses.**
>
> **Q1**: I could not follow the statement of theorems and theoretical results. Definitely, the paper requires refinement of notations and also writing.
>
> **A1**: Thank you for the feedback. The notations are a bit complicated because we try to make the statement rigorous and consistent with existing information-theoretical generalization bounds [12,13]. We will revise the paper by adding  intuitive explanation and by discussing the connection of the theoretical results.
>
> **Q2**: What is exactly the risk for a joint distribution $Q_T^{ S, V_T}$? The input of the risk has to be a parameter, not the joint distribution of a stochastic process.
>
> **A2**: Different from the risk defined on a parameter in PAC, the risk here is defined over a distribution of the parameter, which is the core concerned quantity in the PAC Bayesian and information theoretical generalization literature [13,23].  For any function $f: \mathcal{X}\rightarrow\mathcal{Y}$ and any distribution $Q$ on $\mathcal{X}$, $f(Q)$ is defined as the expectation of $f$ with respect to $x\sim Q$.
>  Since $Q_T^{ S, V_T}$ is a distribution of parameter conditional on $S,V_T$ at time $T$, $\hat{\mathcal{R_{ S}}} $ $(Q_T^{ S, V_T})$ means the expectation of $\hat{\mathcal{R_{ {S}}}}$ with respect to distribution $Q_T^{ S, V_T}$, and $\mathcal{R_{\mathcal{D}}} (Q_T^{ S, V_T})$ means the expectation of $\mathcal{R_{\mathcal{D}}}$ with respect to distribution $Q_T^{ {S}, {V}_T}$.
>
>
> **Q3**:  At the initial point the right sides of equations 4 and 6 are zero, while the left sides are not zero, which seems to be a contradiction.
>
> **A3**:  There is no contradiction. The calculation should be conduct as follows. At the initial point, the distribution of the parameters $Q_0^{ S, V_{[T]}}$ are independent of $ S$ and $ V_{[T]}$. Therefore,
> $$E_{S, {V_{[T]}}}  [\mathcal{R_{\mathcal{D}}} ({Q_0}^{ {S}, V_{[T]}} )-\hat{\mathcal{R_S}}({Q_0}^{ {S}, V_{[T]}})]=\mathcal{R_{\mathcal{D}}}\left({Q_0}^{ {S}, V_{[T]}}\right)-\mathcal{R_{\mathcal{D}}}\left({Q_0}^{ {S}, V_{[T]}}\right)=0$$
> and the equities in 4 and 6 hold.
>
> **Q4**: The optimality is not defined well. The target objective is optimizing the divergence of test error from the training error under a trace constraint for the covariance of the noise, which is meaningless. The optimal algorithm requires to achieve the lowest level of the test error. Indeed, authors require to minimize $\mathbb{E}[\mathcal{R}_{\mathcal{D}}( {W}_T)]$.
>
> **A4**:  It is a standard practice to separate $\mathbb{E}[\mathcal{R_{\mathcal{D}}}( W_T)]$ into $\mathbb{E}[\hat{\mathcal{R}}_ S( W_T)]$ (optimization error) and $\mathbb{E}[\mathcal{R_\mathcal{D}}( W_T)]-\mathbb{E}[\hat{\mathcal{R_S}}( W_T)]$ (generalization error) and analyze them respectively, because it is hard to minimize the expected risk $\mathbb{E}[\mathcal{R}_{\mathcal{D}}( W_T)]$ as we can only gain access to finite training data in practice. In this paper, we also follow this decomposition: by Lemma 2 and Figure 1, the trace of noise covariance is an indicator of the convergence rate of optimization, and it is reasonable to keep the scale of trace of noise covariance as constraint to keep the convergence rate about the same, and analyze the generalization error under this constraint.
>
> **Q5**:  Additional results for the optimal noise covariance is close to that of SGD?
>
> **A5**:  As the optimal solutions of Problem 1 and Problem 2 in Theorems 3 and 4 have explicit form (see explicit form of the solution in Theorem 3 in Appendix A.4), it can be shown theoretically the solutions are close to the noise covariance of mini-batch SGD: solution of Theorem 3 has the same eigenvectors with mini-batch SGD and the same order eigenvalues (see the discussion below Theorem 3), while solution of Theorem 4 is the squared root of the expected covariance of mini-batch SGD. The purpose of experiment on the anti-isotropy of noise of stochastic gradients (Figure 3) is to further indicates the anti-isotropy of the optimal solutions of Problem 1 and Problem 2.

---

> > ### Comment · Reviewer_Zcyu · 2021-08-16
> > **Followup questions**
> >
> > Thank you very much for the clarification!
> >
> > I have a few more questions:
> >
> > 1. Regarding lemma 2, if we multiply the objective by constant $\alpha$, then all terms can be scaled back by a proper choice of stepsize except the trace constraint that is used in your optimization. Would you please explain more about this scaling variant term
> > 2. Let's say the ultimate goal is achieving the lowest expected risk and we impose this goal by the trace constraint algorithm relying on the bound of Lemma2. Is it possible to (at least empirically) show that this constraint approximately leads to the minimal test error.
> > 3. The noise non-isotropy is a minimal similarity between the noise of SGD and the covariance structured extracted in this paper. Can the authors provide more quantitative comparisons between SGD noise and the derived covariance noise?

---

> > > ### Author Response · Authors · 2021-08-19
> > > **Thanks for the raising the score. Additional concerns are addressed below.**
> > >
> > > **Q1**: Regarding lemma 2, if we multiply the objective by constant $\alpha$, then all terms can be scaled back by a proper choice of stepsize except the trace constraint that is used in your optimization. Would you please explain more about this scaling variant term?
> > >
> > > **A1**: In Lemma 2, the upper bound of empirical loss after iteration is
> > > $$ \mathbb{E}\_{t+1|t}\mathcal{R}\_{S} (W\_{t+1})\le \mathcal{R}\_{S} (W\_{t})-(1-\frac{\beta\eta\_{t+1}}{2})\eta\_{t+1}\Vert\nabla \mathcal{R\_{S}}(W\_{t})\Vert^{2}+\frac{\beta}{2}tr\left( \frac{\eta\_{t+1}^2(N-b_t)}{(N-1)b_t}\Sigma^{sd}\_{S,W_t}+\Sigma\_{t+1}(S,W_t)\right).$$
> > >
> > > If loss $\ell$ is multiplied by a constant $\alpha$, i.e., $\tilde{\ell}=\alpha \ell$, we will the obtain a scaled empirical risk $\tilde{\mathcal{R}}_S=\alpha \mathcal{R}_S$, a scaled gradient $\nabla \tilde{\mathcal{R}}_S=\alpha \nabla \mathcal{R}_S$ and a scaled smoothness factor $\tilde{\beta}=\alpha\beta$.  Hence we need to choose new learning rate $\tilde{\eta}\_t=\frac{1}{\alpha}\eta_t$.  Moreover, we note that the new noise covariance of SGD $\tilde{\Sigma}^{sd}\_{S,W_t}=\alpha^2 \Sigma^{sd}\_{S,W_t}$. If  plugging these changes into the above inequality and cancelling an $\alpha$ factor on both side, we will obtain the same inequality as above.
> > >
> > > The main intuition is that the scaling of the gradient and the scaling of the learning rate in the update rule offset each other and the update rule remains the same, which makes the change of the loss the same.
> > >
> > >
> > > **Q2**: Let's say the ultimate goal is achieving the lowest expected risk and we impose this goal by the trace constraint algorithm relying on the bound of Lemma 2. Is it possible to (at least empirically) show that this constraint approximately leads to the minimal test error.
> > >
> > > **A2**: Good question and thanks for asking. As both  the empirical loss in Lemma 2 and the generalization error bound  depend on the trace of the noise covariance, it is possible to optimize them jointly (i.e., optimize the test error) with respect to the trace. At the same time, this is not an easy task because the optimization target is rather complicated and requires deliberate treatment. We believe this is a very interesting question and deserves an independent study.   We defer the derivation of the optimal trace in terms of minimal test error to the future work.
> > >
> > > **Q3**: More quantitative comparisons between SGD noise and the derived covariance noise?
> > >
> > > **A3**: The optimal noise covariance in Theorem 4 is the squared root of the expected covariance of mini-batch SGD. To show their empirical comparison, we run the additional experiment (https://www.dropbox.com/s/bl3lazy055kc9ac/ComparedtoSGD.png?dl=0), where the dataset and the network are the same as Figure 2. It can be observed that the mini-batch SGD and the SGLD with the optimal covariance achieve similar generalization error, smaller than that of the SGLD with isotropic noise.

---

> > > > ### Author Response · Authors · 2021-08-23
> > > > **Appreciate it if the reviewer provides feedback instead of only decreasing the score.**
> > > >
> > > > We find that the reviewer changed the score from 6 to 5 recently but left no feedback about our response. Expect that the reviewer can explain the reason for doing so.

---

> > > > > ### Comment · Reviewer_Zcyu · 2021-08-23
> > > > > **Response**
> > > > >
> > > > > Please find my response in the post rebuttal section. The update in the score is based on the experimental results that show your optimal noise structure does not benefits from any sort of acceleration compared to SGD. I believe this observation does not confirm the paper's claim that "the optimal noise covariance of SGLD
> > > > >  exhibits structure similar to the empirical gradient covariance". Note that overall my score is changed from 4 to 5 after your detailed response. Yet, I believe that this claim is not supported empirically nor theoretically in the paper. Your experiment can imply that your optimality criteria are not useful and you are optimizing a very loose bound on the main objective.

---

> > > > > > ### Author Response · Authors · 2021-08-27
> > > > > > **Further clarification of our results**
> > > > > >
> > > > > > Dear reviewer,
> > > > > >
> > > > > > We are sorry that the reviewer still has misunderstanding about our result after all responses. We further clarify our result as follows.
> > > > > >
> > > > > > **What we do in this paper.**
> > > > > >
> > > > > > Optimizing the information-theoretical generalization bound, with respect to the added noise structure in SGLD $\rightarrow$ The optimal noise exhibits a similar structure to the empirical gradient covariance. $\rightarrow$ The benefit of the optimal noise structure over the usual isotropic noise is both theoretically and empirically demonstrated, which implicitly verifies the implicit regularization effect of stochastic gradient noise.
> > > > > >
> > > > > > **What we do not do in this paper.**
> > > > > >
> > > > > > 1. We do not analyze SGD and compare with SGD because there is no information-theoretical generalization bound for SGD.
> > > > > >
> > > > > > 2. The goal of obtaining optimal noise structure is not to get acceleration over SGD.
> > > > > >
> > > > > > 3. The tightness of the information-theoretical bounds has been claimed and verified by existing works (e.g. [13, 24]) (these two works are both accepted by NeurIPS).

---

> > > > > > > ### Comment · Reviewer_Zcyu · 2021-08-27
> > > > > > > **Further questions**
> > > > > > >
> > > > > > > 1. Would you please explain what is the difference between the noise structure of SGD and the empirical gradient covariance matrix? Isn't the empirical covariance matrix the second moment of SGD noise?
> > > > > > > 2. Why authors sent me this plot: https://www.dropbox.com/s/bl3lazy055kc9ac/Compared_to_SGD_Generalization.png?dl=0?
> > > > > > > Which compares SGD with other methods?

---

> > > > > > > > ### Author Response · Authors · 2021-08-30
> > > > > > > > **Answers for the further comments.**
> > > > > > > >
> > > > > > > > Q1: Would you please explain what is the difference between the noise structure of SGD and the empirical gradient covariance matrix? Isn't the empirical covariance matrix the second moment of SGD noise?
> > > > > > > >
> > > > > > > >
> > > > > > > >
> > > > > > > > A1: The noise covariance of SGD is the same as the empirical gradient covariance matrix. However, the optimal noise of SGLD is a (continuous) Gaussian distribution with covariance is similar to the noise covariance of SGD, while the distribution of SGD noise is discrete which can be enumerated by uniformly sampling from the finite dataset. Therefore, SGD is not a special case of SGLD, and SGLD is only used as an approximation to analyze SGD (which follows a similar routine as [37,32]). Analyzing the dynamics of SGD directly is technically difficult. For example, the KL divergence between prior and posterior is infinity due to the finite support set of the posterior distribution given by SGD.
> > > > > > > >
> > > > > > > >
> > > > > > > >
> > > > > > > > Q2: Why authors sent me this plot that compares SGD with other methods?
> > > > > > > >
> > > > > > > >
> > > > > > > >
> > > > > > > > A2: The comparison is to show even though with such a gap in noise distribution (continuous Gaussian noise v.s. discrete noise), SGLD with optimal noise covariance can recover a similar low generalization error of SGD, significantly better than the isotropic SGLD. This justifies that the structure of noise covariance plays a critical role in that SGD generalizes better than isotropic SGLD. We do not argue that SGLD with optimal noise covariance generalizes better than SGD

---

> > > > > > > > > ### Comment · Reviewer_Zcyu · 2021-08-30
> > > > > > > > > **Confusing claims**
> > > > > > > > >
> > > > > > > > > There are still contradictory claims in your responses. You claimed that you are not analyzing SGD and not comparing with SGD, while the empirical covariance of the stochastic gradient is the noise of SGD. This contradicts your statement :*"the optimal noise exhibits a similar structure to the empirical gradient covariance"* which is a comparison with SGD noise.
> > > > > > > > > This claim is not supported either experimentally or theoretically. You are showing that both are non-isotropic but this does not conclude any sort of similarities. It is very important to support this claim rigorously. Otherwise, you are solving an optimization problem that does not bring any theoretical nor experimental benefits. You can easily compare moments of noises (in terms of norm 2 or fibrinous norm) and show that their moments match or get close to each other during optimization.

---

> > > > > > > > > > ### Author Response · Authors · 2021-08-31
> > > > > > > > > > **Further clarification**
> > > > > > > > > >
> > > > > > > > > > To begin with, we would like to provide an overview of the motivation/logic of this paper.
> > > > > > > > > >
> > > > > > > > > > First of all, SGD achieves excellent performance in terms of generalization error in deep learning. However, the dynamic of SGD is hard to analyze because of the discrete nature of empirical distribution.  Usually, people use SGLD as a theoretic tool to approximate and analyze SGD.
> > > > > > > > > >
> > > > > > > > > > However, SGLD with isotropic noise has a huge generalization gap compared with SGD. This paper tries to fill this gap by studying the optimal noise covariance structure of SGLD with respect to minimizing an information-theoretical generalization bound. There is an interesting observation that the obtained noise covariance is similar to the noise covariance of SGD, which can serve as strong evidence to explain the superior generalization behavior of SGD.
> > > > > > > > > >
> > > > > > > > > > With the logic in mind, we now come to answer the reviewer's questions.
> > > > > > > > > >
> > > > > > > > > > **Q1**: You claimed that you are not analyzing SGD and not comparing with SGD, while the empirical covariance of the stochastic gradient is the noise of SGD. This contradicts your statement: "the optimal noise exhibits a similar structure to the empirical gradient covariance" which is a comparison with SGD noise.
> > > > > > > > > >
> > > > > > > > > > **A1**:  In our earlier reply, by saying "we are not analyzing SGD and not comparing with SGD", we mean that we don't analyze the generalization error bound of SGD (which doesn't exist for now) and hence **cannot compare the generalization error bound of SGLD with that of SGD**. All we do is optimizing the information-theoretical generalization bound of SGLD and seeing what the optimal noise covariance looks like. **What we compare are the noise covariance of SGD and the optimal noise covariance of SGLD.** It turns out that the optimal noise covariance of SGLD is similar to SGD as the optimal noise covariance given by Theorem 4 is the square root of the noise covariance of SGD.
> > > > > > > > > >
> > > > > > > > > > **Q2**: This claim is not supported either experimentally or theoretically. You are showing that both are non-isotropic but this does not conclude any sort of similarities.
> > > > > > > > > >
> > > > > > > > > > **A2**: As mentioned in the earlier reply, the optimal noise covariance is the square root of the noise covariance of SGD. That a matrix $A$ is the square root of another matrix $B$ is **a direct similarity** as they share **the same eigenvectors** and **the eigenvalues of $A$ are the square root of those of $B$**.  Hence, the claim is supported by our theory well.
> > > > > > > > > >
> > > > > > > > > > The experiment is not meant to show the claim as it is already theoretically proved. The experiment (Figure 3) is to show how isotropic noise covariance of SGD is (and therefore how isotropic the optimal noise covariance is).

---

> > > > > > > > > > > ### Comment · Reviewer_Zcyu · 2021-08-31
> > > > > > > > > > > **Convince about SGD link**
> > > > > > > > > > >
> > > > > > > > > > > I went through Thm. 3 today and your reasoning about the square root (some midpoint between identity and covariance of gradient). I also checked the lemma 4 (13). Then, I increased my score. I kindly ask authors to improve the writing of the paper. Also, responses have to be as clear as possible. For example, the SGD comparison result in your response confused me a lot (that is the reason why I decreased my score). The square root of the matrix is not uniquely defined (in your last response). Your goal ultimate goal is a comparison of the optimal noise with SGD noise (in the previous response). You can be more precise about the similarity in the intro and say the eigenvectors are shared when we assume an isotopic prior model. You must highlight that the importance of isotropic prior to your analysis.  I strongly believe that experiments are very confusing. Maybe, this paper does not require any experiments.

---

> > > > > > > > > > > > ### Author Response · Authors · 2021-09-01
> > > > > > > > > > > > **Thanks for the suggestions. We will revise the paper accordingly.**
> > > > > > > > > > > >
> > > > > > > > > > > > We would like to thank the reviewer for the responsible review and patient response. We will revise the paper accordingly. Specifically, we will:
> > > > > > > > > > > >
> > > > > > > > > > > > **1.** Make the description of the similarity between the optimal noise covariance and the noise covariance of SGD more precise and detailed in Section I (Introduction) (e.g., shared eigenvectors, square-rooted eigenvalues, etc.).
> > > > > > > > > > > >
> > > > > > > > > > > > **2.** Highlight the definition of the common prior before Theorem 3.
> > > > > > > > > > > >
> > > > > > > > > > > > **3.** Describe the purpose of experiments more precisely in Section 7.

---

### Official Review · Reviewer_Mhnz · 2021-07-17

**Rating:** 6
**Confidence:** 4

**Summary:**

This paper considers the problem of understanding the generalization of the SGLD algorithms using information-theoretic techniques. This line of work initiates by [27],[24], and [13]. The goal in this paper is to "modify" the SGLD algorithms such that it has a good generalization properties and good performance on the training set. Basically, the modification that the authors considered is to make the covariance of the Gaussian noise in the SGLD update to depend on the trajectory. Note that in the usual setting the noise is isotropic. Then, authors claim that controlling the trace of the SGLD ensures we have good performance on training set and good generalization. Then the authors provide a generalization bound which uses reverse KL between posterior and the prior distribution. I think this bound is not true as stated in my review. Since this bound has been used in the paper, I think the authors should comment on the validity of the theorem.

**Ethical Concerns:**

There is no ethical concern with this paper.

**Limitations And Societal Impact:**

I have some concern on the validity of one of the main results in the paper.

**Main Review:**

- The presentation of the paper is not good.

- Why the assumption 1 is a valid assumption? There is no discussion in the paper.

- I think Theorem 2 in the paper is NOT CORRECT. I will provide a counterexample here. Assume that we have a finite hypothesis class $H= \{ h_1,...,h_M \} $. Assume the learner is an "ERM" so after receiving a training set it outputs a single hypothesis that minimizes the empirical error. Assume we have a consistent way to break the ties. For instance we choose the hypothesis with smallest index. Therefore the posterior distribution is $Q^{S}= \delta_{h_{i^\star}}$ where $h_{i^\star}$ is the ERM classifier. Assume based on your generalization bound in Theorem 2, we choose the uniform prior over the set of hypothesis, i.e., $P(h_i)=\frac{1}{M}$. Based on this theorem $KL(P||Q)=\frac{1}{M}\log \frac{1}{M}$. Therefore we obtain   $EGE<\sqrt{\frac{1}{2}\frac{1}{M}\log \frac{1}{M}}$. It implies that when $M\to \infty$, the generalization bound becomes smaller!!!  From the first part of "Understanding machine learning: From theory to algorithms" we know the expected generalization error is upper and lower bounded by $\log(M)$. Therefore this example depicts that this bound can't be true. I also checked the proof and the proof also has error in Lines 584 and 585.

The main reason is that similar to PAC-Bayesian bounds we can't have upper bound with "revers KL" between the posterior and prior.


As this bound is used all over the paper, I appreciate if the authors comment on how this theorem affects the paper's conclusion. Also if there is a misunderstanding by me, please let me know.

**Time Spent Reviewing:**

7

---

> ### Author Response · Authors · 2021-08-10
> **Thank you for the comments! Here are our point-to-point responses.**
>
> **Q1**: The presentation of the paper is not good.
>
> **A1**: Thank you. We are always trying to improve the presentation. It would be great if the reviewer can point out which part is not good and provide some concrete suggestions.
>
>
> **Q2**: Why the assumption 1 is a valid assumption?
>
> **A2**: The Assumption 1 states that fixing $P$, the optimal $Q$ is to minimize $\text{KL}(P||Q)$ and $\text{KL}(Q||P)$ are the same under some constraint (Constraint 1). This is a valid assumption although the KL divergence is asymmetric. For example, for the case without constraints, fixing $P$, the minima of $\text{KL}(P||Q)$ and $\text{KL}(Q||P)$ are achieved at the same time when $Q=P$. Thanks for pointing this out and we will add the above discussion below the assumption.
>
>
>
> **Q3**: Theorem 2 is not correct because of a counterexample. Error in Line 584 and Line 585.
>
> **A3**: The calculation of $\text{KL}(P ||Q)$ in the counterexample is wrong. The KL divergence in the example should be  $\text{KL}(P||Q)=\frac{1}{M}\log\frac{1}{M}+(M-1)\cdot\infty=\infty$ with $M>1$, which is obviously larger than the finite generalization error. Hence it complies with the conclusion of Theorem 2 nicely.
>
>
> We did not find an error in Line 584-585. Line 584 is a direct application of the Donsker-Varadhan variational formula (Corollary 4.15, [3]) with $Q=P^{S_{J},V_{[T]}}$ and $P=Q^{S,V_{[T]}}$. Line 585 is further a direct application of Line 584 with $g(W) = \lambda\left(\hat{R}_{S_{J^c}}(W)-R_{\mathcal{D}}(W)  \right)$.
> We anticipate that the reviewer can clarify where the error is.
>
> Finally, we would like to provide the intuition why the reverse KL  can bound the generalization error. For the normal KL bound, we have that $\text{KL}(Q||P)\ge g(Q)-g(P)-\log \mathbb{E}_P e^{g-g(P)}$ because of the Donsker-Varadhan variational formula where $-\log \mathbb{E}_P e^{g-g(P)} $ is bounded by the bounded loss assumption. For the reverse KL bound, by permuting $P$ and $Q$ and choosing a new $g$ to be $-g$, we have $\text{KL}(P||Q)\ge g(Q)-g(P)-\log \mathbb{E}_Q e^{g-g(Q)} $  and $\log \mathbb{E}_Q e^{g-g(Q)}$ is bounded by the bounded loss assumption, and the rest of the proof follows exactly the same approach as in the normal KL bound.
>
>
>
> **Q4**: As this bound is used all over the paper, I appreciate if the authors comment on how this theorem affects the paper's conclusion.
>
> **A4**: Theorem 2 is the foundation of the analysis of the optimal posterior noise covariance with the optimal prior. As pointed out in A3 that the counterexample is invalid, Theorem 2 remains flawless.

---

> > ### Comment · Reviewer_Mhnz · 2021-08-11
> > **Thanks for your reponse**
> >
> > Dear Authors,
> >
> > First of all, I would like to apologize for the misunderstanding of your Theorem 2. I reviewed your article again and here is the summary and questions.
> >
> > Summary:
> >
> > This paper tackles an interesting problem. SGLD is a “noisy” variant of SGD in which in each iteration a Gaussian noise is added to the weight vector. However, in practice it has been observed that it is very hard to minimize the empirical risk with SGLD compared to SGD. Having said that, SGLD is more tractable than SGD specifically for obtaining generalization bounds using information-theoretic techniques as shown in [24]. In this paper, the authors consider tuning the noise covariance that SGLD algorithm uses, in order to obtain better generalization bounds. In particular, the authors propose using the update rule in Eq. (3), and the main goal of the paper is obtaining the covariance matrix as a function of W_t and S. Then, under some assumptions,  closed form solution for the optimal covariance matrix is provided.
> >
> > Main Comments:
> >
> > -The presentation of the paper can be improved. I suggest the authors first explain their ideas using full-batch LD so that they can have better notations since they do need to define the random variable which shows random batch indices. Also, the paper has some typos. Moreover, the reader expect after each result the authors provide a discussion of the implications of the results which I could not find.
> >
> > - I think Problem 2 is the main goal of the paper. Why did the author consider Problem 1? As far as I understand the main goal of the paper is to take a principled approach to design the covariance matrix of SGLD that leads to generalization. Note that the prior process, i.e., M~, is just an imaginary process to upper bound the generalization error. Moreover, as stated in [24] you can pick any prior distribution you want as long as it satisfies certain conditions. Therefore, I do not understand why you fix “prior” and then try to find the best covariance matrix. I do think that Problem 2 is the right problem formulation. Therefore, I appreciate it if the authors could provide some explanation why Problem 1 is interesting and relevant.
> >
> > - I could not find any comparison with SGD. I think to show the significance of the results, the authors should provide a comparison with SGD showing that the performance is comparable.
> >
> >
> > - The authors assume that the trace of covariance is fixed. At the end, I could not find any discussion how the authors are going to select the values of c_t? Do we have a result on the convergence of SGLD so that you can theoretically study the impact of c_t on the convergence?
> >
> > - The authors make an assumption, in Assumption 1 at Line 201, which is very strange. Also they do not provide any intuition why it should be true. Can you provide an example other than the case P=Q? As far as I understand this assumption is used for solving Problem 1. Is that correct?
> >
> > - Can you comment on Figure 1? Specifically, why does the green curve start converging faster but all of the sudden its performance degrades? Is the information in the Hessian matrix not useful for designing the noise covariance? What is the implications of the results in this Figure.
> > - Line 153:\ mathcal{H} is not defined.
> > - What is the difference between Theorem 1 and Prop1. I feel that they are exactly the same with different notations.
> >
> > - In Theorem 3, I think choosing Q as a matrix is not a good notation. Since you defined Q as a probability measure before.
> >
> > - Another issue that I found with Problem 1 is that why the prior is not defined based on the noise covariance structure. Note that the prior in [24] is defined for SGLD with isotropic noise structure. Therefore, it seems more reasonable that the prior uses the covariance structure of the posterior here.
> >
> > - Eq. (12): what is z_N?
> >
> > - Eq. (13): I think there is a typo in “J^c \sim D”?
> >
> >
> > **I increase my rating to 5**

---

> > > ### Author Response · Authors · 2021-08-16
> > > **Thanks for the responsive feedback! Additional concerns are addressed below.**
> > >
> > > We greatly appreciate the reviewer's responsive feedback and detailed comments. Here are our point-point responses.
> > >
> > > **Q1**: Suggestions about the presentation. I suggest the authors first explain their ideas using full-batch LD so that they can have better notations since they do need to define the random variable which shows random batch indices. Moreover, the reader expects after each result the authors provide a discussion of the implications of the results which I could not find.
> > >
> > >
> > > **A1**: Thanks for the suggestion. We will illustrate the ideas via the full-batch LD to reduce the notation-wise burden.  Moreover, we will include the proof sketch using the full batch LD while presenting the theorems using the mini-batch SGLD for the sack of generality.
> > >
> > > We do appreciate the reviewer's suggestion and will include more explanations and implications of the theorems  in the next version although there are some discussions about the results for Theorem 1 (Line 203 to Line 207), Theorem 2 (Line 218 to Line 224), Theorem 3 (Line 257 to Line 265), and Theorem 4 (Line 306 to Line 309).
> > >
> > > **Q2**: I think Problem 2 is the main goal of the paper. Why did the author consider Problem 1? As far as I understand the main goal of the paper is to take a principled approach to design the covariance matrix of SGLD that leads to generalization. Note that the prior process, i.e., $\mathcal{M}$, is just an imaginary process to upper bound the generalization error. Moreover, as stated in [24] you can pick any prior distribution you want as long as it satisfies certain conditions. Therefore, I do not understand why you fix “prior” and then try to find the best covariance matrix. I do think that Problem 2 is the right problem formulation. Therefore, I appreciate it if the authors could provide some explanation why Problem 1 is interesting and relevant.
> > >
> > > Another issue that I found with Problem 1 is that why the prior is not defined based on the noise covariance structure. Note that the prior in [24] is defined for SGLD with isotropic noise structure. Therefore, it seems more reasonable that the prior use the covariance structure of the posterior here.
> > >
> > >
> > >
> > > **A2**: We thank the reviewer's thought-provoking questions. We answer them as follows.
> > >
> > >
> > > First, *why Problem 1 is interesting and why the prior is not defined on noise covariance structure?* As a first step of studying how noise structure can be optimized for good generalization, we want to start with some existing priors (for example, the prior in [24]). Moreover,  solving Problem 1 reveals an interesting correlation between the noise covariance matrices of the prior and posterior. As discussed in Line 260 to Line 265, while the optimal noise covariance is affected by the prior noise covariance, it also depends on the distance of means between prior and posterior (see Lemma 3), which brings the information of empirical gradient covariance into the optimal noise structure. That being said, the optimal posterior noise covariance is biased to the empirical gradient covariance from prior covariance whatever the prior covariance is. We note that such analysis can be easily extended to arbitrary priors.
> > >
> > >
> > > Second, *Problem 2 is the right problem formulation and should be the main goal of the paper.* Indeed, Problem 2 is the ultimate goal to solve. We thank the reviewer's suggestion and will highlight Problem 2 and its solution in the next revision.
> > >
> > >
> > > **Q3**: I could not find any comparison with SGD. I think to show the significance of the results, the authors should provide a comparison with SGD showing that the performance is comparable.
> > >
> > > **A3**: Thanks for the suggestion. We run an additional experiment of mini-batch SGD. It can be observed from the plot (can be found at https://www.dropbox.com/s/bl3lazy055kc9ac/ComparedtoSGD.png?dl=0) that the mini-batch SGD and the SGLD with the optimal covariance achieve similar generalization error, smaller than that of the SGLD with isotropic noise.  We will include this experiment in the next revision.
> > >
> > > **Q4**: The authors assume that the trace of covariance is fixed. In the end, I could not find any discussion on how the authors are going to select the values of $c_t$? Do we have a result on the convergence of SGLD so that you can theoretically study the impact of $c_t$ on the convergence?
> > >
> > >
> > > **A4**: In this work, we focus on deriving the optimal structure of noise covariance with respect to the generalization error. The analysis holds with arbitrary $c_t$, which means one can always get the optimal solution of noise covariance from Theorem 3 and Theorem 4 given $c_t$. Indeed to obtain optimal test error, one needs to trade-off between the choice of $c_t$ and the generalization error, which is beyond the scope of this paper and deserves careful study in the future.
> > >
> > > **Q5**: Example such that $\text{KL}(P||Q)$ and $\text{KL}(Q||P)$ are optimized at the same point for fixed $P$.  As far as I understand this assumption is used for solving Problem 1. Is that correct?
> > >
> > >
> > > **A5**: One nontrivial example could be as follows. Let $P$ be any multi-variant Gaussian distribution $N(\mu,\Sigma)$. Let  $Q$ be another Gaussian distribution with the same mean $\mu$ and covariance parameterized by  $\Sigma'$. Supposing that the eigenvalues of $\Sigma'$ are fixed, the minima of $\text{KL}(P||Q(\Sigma'))$ and $\text{KL}(Q(\Sigma')||P)$ are both achieved when $\Sigma$ and $\Sigma'$ can be simultaneously diagonalized, that is, there exists an orthogonal matrix $A$, such that, $A\Sigma'A^\top$ and $A\Sigma A^\top$ are diagonal, and the diagnal values of $A\Sigma'A^\top$ and $A\Sigma A^\top$ are both in non-increasing order.
> > >
> > > The reviewer's understanding is right. Assumption 1 is only for solving Problem 1. We may choose to present it in another way of replacing this assumption with a conjecture. We note this will not hurt our contribution as Problem 2 and Theorem 2 are the ultimate goal.
> > >
> > >
> > > **Q6**: Can you comment on Figure 1? Why does the green curve start converging faster but all of a sudden its performance degrades?
> > >
> > >
> > > **A6**: Figure 1 is to shows that Constraint 1 is a reasonable constraint in terms of controlling the convergence rate of the empirical risk. Hence we can then optimize the information-theoretical bound under Constraint 1.
> > >
> > > As discussed in the paper from *Line 145* to *Line 158*, the difference between the green curve and the other two curves in Figure 1 indicates the quantity $\text{tr}(\mathcal{H}_{S,W}\Sigma_t)$ proposed by [32, Theorem 4.1] is not a good constraint to govern the convergence rate, at least not as good as Constraint 1. %However, the information in the hessian may still have an influence on the generalization error, such as the escaping rate in [38].
> > >
> > > To answer the second question, we run additional experiments with different frequencies for updating the Hessian matrix $\mathcal{H}_{S,W}$ (the results can be found https://www.dropbox.com/s/qnee9y801cvp93u/Scale_and_Training.zip?dl=0). As observed in Figure Trainingfrequency100.png and Trainingfrequency10.png, the green curve (IS-SGLD (H)) does NOT always start convergence early than the rest of two. It does have a larger training error at convergence. The reason can be seen from Scalefrequency100.png and Scalefrequency10.png that IS-SGLD (H) usually has a much larger noise scale than the other two curves, which in turn verify the effectiveness of Constraint 1.
> > >
> > > **Q7**: Difference between Theorem 1 and Prop 1?
> > >
> > >
> > > **A7**: Thanks for asking. The main difference between Theorem 1 and Prop 1 is that Theorem 1 bounds the generalization error by the KL divergence between the posterior and a prior which utilizes **all** the data in $S$ to update the parameters, while Prop 1 uses a prior which utilizes **part of** the data in $S$ to update the parameters. This change helps us to show that there is only one term in the bound depending on $S$: $\sqrt{\frac{(a_2-a_1)^2}{2}KL(Q^S||P^S)}$. Then by Assumption 1, we can exchange the order of $Q^S$ and $P^S$ in the bound. In contrast, we cannot go through the above procedure for the original bound in Prop 1 due to the fact that there are dozens of terms relying on $S$.
> > > We will clarify such differences in the next revision.
> > >
> > > **Q8**: In Theorem 3, I think choosing Q as a matrix is not a good notation. Since you defined Q as a probability measure before.
> > >
> > > **A8**: Thank you for pointing out this. We take the suggestion and will use $\boldsymbol{O}^{sd}_{\boldsymbol{S},\boldsymbol{W}}$  for the matrix.
> > >
> > >
> > > **Q9**: Line 153: $\mathcal{H}$ is not defined.
> > >
> > >
> > > **A9**: It has been defined in  Line 95. We will add a note in Line 153 for easy understanding.
> > >
> > >
> > > **Q10**: What is $z_N$ in Eq. (12)? There is a typo in $J^c \sim D$ in Eq. (13)?
> > >
> > >
> > > **A10**: In Eq. (12), $z_N$ means the $N$-th element in the dataset (please refer to Line 76).  For the unity of the paper's narration, we will change $z_N$ to $z_{J^c}$ in Eq. (12). There is a typo in Eq. (13), and $J^c$ here should be $z_{J^c}$. Thanks for pointing them out, and we will revise the paper accordingly.

---

> > > > ### Comment · Reviewer_Mhnz · 2021-08-22
> > > > **Theorem 2**
> > > >
> > > > I would like to thank the author for their response and feedback.
> > > >
> > > > I read again the proof of Theorem 2, and I think I have found the error in your proof.
> > > >
> > > > Let me start with a counter example. Then, I will discuss which part of the proof is wrong.
> > > >
> > > >
> > > > # counter example:
> > > >
> > > > Consider a finite hypothesis space $\mathcal{H}=\set{h_1,\cdots,h_M}$. Let $\alpha \in (0,1)$ be a constant. Let $U$ denote a Bernoulli random variable with success probability $\alpha$. Then, given a dataset $S$ the posterior distribution is as follows. If $U=1$, then  $Q(S) = \delta_{h_{ERM}}$ , and if $U=0$ then $Q(S) = \text{uniform}(\mathcal{H}\setminus \set{h_{ERM}}) $. Note $U$ is independent of $S$ and the randomness in choosing the uniform random variable. Intuitively, the posterior with probability $\alpha$ chooses the ERM using the training set $S$ and with probability $1-\alpha$ it chooses a hypothesis uniformly at random from the set $\mathcal{H}\setminus \set{h_{ERM}}$.
> > > >
> > > > Consider a uniform prior $P=(1/M,\dots,1/M)$. Lets compute $KL(P||Q(S))$ which is the measure of the generalization in your Theorem 2. We can easily compute that $KL(P||Q(S))=\frac{1}{M} \log \frac{1}{M}  + \frac{1}{M} \log \frac{1}{1-\alpha} + \frac{M-1}{M} \log(\frac{M-1}{M \alpha})$.  In particular it can be seen that when $M \to \infty$, then $KL(P||Q(S)) \to 0 $.
> > > >
> > > > Then, lets lower bound the exact value of the generalization error for $Q(S)$. We have
> > > > $
> > > > E[R_D(Q(S)) - R_S(Q(S))] = E_{S} E_{h \sim Q(S)}[R_D(h) - R_S(h)]
> > > > $
> > > >
> > > > Then, $E_{S} E_{h \sim Q(S)}[R_D(h) - R_S(h)] >= (1-\alpha) E_S [R_D(h_{ERM})-R_S(h_{ERM})] - \alpha$
> > > >
> > > > Then, from the well-known results for the expected generalization error of the ERMs we have
> > > > $
> > > > E_S [R_D(h_{ERM})-R_S(h_{ERM})] = \Omega(\sqrt{\frac{\log(M)}{n}})
> > > > $
> > > > Therefore, for this posterior we have a lower bound on the generalization which is given by $E[R_D(Q(S)) - R_S(Q(S))] = (1-\alpha) \Omega(\sqrt{\frac{\log(M)}{n}})$. In particular note that the lower bound goes to infinity as $M \to \infty$.
> > > >
> > > > This example clearly shows that your bound in Theorem 2 is not correct since your upper bound goes to zero as $M \to
> > > > \infty$ while the lower bound goes to $\infty$ as $M \to \infty$.
> > > >
> > > > # Proof:
> > > > Then, I will discuss which step of the proof is wrong. Lines 583 and 584 of your paper is very similar to the proof of Theorem 2.5 (in appendix) of Negrea et al.  Then, you invoke the DV Lemma which states that
> > > > $$
> > > > D(P||Q) \geq P[f] - \log Q[ \exp[f]].
> > > > $$
> > > > Here you consider $P$ and $Q$ which are random measures. In particular we have $P$ is $\sigma(S_J)$ and $Q$ is $\sigma(S)$ measurable. Then, you consider the function  $f = g-Q[g]$ for DV lemma. However, the step that makes the proof wrong is
> > > > $
> > > > P[g-Q[g]] \neq  P[g] - Q[g].
> > > > $
> > > > The reason is that $Q[g]$ is not a  $\sigma(S_J)$ measurable function so that you can treat it as a constant.
> > > > I would like to remark that in the paper by Negrea et al, this step is correct because they have $KL(Q||P)$.

---

> > > > > ### Author Response · Authors · 2021-08-23
> > > > > **Response to the new counterexample and the proof misunderstanding.**
> > > > >
> > > > > As for the counterexample: for any given dataset $S$, the constructed $Q(S)$ has probability $\alpha$ at $h_{ERM}$, and probability $\frac{1-\alpha}{M-1}$ at every hypothesis in $\mathcal{H}/\{h_{ERM}\}$. Therefore, the KL divergence between $P$ and $Q(S)$ is
> > > > > $$KL(P||Q(S))=\frac{1}{M}\log \frac{1}{M\alpha}+ \frac{M-1}{M}\log \frac{M-1}{M(1-\alpha)}\rightarrow \log(1-\alpha),~as ~M\rightarrow \infty.$$
> > > > >
> > > > > Unless $\alpha=0$, the limit of the KL divergence is non-zero (contrary to the claim in the reply the limit of the KL divergence is zero). On the other hand, the generalization error can be calculated as
> > > > > $$ \mathbb{E}_{S} \mathbb{E}\_{h\sim Q(S)}( R\_{\mathcal{D}}(h)-R\_{S}(h))=\alpha\mathbb{E}\_{S}( R\_{\mathcal{D}}(h\_{ERM})-R\_{S}(h\_{ERM}))+\frac{1-\alpha}{M-1}\sum\_{i=1}^M \mathbb{E}\_{S}( R\_{\mathcal{D}}(h_i)-R\_{S}(h_i))-\frac{1-\alpha}{M-1}\mathbb{E}\_{S}( R\_{\mathcal{D}}(h\_{ERM})-R\_{S}(h\_{ERM})).$$
> > > > >
> > > > > The second term is 0 due to for fixed $i$, $\mathbb{E}\_{S}( R\_{\mathcal{D}}(h_i)-R\_{S}(h_i))=0$. Since the loss is bounded in our theorems, without loss of generality, let $\ell\in[0,1]$. Then $R\_{\mathcal{D}}(h\_{ERM})-R\_{S}(h\_{ERM})\in[-1,1]$, and  $\frac{1-\alpha}{M-1}\mathbb{E}\_{S}( R\_{\mathcal{D}}(h\_{ERM})-R\_{S}(h\_{ERM}))$ converges to zero as $M\rightarrow \infty$. Therefore,
> > > > > $$ \overline{\lim}\_{M\rightarrow \infty} \mathbb{E}_{S} \mathbb{E}\_{h\sim Q(S)}( R\_{\mathcal{D}}(h)-R\_{S}(h))= \overline{\lim}\_{M\rightarrow \infty}  \alpha\mathbb{E}\_{S}( R\_{\mathcal{D}}(h\_{ERM})-R\_{S}(h\_{ERM}))\le \alpha,$$ which is contrary to the claim in the reply that the generalization error diverges to infinity (One possible problem in the reviewer's argument is that log(M)/n is the upper bound instead of lower bound).
> > > > >
> > > > > Finally, we will show the "counterexample" actually support our Theorem 2. As $M \rightarrow \infty$, Theorem 2 gives the bound $\sqrt{\frac{-\log(1-\alpha)}{2}}$, which is always larger than the upper bound $\alpha$ obtained above (see https://www.dropbox.com/s/bjmzeq7xawcje6y/Validness%20of%20bound.png?dl=0 for illustration). Therefore, our bound is valid for this "counterexample".
> > > > >
> > > > > Also, there is a misunderstanding for the proof. We use $f(W)=\lambda(\hat{\mathcal{R}}\_{S\_{J^c}}(W)-\mathcal{R}\_{\mathcal{D}}(W)  )$, which has nothing to do with $Q$. Therefore, we don't need to calculate $P(g-Q(g))$ in the proof, and the problem raised by the reviewer will not occur.

---

> > > > > > ### Comment · Reviewer_Mhnz · 2021-08-23
> > > > > > **The application of DV**
> > > > > >
> > > > > > I will clear the confusion about my counter example in another post. Let me give more details about the step of the proof which might be wrong.
> > > > > >
> > > > > > In the first line of the proof in Theorem 2, you have
> > > > > >
> > > > > > $$
> > > > > > KL(P||Q) \geq (P-Q)(g) - \log Q(\exp(g - Q(g)))
> > > > > > $$
> > > > > > is that correct?
> > > > > >
> > > > > > From the DV lemma we have
> > > > > >
> > > > > > $$
> > > > > > KL(P||Q) \geq P(h) - \log Q(\exp(h))
> > > > > > $$
> > > > > >
> > > > > > for any function $h$. My question was what did you pick for h? and how you arrived in the first line of your proof, i.e., $
> > > > > > KL(P||Q) \geq (P-Q)(g) - \log Q(\exp(g - Q(g)))$.
> > > > > >
> > > > > > My understanding is that you followed the proof in Theorem 2.5 of Negrea et al, and you choose h = g - Q(g). Is that correct?
> > > > > >
> > > > > > If this is the case then the first line of your proof is not correct. Exactly because of the reason I mentioned earlier.
> > > > > >
> > > > > > If it is not the case, please reason about the first line of your proof using DV lemma or any other technique you have used.

---

> > > > > > > ### Author Response · Authors · 2021-08-23
> > > > > > > **The two inequality are exactly the same by setting $h=g$.**
> > > > > > >
> > > > > > > First of all, we would like to point out that the inequality $$KL(P||Q)\ge P(g)-Q(g)-\log Q(e^{g-Q(g)})~~~~~~~~~~~~~~~~~~~~~~~~~~~~~~~~~~~~~~~~~~~~~~~~~~~~~~~~~~~~~~~~~~~~~~~~~~~~~~~~~~~~~~~~~~~~~~~~~~~~~~(1)$$ is exactly equivalent with the inequality given by the DV Lemma $$ KL(P||Q)\ge P(g)-\log Q(e^g).$$ Actually, $$KL(P||Q)= \sup\_{g: Q(e^g)<\infty} P(g)-Q(g)-\log Q(e^{g-Q(g)})$$ is an equivalent formulation of the DV Lemma.
> > > > > > >
> > > > > > >  The reason is $$ -Q(g)-\log Q(e^{g-Q(g)})=-Q(g)-\log e^{-Q(g)}\cdot Q(e^{g})=-Q(g)+Q(g)-\log Q(e^g)=-\log Q(e^g).$$ (Here we use the fact that $Q(g)$ is a constant with respect to the parameter $W$.) Therefore, the first inequality in the "Proof of Theorem 2.5" in Negrea et al., i.e.,  $$KL(Q(S,U)||P(S_J,U))\ge \sup\_{g\in \mathcal{G}}(Q(S,U)(g)-P(S_J,U)(g)-\log(P(S_J,U)(e^{g-P(S_J,U)(g)})))$$ can be obtained by setting $P=Q(S,U)$ and $Q=P(S_J,U)$  and taking supremum with respect to $g$ in (1) (don't need to set $g=g-Q(S,U)(g)$). On the other hand, the inequality used in our paper
> > > > > > > $$KL(P^{S_J,V\_{[T]}}||Q^{S, V\_{[T]}})\ge P^{S_J,V\_{[T]}}(g)-Q^{S, V\_{[T]}}(g)-\log Q^{S, V\_{[T]}}(e^{g-Q^{S, V\_{[T]}}(g)})$$ is obtained by setting $P=P^{S_J,V\_{[T]}}$ and $Q=Q^{S, V\_{[T]}}$ in (1).
> > > > > > >
> > > > > > > Secondly, we would like to point out for both the inequality in Line 584 and the first inequality in the "Proof of Theorem 2.5" in Negrea et al., all $S,J, V\_{[T]}$, and $U$ are fixed, and $P^{S_J,V\_{[T]}}$, $Q^{S, V\_{[T]}}$, $P(S_J,U)$ and, $Q(S,U)$ are distributions over $W$(just as $P$ and $Q$), therefore, we indeed have $P^{S_J,V\_{[T]}}(g-Q^{S, V\_{[T]}}(g))=P^{S_J,V\_{[T]}}(g)-Q^{S, V\_{[T]}}(g)$ as here $Q(g)$ is a constant with respect to the parameter $W$ (while we don't use this fact).

---

> > > > > > > > ### Comment · Reviewer_Mhnz · 2021-08-23
> > > > > > > > **Finally I understand the proof**
> > > > > > > >
> > > > > > > > Dear Authors,
> > > > > > > >
> > > > > > > > I would like to thank you for your answers.
> > > > > > > >
> > > > > > > > I read again the proof and now I understand what is going on.
> > > > > > > >
> > > > > > > > Best!

---

> > > > > > > > > ### Author Response · Authors · 2021-08-24
> > > > > > > > > **We are glad that your concerns are addressed.**
> > > > > > > > >
> > > > > > > > > Dear reviewer,
> > > > > > > > >
> > > > > > > > > Are there any other concerns? Would you please consider increasing the score if your concerns have been well addressed?
> > > > > > > > >
> > > > > > > > > Best!

---

> ### Comment · Reviewer_Mhnz · 2021-08-26
> **My comments after the rebuttal:**
>
> 1- I think the paper is interesting, and I think the problem formulation and the solutions are novel.
>
> 2- The writing of the paper can be greatly improved. There are lots of typos. Also, the expectation notation in the paper is not consistent; the authors should either use P[f]=int f dP or E_{X~P} f(X) and not both. For instance in the Proof of Theorem 2, the authors use both notations.
>
> 3- I still have the same concerns regarding the problem formulation 1. After reading the author's response, I think it is still interesting to consider Problem 1. However, it should be highlighted that Problem 2 is the right formulation. Also, the authors should provide a detailed discussion of the similarity and differences between the optimal covariance and the results in [32]. For instance the authors should discuss what we learn about the structure of the optimal prior for Problems 1 and 2.
>
> 4- I did find the experimental section rushed and unsatisfactory. More details, and discussion would be appreciated. It would also be good to discuss the intuition behind the structure of the noise and SGD noise. Also, is it possible to compare the performance of LD with optimal covariance with the SGD in terms of both generalization error as well as empirical error? I think some preliminary studies over some simple models and datasets can greatly improve the paper.
>
>
> I increased my score to 6. I would like to hear the author's detailed plan for the revision of the paper. Also, if accepted, there is an additional page that the author can use to address the following concern.

---

> > ### Author Response · Authors · 2021-08-27
> > **We would like to thank the reviewer for the comments of revising.**
> >
> > First, we would like to thank the reviewer for the responsible review and responsive reply, which finally helps to clarify the value of this paper. The comments for paper revision are very useful, and we discuss them respectively as follows:
> >
> > **Q1**:  I think the paper is interesting, and I think the problem formulation and the solutions are novel.
> >
> > **A1**: Thank the reviewer for acknowledging our problem and solutions.
> >
> > **Q2**: The writing of the paper can be greatly improved. There are lots of typos. Also, the expectation notation in the paper is not consistent; the authors should either use $P[f]$ or $\mathbb{E}_{X\sim P} f(X)$ and not both.
> >
> > **A2**: Thanks for pointing out. We will correct the typos accordingly and make sure the notations consistent throughout the context. Specifically,  in the Proof of Theorem 2, we will use $P[f]$ as the expectation of function $f$ with respect to distribution $P$ and replace the $\mathbb{E}_{X\sim P} f(X)$ with  $P[f]$.
> >
> >
> >
> > **Q3**: Regarding to the Problem 1 and Problem 2.
> >
> > **A3**: Thanks for the suggestion. We plan to rearrange the paper as follows: We will propose Problem 2 as our ultimate goal in Section 3, and give Theorem 2 in Section 4 and solve Problem 2 in Section 5. Then we will propose Problem 1 as an interesting extension and solve it in Section 6.
> >
> > Also, we will provide a detailed discussion between the noise structure proposed in [32,37] and our optimal noise structure with an additional section from both the theoretical and empirical points of view.
> >
> >
> >
> > **Q4**: I did find the experimental section rushed and unsatisfactory. More details and discussion would be appreciated. It would also be good to discuss the intuition behind the structure of the noise and SGD noise.Is it possible to compare the performance of LD with optimal covariance with the SGD in terms of both the generalization error as well as the empirical error? I think some preliminary studies over some simple models and datasets can greatly improve the paper.
> >
> >
> >
> > **A4**: Thanks for the suggestion. We will provide a more detailed discussion and intuition in the experimental section. We will also include new experiments to compare the generalization error and empirical risk between SGD and SGLD with optimal noise covariance (some preliminary results on Fashion-MNIST can be found here https://www.dropbox.com/s/92d83c8ct49n3hx/Compared_to_SGD.zip?dl=0 with both the generalization error and the training loss curves, and we will conduct the experiments on more datasets with more models).
> >
> > We will add some preliminary studies over some simple models and datasets. Specifically, we will add a figure plotting the generalization error of SGD, SGLD with isotropic noise, and SGLD with our optimal noise covariance in the Introduction to help the readers quickly grasp the results of this paper.

---

### Official Review · Reviewer_DUnn · 2021-07-19

**Rating:** 6
**Confidence:** 2

**Summary:**

From usual isentropic noise to data-dependent priors, the authors extend the existing information-theoretic analysis of SGLD. Additionally, they identify the optimal choice of prior distribution covariance. Last but not least, the paper includes empirical observations to validate its technical findings.

**Ethics Review Area:**

["I don’t know"]

**Limitations And Societal Impact:**



**Main Review:**

I must admit that I am not an expert in this field. Nevertheless, the paper is very well-written, the literature review is extensive, and the technical contributions appear to be solid. Bellow you can find my comments.

- Is there a regularity condition (such as having a full-rank matrix) for the covariance matrix \Sigma_{S, W}^{sd}?

- How do you compare the value of \tilde \sigma_{i}^{S, W} and \sigma_{i}^{S, W}? Which one is larger, and can we conclude that the condition number of \Sigma_t^\star is larger than the condition number of \Sigma_{S, W}^{sd}?

- Does \Sigma_t^\star converge to a certain matrix as t tends to infinity?

**Time Spent Reviewing:**

12

---

> ### Author Response · Authors · 2021-08-10
> **Thanks for your feedback. Concerns are addressed below.**
>
>
> **Q1**: Is there a regularity condition (such as having a full-rank matrix) for the covariance matrix $\Sigma_{S, W}^{sd}$?
>
> **A1**: Thanks for the question. The answer is no, and the analysis still holds if the covariance matrix $\Sigma_{S, W}^{sd}$ or $\Sigma_{S, W}^{pop}$ is not full-rank.
>
> **Q2**: How do you compare the value of $\tilde \sigma_{i}^{S, W}$ and $\sigma_{i}^{S, W}$? Which one is larger, and can we conclude that the condition number of $\Sigma_t^\star$ is larger than the condition number of $\Sigma_{S, W}^{sd}$?
>
> **A2**:  The value of the $i$-th eigenvalue $\tilde \sigma_{i}^{S, W}$ of the optimal noise covariance $\Sigma_t^*$ is not comparable to the $i$-th eigenvalue $ \sigma_{i}^{S, W}$ of the SGD noise covariance $\Sigma_{S, W}^{sd}$. The reason is that the  optimal $\Sigma_t^*$ is affected by the prior noise and the  posterior noise covariance, which are freely chosen, while $\Sigma^{sd}_{S, W}$ is  determined by the data $S$ and the weight $W$.
>
> However, it can be shown that the condition number of $\Sigma_t^*$ is smaller than $\Sigma_{S, W}^{sd}$. Firstly, the noise covariance of the prior is isotropic, has condition number $1$, and push the condition number of  $\sigma_t\mathbb{I}+\frac{\eta_t^2}{Nb_t}\left(\frac{N}{N-1}\right)^2\Sigma_{S, W}^{sd}$ smaller than $\Sigma_{S, W}^{sd}$. Secondly, the optimal solution $G$ of Lemma 4 always has a smaller condition number than $B$, which implies that $\Sigma_t^*$ has smaller condition number than $B=\sigma_t\mathbb{I}+\frac{\eta_t^2}{Nb_t}\left(\frac{N}{N-1}\right)^2\Sigma_{S, W}^{sd}$. Hence the condition number of $\Sigma_t^*$ is smaller than $\Sigma_{S, W}^{sd}$. We will add the above discussion in next version.
>
> **Q3**: Does $\Sigma_t^\star$ converge to a certain matrix as $t$ tends to infinity?
>
> **A3**: $\Sigma_t^\star(S, W)$ is determined by  $\Sigma_{S,W} ^{sd}$, $\sigma_t$, and $c_t(S, W)$. Therefore, the convergence of $\Sigma_t^\star$ also depends on the convergence of $\Sigma_{S, W}^{sd}$, $\sigma_t$, and $c_t(S, W)$. Specifically, if  $\sigma_t$ and $c_t(S, W)$ stay as constants and $\Sigma_{S, W}^{sd}$ converges, then $\Sigma_t^\star(S, W)$ will converge.

---

### Official Review · Reviewer_ha8D · 2021-08-11

**Rating:** 7
**Confidence:** 4

**Summary:**

This interesting paper studies the connection between generalization abilities of of SGLD and the covariance structure of its noise term. It first proves that with constraint to guarantee low empirical risk and commonly used data-dependent priors, the optimal noise covariance of SGLD in terms of the bound is similar to the empirical gradient covariance. The paper further proves that if the generalization bound is jointly optimized with respect to both prior and posterior, the optimal noise covariance is the square root of the expected gradient covariance. These facts can be used to further support the belief of the superiority of SGD noise over isotropic noise.

**Main Review:**

This paper studies the connection between generalization abilities of of SGLD and the covariance structure of its noise term. It proves that with constraint to guarantee low empirical risk and commonly used data-dependent priors, the optimal noise covariance of SGLD in terms of the bound is similar to the empirical gradient covariance. The paper further proves that if the generalization bound is jointly optimized with respect to both prior and posterior, the optimal noise covariance is the square root of the expected gradient covariance. Overall, the concern of optimization methods with generalization abilties from the viewpoint of noise covariance structure is novel, interesting and important. It is important because these facts can be used to further support the belief of the superiority of SGD noise over isotropic noise, an important theme towards the understanding of implicit regularization of SGD.

**Time Spent Reviewing:**

6 hours

---

> ### Author Response · Authors · 2021-08-16
> **Thanks for your positive feedback.**
>
> We would like to thank the reviewer for the encouraging feedback. Indeed, our paper is the first work to study the optimal noise covariance of SGLD with respect to the generalization error through the information-theoretical bounds, which further demonstrates the benefits of the SGD-like noise structure. We believe this work will bring new perspectives/tools to the community.

---

### Decision · Program_Chairs · 2021-09-27

**Decision:**

Accept (Poster)

**Comment:**

The paper studies the connection between the generalization ability of SGLD and the covariance structure of its noise term. The reviewers initially had a lot of concerns about the lack of clarity of the paper and some details in the derivation of the proofs. Most of these concerns were addressed by the authors in the rebuttal and some reviewers raised their scores as a result. Overall, the contribution made in the paper is seen as interesting and worthy of acceptance. However, many reviewers, and myself included, think that the writing in some sections of the paper and the clarity are poor. I strongly advise the authors to follow the suggestions made by the reviewers to improve the manuscript for the camera-ready version.